# Compressed Over-parameterized Federated Learning for Multiple Access Channels

## Abstract

Federated Learning (FL) is a distributed machine learning (ML) paradigm that addresses user data privacy concerns. Here, a global ML model is learned by aggregating local models that were learned over local data at each edge user (also known as a client). Realizing the benefits of FL is challenging, particularly in communication-constrained environments, such as the Internet of Things (IoT) framework and wireless communication characterized by low bandwidth links over wireless physical channels. A well-known FL protocol over such resource-constrained channels is FL over multiple access channels, also known as FL-MAC, where edge users use the transmission medium simultaneously, hence avoiding the need for orthogonal resources. However, the communication bottleneck at the server in FL can still get choked since modern-day neural networks (NNs) are over-parameterized. Over-parameterized neural networks (ONNs) are trained in the lazy training regime, where the model weights of the NN change very slowly across gradient descent epochs. This motivates the use of incremental model weights. Since such updates are highly sparse, this allows for algorithms that employ compressive sensing (CS), thus allowing compressed model update communication. Accordingly, we propose Compressed Over-parameterized Federated Learning over MAC (or COFL-MAC). We employ a common Gaussian sensing matrix as the dictionary to compress the per-user model updates. By means of NTK theory, we show that the COFL-MAC framework exhibits exponential convergence in addition to being communication efficient. Using the CIFAR-10 and FMNIST datasets, we empirically demonstrate that the proposed framework outperforms the gradient compression benchmark strategies - Top-k (with correction), SignSGD, and MQAT, in terms of communication efficiency for a given test accuracy for different data heterogeneity levels among the clients.

## 1 Introduction

Increased computational capabilities of the devices at the edge and the availability of data for machine learning (ML) have triggered efforts toward enhancing user experience on edge devices. Resource and data privacy constraints trigger the idea of distributed learning approaches such as federated learning (FL). FL is a distributed ML framework consisting of edge devices, referred to as *clients*, and a central coordinating entity, referred to as the *server*. The client observes private data at the edge, learns a local ML model, and communicates this model to the server. The server then aggregates the models conveyed by the clients and broadcasts the updated global model to the clients. For a multi-user system with a server and $C$ clients, each having a private dataset $\mathcal{S}_c$ consisting of $D_c$ samples, the global optimization problem in FL is defined as

$$\min_{\mathbf{u}(t)\in\mathbb{R}^N} F(\mathbf{u}(t)) \triangleq \frac{1}{C}\sum_{c=1}^{C} f_c(\mathbf{u}(t)), \tag{1}$$

where $f_c(\cdot)$ is the local optimization problem at the $c$-th client and $\mathbf{u}(t)$ represents the global parameters in the $t$-th communication round, where $t \in [T]$. A popular aggregation algorithm for FL is federated averaging

(FedAvg) McMahan et al. (2017), where the global parameters in the $t$-th round are obtained as

$$\mathbf{u}(t) = \frac{1}{C} \sum_{c=1}^{C} \mathbf{w}_c(t), \tag{2}$$

where $\mathbf{w}_c(t)$ represents the local parameters of the $c$-th client.

In the era of foundation models such as GPT-3, which has 175 billion parameters, neural network (NN) based ML models tend to be over-parameterized, where the number of parameters ($N$) is much larger than the number of data points. In FL, the condition for over-parameterization occurs at the edge-user level, i.e., $N \gg \max_c\{D_c\}$. In general, NN objective functions at the edge are highly non-convex; thus, the convergence of the gradient descent or stochastic gradient descent ((S)GD) algorithms to the global optimum is not guaranteed Karimireddy et al. (2020). However, recent studies indicate that if the edge NNs are over-parameterized, (S)GD converges exponentially to the global optimum with a high probability Huang et al. (2021a); Song et al. (2023). A comprehensive survey on convergence guarantees for FL in modern over-parameterized settings is provided in Haddadpour & Papailiopoulos (2021).

## 1.1 Over-parameterized FL and FL-NTK

In an FL setup, the amount of data is low at individual edge devices, inevitably leading to client-level over-parameterization. Therefore, adequate motivation exists to analyze the convergence of over-parameterized neural networks (ONNs) in an FL setup. As local convexity does not apply to ONNs, the earlier convergence proofs requiring this assumption, Li et al. (2020b), Karimireddy et al. (2020) cannot be used. Neural Tangent Kernel (NTK) theory is widely applicable to ONNs, offering a unified analysis approach Huang et al. (2021a). NTK theory has been extended to prove exponential convergence in FL-based NTK framework (over-parameterized FL setup) Huang et al. (2021a), in FedBN (FL with local batch normalization) Li et al. (2021), and recently extended in non-iid and multi-layer settings Song et al. (2023). The NTK framework has further been extended to study gradient diversity in heterogeneous clients Li et al. (2023), multi-task personalization in federated settings Du et al. (2022), and kernel compression methods for scalability Wang & Zhang (2023). Federated kernel regression, leveraging NTK-based fixed features, has also been proposed for decentralized systems in Xu et al. (2022).

It is known that the model weights in an ONN change very slowly across the gradient descent iterations due to lazy training Chizat et al. (2019). According to NTK dynamics, the relative change in the norm of model weights formed at a hidden layer in the NN is of the order $\mathbb{O}(1/\sqrt{M})$, where $M$ is the number of neurons in the hidden layer Wang et al. (2022). Therefore, if the clients transmit incremental weight updates to the server, the transmitted vectors will be sparse, allowing for compression.

## 1.2 Overparametrized FL-MAC

The immense capabilities offered by ONNs need to be considered with a pinch of salt, especially in a communication-constrained distributed ML setup such as FL, due to the exchange of many parameters between clients and the server. A basic technique to improve communication efficiency in FL is to perform multiple local steps before transmitting the parameters to the server McMahan et al. (2017). Further, the dynamic range of updates is often controlled by transmitting incremental parameter updates, i.e., $\Delta \mathbf{w}_c(t) = \mathbf{w}_c(t) - \mathbf{u}(t-1)$.

In order to avoid multiplexing, and thereby improve the throughput in an FL setup, FL over a multiple access channel (FL-MAC) is a preferred choice for efficient communication over a common uplink MAC Wei et al. (2021); Sery & Cohen (2021); Chang & Tandon (2020), as it enables simultaneous analog transmission of client updates in a non-orthogonal manner. For noiseless FL-MAC, the signal received at the server is:

$$\Delta \mathbf{u}'(t) = \sum_{c=1}^{C} \Delta \mathbf{w}_c(t), \tag{3}$$

and the global update becomes:

$$\mathbf{u}(t) = \mathbf{u}(t-1) + \frac{1}{C}\Delta\mathbf{u}'(t). \tag{4}$$

Despite the throughput gains, over-parameterized FL-MAC can overwhelm communication pipelines. To address this, gradient compression strategies such as quantization and sparsification Oh et al. (2022); Haddadpour et al. (2021); Zhang et al. (2024) are widely adopted. Quantization includes scalar Liu et al. (2023); Lin et al. (2021) and vector techniques Oh et al. (2023); Shlezinger et al. (2020). Sparsification methods drop lower-magnitude gradient entries, further enhancing compression Li et al. (2020a); Lin et al. (2018).

Recently, over-the-air analog FL (OTA-FL) has been explored using rate-distortion theory Chen et al. (2023), power-efficient aggregation schemes Zhang et al. (2023), and adaptive sparsification under fading channels Li & Wang (2023). Robust analog aggregation under practical hardware constraints is also studied in Amiri & Gündüz (2021). However, none of these works on wireless, and OTA-FL handles ONNs. Further, none of the prior works in FL investigate NTK-based convergence guarantees of compressed ONNs over MAC in the interpolation regime. This paper addresses these gaps.

## 1.3 Contributions

Since ONNs have a large number of parameters, employing ONNs in a distributed setup such as FL leads to higher communication costs and latency violations. Compression techniques help reduce the communication overhead Li et al. (2020a), Oh et al. (2023; 2022), Amiri & Gündüz (2020),Oh et al. (2022). In this work, we propose the use of compressive sensing (CS) for sparsifying the ONN-based analog gradient updates for communication-efficient FL over MAC, which we refer to as *Compressed Over-parameterized Federated Learning framework for MAC channels (COFL-MAC)*. At the end of local training, each client generates incremental model updates, which are sparse in the case of ONNs. Thus, CS is employed at the clients to compress these sparse incremental model updates over the uplink MAC, i.e., the $c$-th client transmits a compressed incremental weight update, $\Delta\widetilde{\mathbf{w}}_c(t)$, instead of $\Delta\mathbf{w}_c(t)$, obtained as follows:

$$\Delta\widetilde{\mathbf{w}}_c(t) = \mathbf{A}\Delta\mathbf{w}_c(t), \tag{5}$$

where $\mathbf{A}$ is an over-complete Gaussian sensing matrix. Essentially, $\mathbf{A}$ projects a higher-dimensional vector, $\Delta\mathbf{w}_c(t)$ onto a lower-dimensional space to obtain $\Delta\widetilde{\mathbf{w}}_c(t)$. Transmitting the lower-dimension update ensures communication efficiency. The main contributions of this paper are summarized as follows:

- We propose a framework that employs ONNs for improved generalization. To address the challenges of communication efficiency, we employ CS at the clients, which enables the deployment of ONNs. We call this setup as COFL-MAC.

- We provide convergence guarantees for the COFL-MAC framework. In particular, using NTK theory for a ReLU-based ONN, we demonstrate that COFL-MAC exhibits exponential convergence.

- We demonstrate that the COFL-MAC framework has a much lower communication cost than some well-established gradient compression benchmark strategies in an FL setup for the same test accuracy. Further, our proposed setup achieves considerably higher accuracy for an under-parameterized FL setup with the same communication cost.

Specifically, our experimental results on real-world datasets, namely FMNIST and CIFAR-10, demonstrate that the proposed COFL-MAC approach outperforms the state-of-the-art benchmark strategies such as Top-$k$, SignSGD, and MQAT in terms of communication cost over the uplink. An important takeaway from our experiments is that COFL-MAC transmits the same number of parameters over a MAC channel as an under-parameterized NN yet achieves higher accuracies. To the best of the authors' knowledge, this is the first work that proposes the deployment of ReLU-based ONNs in a CS-based FL-MAC setup employing NTK theory-based convergence guarantees.

The rest of this paper is organized as follows: In 2, we propose a compression approach that allows the deployment of ONN in an FL-MAC setup and discuss the proposed COFL-MAC framework. In 3 we analyze

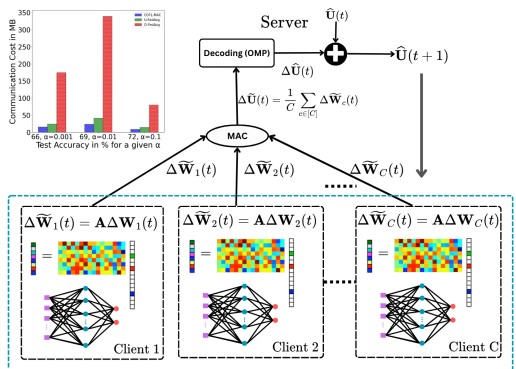

Figure 1: This figure depicts the COFL-MAC process and its communication efficiency. **Upper Left:** This bar plot shows that COFL-MAC has the best communication efficiency among under-parameterized FedAvg (U-FedAvg), over-parameterized FedAvg (O-FedAvg), and COFL-MAC for non-IID data distribution among the clients for FMNIST dataset for a given data heterogeneity ($\alpha$) among the clients. **Lower Right:** Implementation of COFL-MAC setup showing the enabling of wide ONNs at the clients via compressive sensing and the sparse signal recovery (SSR) at the server post aggregation. Only the uplink communication is compressed.

**Algorithm 1** : COFL-MAC training under NTK setting.

1: **function** INITIALIZATION
2: $\quad$ $\mathbf{u}_m(0) \sim \mathcal{N}(0, \mathbf{I}_N)$ for $m \in [M]$.
3: **function** SERVER_PROCESS
4: $\quad$ **for** $t = 0, \ldots, T$ **do**
5: $\quad\quad$ broadcast $\widehat{\mathbf{U}}(t)$ to the clients
6: $\quad\quad$ **for** $c = 1, \ldots, C$ **do**
7: $\quad\quad\quad$ $\Delta \widetilde{\mathbf{W}}_c(t) \leftarrow$ **Client_Update**$(\Delta\widehat{\mathbf{U}}(t))$
8: $\quad\quad$ $\Delta\widetilde{\mathbf{U}}(t) \leftarrow \frac{1}{C} \sum_{c \in [C]} \Delta\widetilde{\mathbf{W}}_c(t)$
9: $\quad\quad$ $\Delta\widehat{\mathbf{U}}(t) \leftarrow SSR(A, \Delta\widetilde{\mathbf{U}}(t))$
10: $\quad\quad$ $\widehat{\mathbf{U}}(t+1) \leftarrow \widehat{\mathbf{U}}(t) + \Delta\widehat{\mathbf{U}}(t)$
11: $\quad\quad$ $t \leftarrow t + 1$
12: **function** CLIENT_UPDATE$(\Delta\widehat{\mathbf{U}}(t))$
13: $\quad$ $\Delta\mathbf{W}_c(t) \leftarrow$ **Client_Training**$(\widehat{\mathbf{U}}(t))$
14: $\quad$ $\Delta\widetilde{\mathbf{W}}_c(t) \leftarrow \mathbf{A}\Delta\mathbf{W}_c(t)$
$\quad$ **return** $\Delta\widetilde{\mathbf{W}}_c(t)$ to server
15: **function** CLIENT_TRAINING$(\widehat{\mathbf{U}}(t))$
16: $\quad$ $\mathbf{W}_c^0(t) \leftarrow \widehat{\mathbf{U}}(t)$
17: $\quad$ **for** $k = 1, \ldots, K$ **do**
18: $\quad\quad$ **for** $m = 1, \ldots, M$ **do**
19: $\quad\quad\quad$ $\mathbf{w}_{c,m}^k(t) \leftarrow \mathbf{w}_{c,m}^{k-1}(t) - \eta \frac{\partial \mathcal{L}_c(\mathbf{W}_c^k(t))}{\partial \mathbf{w}_{c,m}^k(t)}$
20: $\quad$ $\Delta\mathbf{W}_c(t) \leftarrow \mathbf{W}_c^K(t) - \mathbf{W}_c^0(t)$
$\quad$ **return** $\Delta\mathbf{W}_c(t)$

the convergence of COFL-MAC using NTK theory. We subsequently provide discussions and present the experimental results in 4 and 5 respectively. Finally, we conclude our work in 6.

**Notation**: In the sequel, boldface small letters denote vectors, and boldface capital letters denote matrices. The symbols $(\cdot)^T$ and $\lambda_{min}(\cdot)$ denote the transpose of a matrix and the minimum eigenvalue of a matrix, respectively. Further, $\mathbf{I}_M$ denotes an $M \times M$ identity matrix, and the indicator function is denoted as $\mathbf{1}_a$ where the value is 1 if $a > 0$. The complement of an event $E$ is denoted using the symbol $\neg$ as $\neg E$. The symbol $\vee$ stands for logical or, and the symbol $\wedge$ stands for logical and. The symbol $\|$ stands for concatenation operator; e.g., $\|_{c=1}^C \mathbf{y}_c$ denotes concatenation of vectors $\mathbf{y}_1 \cdots \mathbf{y}_C$.

## 2 Deploying ONNs in an FL-MAC setup

In order to deploy ONNs in an FL-MAC setup, it is essential to compress the parameter updates to facilitate communication over the resource-constrained uplink channel between the clients and the server. Since the model weights in an ONN change very slowly across the gradient descent iterations due to lazy training Chizat et al. (2019), the transmitted incremental weight updates are sparse, allowing for compression. ONNs obey NTK dynamics at infinite width assumptions; therefore, the NTK theory can be used to study and prove the convergence of ONNs. In this section, we provide an overview of the established NTK theory based FL Huang et al. (2021a) convergence theory, followed by the proposed COFL-MAC algorithm.

### 2.1 FL-NTK Framework

In this section, we describe the basic NN framework followed by SGD-based learning and the client dynamics based on NTK theory in FL (or vanilla FL). As depicted in Fig. 1, we consider a multi-user system with a server and $C$ clients. Each client has access to a private dataset $\mathcal{S}_c = \{(\mathbf{x}_{c,i}, y_{c,i})\}_{i=1}^{D_c}$, where the features

are denoted as $\mathbf{x}_{c,i} \in \mathbb{R}^N$ and the labels are given by $y_{c,i} \in \mathbb{R}$. We assume that the model is parameterized by $\mathbf{U}(t) \in \mathbb{R}^{N \times M}$ in the communication round $t \in [T]$. We denote $\mathbf{y}_c \in \mathbb{R}^{D_c}$ as the ground truth labels at the $c$-th client. Furthermore, we denote $\mathbf{y}_c^k(t) \in \mathbb{R}^{D_c}$ to be the (local) model's prediction in the $t$-th global round and $k$-th local step, i.e., given the input data point $\mathbf{x}_{c,i}$, the prediction of the local model is given by $y_{c,i}^k(t)$. Also note that $\mathbf{y} \in \mathbb{R}^D$ denotes the concatenation of all local ground truth vectors $\mathbf{y}_c \in \mathbb{R}^{D_c}$ and is given by $\mathbf{y} = \|_{c=1}^C \mathbf{y}_c$. Similarly, $\mathbf{y}(t) \in \mathbb{R}^D$ is the concatenation of all local prediction vectors in the $t$-th communication round and is given by $\mathbf{y}(t) = \|_{c=1}^C \mathbf{y}_c(t)$.

We consider a single $M$-width hidden layer ONN at the clients defined as $g(\cdot, \cdot) : \mathbb{R}^{N \times M} \to \mathbb{R}$. Let the matrix $\mathbf{W}_c^k(t) \in \mathbb{R}^{N \times M}$ represent the NN weights at the interface of the input layer and the hidden layer in the $k$-th local step of the $t$-th communication round, and $\phi(\cdot)$ denotes the ReLU activation function and is given by $\phi(z) = \max(0, z)$. At the $c$-th client, given an input $\mathbf{x}_{c,i}$, we have the prediction, $y_{c,i}(t) = g(\mathbf{W}_c^k(t), \mathbf{x}_{c,i})$, given by Huang et al. (2021a):

$$y_{c,i}(t) = \frac{1}{\sqrt{M}} \sum_{m=1}^M b_m \phi(\mathbf{x}_{c,i}^T \mathbf{w}_{c,m}^k(t)), \tag{6}$$

where $\mathbf{w}_{c,m}^k(t) \in \mathbb{R}^N$ denotes the $m$-th column of the weight matrix $\mathbf{W}_c^k(t)$, and it represents the weights between the $N$ input nodes and the $m$-th node in the hidden layer. We make the following two assumptions regarding the magnitude of $b_m$ and $\mathbf{x}_{c,i}$.

**Assumption 1.** $\|b_m\|_2 = 1$. *This is because $b_m$ is sampled from $\{-1, +1\}$ uniformly at random $\forall m \in [M]$.*

**Assumption 2.** *Data is normalized, i.e., $\|\mathbf{x}_{c,i}\|_2 = 1, \forall i \in [D_c]$.*

Typically, a squared loss function is assumed while incorporating the NTK kernel, and hence, at each of the clients, we have the local loss function given by

$$\mathcal{L}_c(\mathbf{W}_c) := \frac{1}{2} \sum_{i=1}^{D_c} (y_{c,i}(t) - y_{c,i})^2. \tag{7}$$

Hence, with $\eta$ as the learning rate at the client (local epochs), the gradient descent (GD) update at the $k+1$-th local epoch is given by

$$\mathbf{W}_c^{k+1}(t) \leftarrow \mathbf{W}_c^k(t) - \eta \frac{\partial \mathcal{L}_c(\mathbf{W}_c^k(t))}{\partial \mathbf{W}_c^k(t)}, \tag{8}$$

where each column of $\dfrac{\partial \mathcal{L}_c(\mathbf{W}_c^k(t))}{\partial \mathbf{W}_c^k(t)}$ is obtained as :

$$\frac{\partial \mathcal{L}_c(\mathbf{W}_c^k(t))}{\partial \mathbf{w}_{c,m}^k(t)} = \frac{1}{\sqrt{M}} \sum_{i=1}^{D_c} (y_{c,i}^k(t) - y_{c,i}) b_m \mathbf{x}_{c,i} \mathbf{1}_{\mathbf{x}_{c,i}^T \mathbf{w}_{c,m}^k}. \tag{9}$$

We will place an upper bound on the above gradient in the upcoming lemma as this bound will be incorporated in the sequel.

**Lemma 1.** *The norm of the loss function function gradient at the $c$-th client in the $k$-th local epoch is bounded as follows:*

$$\left\| \frac{\partial \mathcal{L}_c(\mathbf{W}_c^k(t))}{\partial \mathbf{w}_{c,m}^k(t)} \right\|_2 \leq \frac{\sqrt{D}}{\sqrt{M}} \left\| \mathbf{y}_c^k(t) - \mathbf{y}_c \right\|_2. \tag{10}$$

Therefore, the norm of the gradient of the local loss function at a client is of the order $\mathcal{O}(1/\sqrt{M})$.

Next, let us proceed towards writing the FedAvg rule for FL, where each client $c$ shares an incremental model update, $\Delta \mathbf{W}_c(t)$, given by

$$\Delta \mathbf{W}_c(t) = \mathbf{W}_c^K(t) - \mathbf{W}_c^0(t), \tag{11}$$

where $\mathbf{W}_c^0(t) = \mathbf{U}(t)$. The parameter aggregation at the server using the FedAvg algorithm is then given by

$$\Delta \mathbf{U}(t) = \frac{1}{C} \sum_{c=1}^{C} \Delta \mathbf{W}_c(t). \tag{12}$$

Accordingly, the global update at the server is given by

$$\mathbf{U}(t+1) \leftarrow \mathbf{U}(t) + \Delta \mathbf{U}(t). \tag{13}$$

The movement in the predictions across FL rounds is the outcome of local updates at the clients and subsequent aggregation. The following lemma provides the prediction dynamics at the $c$-th FL client based on the prediction dynamics in a centralized ML setup as given in Du et al. (2018); Arora et al. (2020).

**Lemma 2.** *The dynamics of prediction at the c-th FL client under gradient flow update rule for any local epoch $k$ can be expressed as*

$$\frac{d}{dk} \mathbf{y}_c^k(t) = \mathbf{J}_c^k(t)(\mathbf{y}_c - \mathbf{y}_c^k(t)), \tag{14}$$

where the $m$-th column of the local FL-NTK Gram matrix, $\mathbf{J}_c^k(t)$ is given by

$$\mathbf{J}_{c,m}^k(t) = \left\langle \frac{\partial g(\mathbf{W}_c^k(t))}{\partial \mathbf{w}_{c,m}^k(t)}, \frac{\partial g(\mathbf{W}_c^k(t))}{\partial \mathbf{w}_{c,m}^k(t)} \right\rangle. \tag{15}$$

The result in Lemma 2 is one of the critical results essential to establish the exponential convergence of the NTK-based quadratic loss (Theorem 4.1 in Du et al. (2018), Theorem 3.7 in Song & Yang (2019)).

The exponential convergence in the centralized NTK framework from Du et al. (2018) and Song & Yang (2019) follows from the above lemma and can be directly used for establishing exponential decay of the prediction error during local training at the $c$-th client in a distributed FL setup as follows:

$$\|\mathbf{y}_c^k(t) - \mathbf{y}_c\|_2^2 \le \left(1 - \frac{\eta \lambda_c(t)}{2}\right)^k \|\mathbf{y}_c^0(t) - \mathbf{y}_c\|_2^2, \tag{16}$$

where $\lambda_c(t)$ denotes the minimum eigenvalue of $\mathbf{J}_c^k(t)$, and $\mathbf{y}_c^0(t)$ is the prediction for the first epoch of local training in the $t$-th global round. The above expression shows that in an FL setup, the training loss at each client decays exponentially across local epochs. The speed of this decay is dictated by the values of $\lambda_c(t)$, $\eta$, and $k$. Further, the above result is extended to a wide over-parameterized FL setup in Huang et al. (2021a), where the authors show that the global quadratic loss shrinks exponentially in $t$ FL rounds w.r.t. initialization as follows:

$$\|\mathbf{y}(t) - \mathbf{y}\|_2^2 \le \left(1 - \frac{\lambda \eta K}{2C}\right)^t \|\mathbf{y}(0) - \mathbf{y}\|_2^2, \tag{17}$$

where $\mathbf{y}$ denotes the ground truth, $\mathbf{y}(t)$ is the prediction in the $t$-th FL round, and $\lambda$ denotes the minimum eigenvalue of the FL-NTK Gram matrix at initialization (denoted as $\mathbf{H}(0)$). However, the above convergence demands the transmission of $\Delta \mathbf{W}_c(t)$ from each client, and the communication complexity is high as the number of incremental weights is humongous since $M \to \infty$ in NTK-based ONNs. We propose Compressed Over-parameterized FL over MAC (COFL-MAC) to solve the above problem and analyze if our proposed framework exhibits exponential convergence while being inherently communication efficient.

## 2.2 Proposed Communication Framework of COFL-MAC

We propose a novel COFL-MAC framework for FL, which incorporates parameter communication over the resource-constrained up-link channel via sparsified gradients. We propose to achieve the sparsification of the gradients using CS. This helps overcome the communication bottleneck that entails the deployment of ONNs at the clients.

Compressive Sensing (CS) is a signal processing paradigm for obtaining a compressed representation of a sparse vector using over-complete dictionaries Eldar & Kutyniok (2012). The two crucial aspects for perfect reconstruction are sparsity (or compressibility) of the signal in the transform domain and incoherence, which are applied through the restricted isometry property (RIP) of the over-complete dictionary Candès & Wakin (2008). As established earlier, incremental weight updates are sparse; hence, CS can be employed at the clients to compress the updates. Using a common RIP-obeying Gaussian sensing matrix at each client, denoted as $\mathbf{A} \in \mathbb{R}^{n_r \times N}$, we obtain the compressed update given as $\Delta \widetilde{\mathbf{w}}_{c,m}(t) \in \mathbb{R}^{n_r}$ at each client as given in equation 5. The clients transmit these updates simultaneously over the MAC channel, which leads to the server receiving a superposition of the transmitted updates, $\sum_{c=1}^{N} \Delta \widetilde{\mathbf{w}}_{c,m}(t)$, which is then scaled down by the number of clients to obtain $\Delta \widetilde{\mathbf{u}}_m(t)$ via the FedAvg rule as follows:

$$\Delta \widetilde{\mathbf{u}}_m(t) = \frac{1}{C} \sum_{c=1}^{N} \Delta \widetilde{\mathbf{w}}_{c,m}(t) = \frac{1}{C} \left( \mathbf{A} \sum_{c=1}^{C} \Delta \mathbf{w}_{c,m}(t) \right). \tag{18}$$

We make the following assumption regarding the sparsity in the per-client updates.

**Assumption 3.** *There is sufficient sparsity in the per-client updates, $\Delta \mathbf{w}_{c,m}(t)$ so that $\sum_{c=1}^{C} \Delta \mathbf{w}_{c,m}(t)$ is sparse enough to allow for a common sensing matrix, $\mathbf{A}$, and therefore, $\Delta \mathbf{w}_{c,m}(t)$ can be compressed and recovered as per CS theory.*

The COFL-MAC process is depicted in Fig. 1, and its steps are listed in Algorithm 1. Note that at the beginning of the $t$-th FL round, each client $c$ receives the reconstructed global parameter update by the server from the previous round given by $\widehat{\mathbf{U}}(t)$. Also, at initialization, we have $\widehat{\mathbf{U}}(0) = \mathbf{U}(0)$. The server reconstructs $\Delta \widehat{\mathbf{U}}(t)$ from $\Delta \widetilde{\mathbf{U}}(t)$ using a sparse signal recovery (SSR) algorithm. We have used orthogonal matching pursuit (OMP) as the SSR algorithm.

Further, note that $\Delta \widetilde{\mathbf{w}}_{c,m}(t) = \mathbf{A} \Delta \mathbf{w}_{c,m}(t)$, where $\Delta \widetilde{\mathbf{w}}_{c,m}(t)$ is a column of the compressed update $\Delta \widetilde{\mathbf{W}}_c(t)$. Similarly, $\Delta \widehat{\mathbf{u}}_m(t)$ is a column of the reconstructed global weight update, $\Delta \widehat{\mathbf{U}}(t)$.

In the next section, we provide convergence guarantees for the proposed COFL-MAC framework using NTK analysis.

## 3 Convergence Analysis of the proposed COFL-MAC Scheme

Consider an NN of sufficient width in the over-parameterized lazy training regime. Hence, the NN parameter dynamics and prediction dynamics can be described via system continuous-time differential equations given by equation 14. Further, since we have used CS, the reconstruction error resulting from SSR at the server is expected to affect the convergence. In this work, we demonstrate that under the $\epsilon^2$-exact recovery condition (defined in the next section), the proposed COFL-MAC framework converges under the NTK analysis.

### 3.1 Sparse Signal Recovery in the COFL-MAC framework and Reconstruction Error Limits

We incorporate an SSR algorithm, namely, OMP, to recover the global parameter increment, $\Delta \widehat{\mathbf{u}}_m(t)$ from the aggregated compressed global update, i.e., $\Delta \widetilde{\mathbf{u}}_m(t)$ as given in Algorithm 1. We assume $\epsilon^2$-exact recovery condition framework ($\epsilon^2 C$ condition in Sturm (2012)), where $\epsilon$ is the maximum relative reconstruction error represented by

$$\frac{\|\Delta \widehat{\mathbf{u}}_m(t) - \Delta \mathbf{u}_m(t)\|_2}{\|\Delta \mathbf{u}_m(t)\|_2} < \epsilon. \tag{19}$$

We state the assumption regarding the probability of exact recovery as follows and then make the subsequent claim based on this assumption.

**Assumption 4.** *The probability of exact recovery is given as*

$$P \left[ \frac{\|\Delta \widehat{\mathbf{u}}_m(t) - \Delta \mathbf{u}_m(t)\|_2}{\|\Delta \mathbf{u}_m(t)\|_2} < \epsilon \right] > 1 - \delta', \tag{20}$$

*where $\epsilon \in (0, 1)$ and $\delta' \in (0, 0.5)$.*

**Claim 1.** *The probability of the $\ell_2$-norms of the reconstructed incremental weight vector, $\Delta \widehat{\mathbf{u}}_m(t)$ being greater than $1 - \epsilon$ times the $\ell_2$-norms of the incremental weight vector equals the probability of the $\ell_2$-norm of the reconstructed incremental weight vector, $(\Delta \widehat{\mathbf{u}}_m(t))$ being less than $1 + \epsilon$ times the $\ell_2$-norms of the incremental weight vector is given as follows:*

$$P\left[\|\Delta \widehat{\mathbf{u}}_m(t)\| > (1 - \epsilon)\|\Delta \mathbf{u}_m(t)\|\right] = P\left[\|\Delta \widehat{\mathbf{u}}_m(t)\| < (1 + \epsilon)\|\Delta \mathbf{u}_m(t)\|\right] > 1 - \delta, \tag{21}$$

*where $\delta = 0.5\delta'$, i.e., $\delta \in (0, 0.25)$.*

## 3.2 Movement of Local and Global Weights

We analyze the movement of global weights and local weights at the *c*-th client. In the upcoming lemma, we establish that the order of change in local model parameters across the local epochs is $\mathcal{O}(1/\sqrt{M})$ in COFL-MAC.

**Lemma 3.** *The movement of local weights in the $k + 1$-th local epoch at the c-th client in the t-th global communication round in COFL-MAC follows NTK dynamics, i.e.,*

$$\|\mathbf{w}_{c,m}^{k+1}(t) - \mathbf{w}_{c,m}^0(t)\|_2 \leq \frac{4\sqrt{D_c}\|\mathbf{y}_c - \mathbf{y}_c^0(t)\|_2}{\sqrt{M}\lambda_c(t)}, \tag{22}$$

*where $\lambda_c(t)$ is the minimum eigenvalue of $\mathbf{J}_c^0(t)$.*

The above lemma proves that movement of the local model parameters across local epochs at a client in the COFL-MAC framework obeys NTK dynamics and follows the lazy training regime. In the upcoming lemma, we analyze the movement of reconstructed global weights given by $\Delta \widehat{\mathbf{u}}_m(t)$ in the COFL-MAC framework. Based on equation 19, we have the following lemma.

**Lemma 4.** *The movement of the reconstructed global weights, $\Delta \widehat{\mathbf{u}}_m(t)$, in consecutive rounds of COFL-MAC is upper bounded as follows:*

$$\|\Delta \widehat{\mathbf{u}}_m(t)\|_2 \leq (1 + \epsilon)\frac{2\eta K(1 + 2\eta DK)\sqrt{D}}{C\sqrt{M}}\|\mathbf{y} - \mathbf{y}(t)\|_2.$$

The NTK theory is applicable to an NN when the change in the norm of model weights across iterations is $\mathcal{O}(1/\sqrt{M})$. We see that, albeit using compressed updates via the $\mathbf{A}$ matrix, the order of change in the norm of reconstructed global model weights across rounds in COFL-MAC continues to be $\mathcal{O}(1/\sqrt{M})$; hence, NTK theory based convergence analysis holds for the proposed approach. Also, note that the movement of reconstructed global weights varies inversely with an increase in the number of clients in the setup, which means as more clients join the setup, the movement of parameters across the communication rounds reduces. In the subsequent lemma, we present a bound on the movement of reconstructed global weights in COFL-MAC w.r.t. initialization.

**Lemma 5.** *The movement of the reconstructed global weights w.r.t. initialization in COFL-MAC is given by*

$$\|\widehat{\mathbf{u}}_m(t + 1) - \mathbf{u}_m(0)\|_2 \leq (1 + \epsilon)\left(\frac{8\sqrt{D}\|\mathbf{y} - \mathbf{y}(0)\|_2}{\sqrt{M}\lambda}\right),$$

*where $\mathbf{y}$ represents the ground truth and $\lambda = \lambda_{min}(\mathbf{G}(0))$, where $\mathbf{G}(0)$ is the COFL-MAC-NTK Gram matrix at initialization as defined in the sequel.*

Since we are using CS and, subsequently, SSR, the length (norm) of the reconstructed global weights could be either slightly higher or slightly lower in COFL-MAC than in uncompressed FL. This is also evident from Claim 1. Since we want an upper bound, we are looking at the maximum norm of movement of the reconstructed global weights from initialization. Therefore, recursively applying the global update step and

incorporating the reconstructed global weight movement in subsequent rounds as per Lemma 4, we arrive at the upper bound on the movement of reconstructed global weights in COFL-MAC and find that it is slightly higher than that of global weights in vanilla FL; i.e., the maximal movement of reconstructed weights from initialization in COFL-MAC is $(1 + \epsilon)$ times the maximal movement of global weights from initialization in vanilla FL. We extend this result further in the upcoming lemma.

**Lemma 6.** *Let the maximal movement of global weights from initialization in FL be within a radius $R$, and that in COFL-MAC be within a radius $\widehat{R}$; i.e., $R \triangleq \max_{m \in [M]} \left( \sum_{\tau=0}^{t-1} \|\Delta \mathbf{u}_m(\tau)\|_2 \right)$ and, $\widehat{R} \triangleq \max_{m \in [M]} \left( \sum_{\tau=0}^{t-1} \|\Delta \widehat{\mathbf{u}}_m(\tau)\|_2 \right)$. Then, the maximal movement of reconstructed global weights from initialization in COFL-MAC is within a radius $(1 + \epsilon) R$, i.e.,*

$$\widehat{R} \triangleq (1 + \epsilon)R. \tag{23}$$

The theory, until here, forms the groundwork/preliminaries for the main part of the convergence theory for COFL-MAC, which follows in the sequel. Based on the radii: $R$ and $\widehat{R}$, we define the events $\mathcal{B}_{i,m}$ and $\mathcal{E}_{i,m}$ in the sequel, which is foundational in quantifying the stability of neuron activation patterns, thereby, linking the parameter stability conditions to the convergence guarantees of the COFL-MAC framework.

### 3.3 Local/Global Update Deviation

In this subsection, We define probabilistic events to keep track of model weight deviation in the COFL-MAC in order to identify when a neuron's ReLU activation on input might flip within a small neighborhood of its initial weight. The event $\mathcal{B}_{i,m}$ for vanilla FL is defined as

$$\mathcal{B}_{i,m} = \left\{ \mathbf{z} : \|\mathbf{z} - \mathbf{u}_m(0)\|_2 \leq R, \mathbf{1}_{\mathbf{x}_i^T \mathbf{u}_m(0)} \neq \mathbf{1}_{\mathbf{x}_i^T \mathbf{z}} \right\}, \tag{24}$$

i.e., $\mathcal{B}_{i,m}$ considers the set of weights in the vanilla FL setting that stay close to initialization, yet see a flip in ReLU outputs. Analogous to $\mathcal{B}_{i,m}$, we define the event $\mathcal{E}_{i,m}$ for COFL-MAC as

$$\mathcal{E}_{i,m} = \left\{ \widehat{\mathbf{z}} : \|\widehat{\mathbf{z}} - \mathbf{u}_m(0)\|_2 \leq \widehat{R}, \mathbf{1}_{\mathbf{x}_i^T \mathbf{u}_m(0)} \neq \mathbf{1}_{\mathbf{x}_i^T \widehat{\mathbf{z}}} \right\}. \tag{25}$$

The event $\mathcal{E}_{i,m}$ characterizes neuron activations that flip due to parameter updates after CS and subsequent reconstruction, occurring within a slightly larger radius $\hat{R}$. The factor $(1 + \epsilon)$ in $\hat{R}$ accounts explicitly for the reconstruction uncertainty introduced by CS. The probability of these events is associated with the likelihood of model updates remaining in a regime where the NTK remains approximately constant. Bounding the probability of these activation-flipping events guarantees convergence robustness by controlling the magnitude of the complementary NTK Gram matrix (defined in the sequel), thus facilitating exponential convergence of the COFL-MAC method.

In the sequel, we provide a lower bound for the probability of the event $\mathcal{E}_{i,m}$, which will be used in 3.4. This probability is also used indirectly in 3.5 to count the number of neurons whose ReLU activation flips for a given input under COFL-MAC. Next, we present a lemma that establishes the relationship between events $\mathcal{E}_{i,m}$ and $\mathcal{B}_{i,m}$, quantifying how neuron activation flips in the COFL-MAC setting are related to those in the vanilla FL setting.

**Lemma 7.** *Assuming the activation pattern of the hidden layer neurons to be varying across communication rounds, i.e., $\mathbf{1}_{\mathbf{x}_i^T \mathbf{u}_m(0)} \neq \mathbf{1}_{\mathbf{x}_i^T \widehat{\mathbf{z}}}$, the event $\|\Delta \widehat{\mathbf{u}}_m(t)\|_2 \leq (1 + \epsilon)\|\Delta \mathbf{u}_m(t)\|_2$ implies the occurrence of the event $\mathcal{E}_{i,m} \mid \mathcal{B}_{i,m}$, where*

$$\mathcal{E}_{i,m} \mid \mathcal{B}_{i,m} = \{ \widehat{\mathbf{z}} : \|\widehat{\mathbf{z}} - \mathbf{u}_m(0)\|_2 \leq \widehat{R}, \mathbf{1}_{\mathbf{x}_i^T \mathbf{u}_m(0)} \neq \mathbf{1}_{\mathbf{x}_i^T \widehat{\mathbf{z}}} \mid$$
$$\mathbf{z} : \|\mathbf{z} - \mathbf{u}_m(0)\|_2 \leq R, \mathbf{1}_{\mathbf{x}_i^T \mathbf{u}_m(0)} \neq \mathbf{1}_{\mathbf{x}_i^T \mathbf{z}} \}, \tag{26}$$

*where, $\widehat{R} = (1 + \epsilon)R$.*

The event $\mathcal{E}_{i,m} \mid \mathcal{B}_{i,m}$ considers the set of weights in the COFL-MAC setting that stay close to initialization in a ball of radius $\widehat{R}$ and yet cause a flip in ReLU outputs given the weights stay within a smaller ball of radius $R$ in vanilla FL and cause activation flips. In order to calculate the probability of occurrence of event $\mathcal{E}_{i,m}$, we make the following proposition and subsequent claim:

**Proposition 1.** *Assuming the activation pattern of the hidden layer neurons to be varying across communication rounds, i.e., $\mathbf{1}_{\mathbf{x}_i^T \mathbf{u}_m(0)} \neq \mathbf{1}_{\mathbf{x}_i^T \widehat{\mathbf{z}}}$, the probability of event $\mathcal{E}_{i,m}$ conditioned on event $\mathcal{B}_{i,m}$ is at least $1 - \delta$, where $\delta \in (0, 0.25)$. In other words, we have, $P[\mathcal{E}_{i,m} \mid \mathcal{B}_{i,m}] \geq 1 - \delta$.*

According to this claim, if a neuron flips activation due to learning in vanilla FL, it has a high likelihood of flipping a neuron in COFL-MAC.

**Claim 2.** *Assuming the activation pattern of the hidden layer neurons to be varying across communication rounds, i.e., $\mathbf{1}_{\mathbf{x}_i^T \mathbf{u}_m(0)} \neq \mathbf{1}_{\mathbf{x}_i^T \widehat{\mathbf{z}}}$, the probability of the event $\mathcal{E}_{i,m}$ is at most $1 - \delta R$, i.e.,*

$$P[\mathcal{E}_{i,m}] < 1 - \delta R. \tag{27}$$

This claim tells us that the probability that ReLU activation flips for neurons on input $\mathbf{x}_i$ (due to the update in weights and reconstruction error) in COFL-MAC is small. The major implication of Claim 2 is that most neurons retain their initial activation pattern.

### 3.4 Movement of the COFL-MAC-NTK Gram Matrix

Now, we will provide an upper bound on the movement of the COFL-MAC Gram matrix from initialization to ensure the NN stays in a regime where training dynamics are predictable. This makes it possible to apply NTK-based convergence theory and prove linear rates for COFL-MAC.

The vanilla local FL-NTK Gram matrix, denoted as $\mathbf{H}_c^k(t) \in \mathbb{R}^{D \times D}$, allows us to analyze the relationship of the inner product of the gradient of the network function for two datapoints, without accessing the points directly. Since we have used ReLU activation functions, the elements of the FL-NTK Gram matrix at the $c$-th client, $H_{c,ij}^k(t)$ are defined as follows.

$$H_{c,ij}^k(t) = \frac{1}{M} \mathbf{x}_{c,i}^T \mathbf{x}_{c,j} \sum_{m=1}^{M} \mathbf{1}_{\mathbf{x}_{c,i}^T \mathbf{u}_m(t), \mathbf{x}_{c,j}^T \mathbf{w}_{c,m}^k(t)}. \tag{28}$$

This asymmetric Gram matrix, $\mathbf{H}_c^k(t)$ is close to the symmetric NTK Gram matrix at initialization, $\mathbf{H}(0)$, as shown in Huang et al. (2021a). Note that as $t \to 0$ and $M \to \infty$, $\mathbf{H}_c^k(t) \to \overset{\infty}{\mathbf{H}}$, which is the NTK Jacot et al. (2018). The elements of $\overset{\infty}{\mathbf{H}}$ are given by

$$\overset{\infty}{H}_{ij} := \mathbb{E}_{\mathbf{u}_m(0) \sim \mathcal{N}(0, \mathbf{I})} \left[ \mathbf{x}_{c,i}^\top \mathbf{x}_{c,j} \mathbf{1}_{\mathbf{x}_{c,i}^T \mathbf{u}_m(0), \mathbf{x}_{c,j}^T \mathbf{u}_m(0)} \right]. \tag{29}$$

Further, the COFL-MAC Gram Matrix at the $c$-th client is denoted by $\mathbf{G}_c^k(t) \in \mathbb{R}^{D \times D}$ and its elements are given as

$$G_{c,ij}^k(t) = \frac{1}{M} \mathbf{x}_i^T \mathbf{x}_j \sum_{m=1}^{M} \mathbf{1}_{\mathbf{x}_i^T \widehat{\mathbf{u}}_m(t), \mathbf{x}_j^T \mathbf{w}_{c,m}^k(t)}. \tag{30}$$

Note that $\mathbf{G}_c^k(t)$ involves the inner product of the derivative of the network function at a client w.r.t local weights with the derivative of global network function w.r.t. reconstructed global weights, whereas $\mathbf{H}_c^k(t)$ involves uncompressed global weights rather than reconstructed global weights. Further, the global COFL-MAC Gram matrix, $\mathbf{G}^k(t) \in \mathbb{R}^{D \times D}$ is obtained by combining the $\mathcal{S}_c$ columns of the Gram matrices $\mathbf{G}_c^k(t)$ for all $c \in [C]$, i.e.,

$$\mathbf{G}^k(t) = \frac{1}{C} \sum_{c \in C} \mathbf{G}_c^k(t). \tag{31}$$

In the upcoming lemma, we show that this Gram matrix, $\mathbf{G}^k(t)$, is close to the COFL-MAC-NTK Gram matrix at initialization, $\mathbf{G}(0)$.

**Lemma 8.** *For the set of reconstructed global weight vectors* $\widehat{\mathbf{u}}_1(t), \cdots, \widehat{\mathbf{u}}_M(t) \in \mathbb{R}^N$ *and the set of local weight vectors* $\mathbf{w}_{c,1}^k(t), \cdots, \mathbf{w}_{c,M}(t) \in \mathbb{R}^N$, *such that* $\|\widehat{\mathbf{u}}_m(t) - \mathbf{u}_m(0)\|_2 \leq (1+\epsilon)R$ *and* $\|\mathbf{w}_{c,m} - \mathbf{u}_m(0)\|_2 \leq (1+\epsilon)R$ *for any* $m \in [M]$, *with probability at least* $1 - D^2 \exp\left(\frac{-MR_1}{4R'/9R_1 + 2/9}\right)$, *the following holds*

$$\|\mathbf{G}^k(t) - \mathbf{G}(0)\|_F < \frac{2DR}{C},$$

*where* $R_1 = R - 2/3R'$.

This means that the COFL-MAC Gram matrix is fairly constant during training, ensuring the kernel regime of the ONN, where the NTK-based convergence guarantees hold. The proof of Lemma 8 quantifies the impact of activation changes on the COFL-MAC-NTK matrix using $\mathcal{E}_{i,m}$, $\mathcal{B}_{i,m}$, and $\mathcal{E}_{i,m} \mid \mathcal{B}_{i,m}$. Due to the bounded weight movement and controlled number of activation flips, the NTK kernel remains stable throughout training. This justifies using NTK-based convergence theory in the COFL-MAC framework.

### 3.5 The Complementary COFL-MAC-NTK Gram Matrix

The complementary COFL-MAC-NTK Gram matrix is a crucial entity in the convergence analysis. A small magnitude (Frobenius norm) of this matrix would imply fast and stable convergence. In the sequel, we motivate the significance of the complementary Gram matrix, followed by providing an upper bound on its Frobenius norm.

The hidden layer neurons in the NN can be split into two sets, one consisting of neurons whose activation pattern changes over time and another consisting of neurons whose activation pattern does not change over time. For each $i \in [D]$, we define the set $\mathcal{Q}_i \subset [M]$ of neurons whose activation pattern is certified to hold during the COFL-MAC process.

$$\mathcal{Q}_i := \left\{ m \in [M] : \widehat{\mathbf{z}} : \|\widehat{\mathbf{z}} - \mathbf{u}_m(0)\|_2 \leq (1+\epsilon)R, \quad \mathbf{1}_{\mathbf{x}_i^T \mathbf{u}_m(0)} = \mathbf{1}_{\mathbf{x}_i^T \widehat{\mathbf{z}}} \right\}. \tag{32}$$

Similarly, the set $\overline{\mathcal{Q}}_i$ is the complement of $\mathcal{Q}_i$ and can be defined as follows:

$$\overline{\mathcal{Q}}_i := \left\{ m \in [M] : \widehat{\mathbf{z}} : \|\widehat{\mathbf{z}} - \mathbf{u}_m(0)\|_2 \leq (1+\epsilon)R, \quad \mathbf{1}_{\mathbf{x}_i^T \mathbf{u}_m(0)} \neq \mathbf{1}_{\mathbf{x}_i^T \widehat{\mathbf{z}}} \right\}. \tag{33}$$

Based on the sets $\mathcal{Q}_i$ and $\overline{\mathcal{Q}}_i$, the COFL-MAC Gram matrix can be split into two components as follows

$$\mathbf{G}_c^k(t) = \mathbf{G}_c^k(t)^{\|} + \mathbf{G}_c^k(t)^{\perp}, \tag{34}$$

where $\mathbf{G}_c^k(t)^{\perp}$ represents the complementary COFL-MAC-NTK Gram matrix at the $c$-th client, whose elements are defined as follows

$$G_{c,ij}^k(t)^{\perp} = \frac{1}{M} \mathbf{x}_i^T \mathbf{x}_j \sum_{m \in \overline{\mathcal{Q}}_i} \mathbf{1}_{\mathbf{x}_i^T \widehat{\mathbf{u}}_m(t), \mathbf{x}_j^T \mathbf{w}_{c,m}^k(t)}. \tag{35}$$

Further, the global complementary COFL-MAC-NTK matrix, $\mathbf{G}^k(t)^{\perp} \in \mathbb{R}^{D \times D}$ is obtained by combining the $\mathcal{S}_c$ columns of the Gram matrices $\mathbf{G}_c^k(t)^{\perp}$ for all $c \in [C]$; i.e.,

$$\mathbf{G}^k(t)^{\perp} = \frac{1}{C} \sum_{c \in C} \mathbf{G}_c^k(t)^{\perp}. \tag{36}$$

The complementary Gram matrix, $G_k(t)^{\perp}$, captures contributions from neurons whose activation patterns change during training. By bounding its magnitude, we control deviations from the NTK regime, ensuring

that neuron activations remain largely stable, which is essential for preserving exponential convergence guarantees under the NTK framework. Specifically, maintaining a small norm of $G_k(t)^\perp$ ensures the stability of the NTK Gram matrix, thus validating the assumptions underpinning our convergence analysis.

As discussed earlier, the set $\overline{\mathcal{Q}}_i$ consists of weights that vary slowly; however, there is a change in their activation as compared to initialization. In order to quantify such weights, the random variable $\zeta_i$ is defined as

$$\zeta_i = \sum_{m=1}^{M} \mathbf{1}_{m \in \overline{\mathcal{Q}}_i}, \tag{37}$$

where $i \in [D]$. $\zeta_i$ becomes an important parameter in the convergence analysis. The set $\overline{\mathcal{Q}}_i$, captures the indices of neurons whose activation has flipped for input $\mathbf{x}_i$ compared to their initial state. This means that the scalar $\zeta_i$ is the cardinality of this complement set, i.e.,

$$\zeta_i = \left| \overline{\mathcal{Q}}_i \right|,$$

and thus, it serves as a stability indicator for the input $\mathbf{x}_i$ during training and quantifies the number of hidden neurons whose activation pattern with respect to $\mathbf{x}_i$ has changed compared to initialization.

In the sequel, we study the bound on the number of neurons whose activation patterns change with communication rounds.

**Claim 3.** *The random variable $\zeta_i$, defined as: $\zeta_i = \sum_{m=1}^{M} \mathbf{1}_{m \in \overline{Q}_i}$ for each $i \in [D]$, is bounded as $\zeta_i < 4MR$ with a probability of at least $1 - \exp\left(\frac{-MR_2}{\frac{R'}{8R_2}+1/6}\right)$, where $R_2 = R - R'/4$.*

A small $\zeta_i$ means that most neurons agree with their initial decision, and the NN stays in the NTK regime where the linearized approximation remains valid.

Next, we propose the following lemma regarding the complementary COFL-MAC-NTK Gram matrix.

**Lemma 9.** *Assuming that for each $i \in [D]$, $\zeta_i$ is bounded as $\zeta_i < 4MR$ with a probability of at least $1 - \exp\left(\frac{-MR_2}{\frac{R'}{8R_2}+1/6}\right)$, the norm of the complementary COFL-MAC-NTK Gram matrix is bounded as $\|\mathbf{G}^k(t)^\perp\|_F < \frac{4DR}{C}$, with the same probability over random initialization, where $R_2 = R - R'/4$.*

Claim 3 provides a high-probability upper bound on the number of neurons whose ReLU activation pattern changes for a data point $\mathbf{x}_i$, quantified by $\zeta_i$. Lemma 9 provides an upper bound on the Frobenius norm of the complementary COFL-MAC Gram matrix $\mathbf{G}_k^\perp(t)$, whose entries are defined using only those neurons in $\overline{\mathcal{Q}}_i$. This sum directly depends on $\zeta_i$, since $|\overline{\mathcal{Q}}_i| = \zeta_i$. A large $\zeta_i$ increases the number of non-zero terms in the sum, leading to a larger magnitude of each entry in $\mathbf{G}_k^\perp(t)$, and hence a larger Frobenius norm. Using the bounded number of activation-flipping neurons (Claim 3) and the bounded magnitude of each entry of $\mathbf{G}^k(t)^\perp$, we can treat the magnitude of the complementary Gram matrix as a small discrepancy, which, therefore has minimal impact on convergence. This helps in enabling the convergence of the proposed COFL-MAC setup.

In the upcoming section, we complete the convergence analysis and provide a proof sketch of our main theorem.

## 3.6 Exponential Convergence of COFL-MAC

We will shortly present our main convergence theorem. We list the three main aspects our convergence proof relies upon: (*i*) the confinement of local and global weights in a ball with a small radius around the initialization, (*ii*) the proximity of the COFL-MAC-NTK Gram matrix in any communication round to the COFL-MAC-NTK Gram matrix at initialization, and (*iii*) the magnitude of the complementary COFL-MAC-NTK Gram matrix.

Let us begin by understanding the role of the first aspect. When the weights do not move too far from initialization, the NTK matrix remains almost constant, and the NN is approximated as a linear model

in function space. This is an important aspect of NTK-based convergence analysis. Moving ahead to the second aspect, the proximity of the COFL-MAC-NTK matrix to its initial value implies a stable minimum eigenvalue of the COFL-MAC-NTK matrix, ensuring exponential convergence of the loss. Coming to the third aspect, explicitly bounding the complementary COFL-MAC-NTK Gram matrix validates that the NTK assumptions remain reasonable even when we employ CS and SSR. A small magnitude of this matrix ensures tight control of deviations from the idealized NTK dynamics, reinforcing exponential convergence guarantees.

### 3.7 Convergence Result

In this subsection, we present our main theorem.

**Theorem 1.** *Consider* $\lambda = \lambda_{\min}(H(0)) > 0$. *Let* $M = \Omega(\lambda^{-4}n^4 \log(D/\delta))$, *we iid initialize* $\mathbf{u}_m(0) \sim \mathcal{N}(0, I)$, $b_m$ *sampled from* $\{-1, +1\}$ *uniformly at random* $\forall m \in [M]$, *and we set the step size* $\eta = \mathcal{O}(\lambda/(\kappa K D^2))$. *With probability at least* $1 - \rho$ *over the random initialization for* $t = 0, 1, 2, \cdots$, *we have*

$$\|\mathbf{y}(t) - \mathbf{y}\|_2^2 \leq \left(1 - \gamma\left(\frac{\eta_g \eta \lambda K}{C}\right)\right)^t \|\mathbf{y}(0) - \mathbf{y}\|_2^2, \tag{38}$$

*where* $\gamma \in (0.44, 0.82)$.

*Proof.* We prove the theorem by mathematical induction. The base case is $t = 0$, and it is trivially true. Assuming that equation 38 is true for $1, \cdots, t$, we show equation 38 holds for $t + 1$. Decomposing the loss in the $(t+1)$-th global round, we obtain

$$\|\mathbf{y} - \mathbf{y}(t+1)\|_2^2 = \|\mathbf{y} - \mathbf{y}(t)\|_2^2 - 2(\mathbf{y} - \mathbf{y}(t))^T(\mathbf{y}(t+1) - \mathbf{y}(t)) + \|\mathbf{y}(t+1) - \mathbf{y}(t)\|_2^2. \tag{39}$$

Let us focus on the second term above, i.e., $-2(\mathbf{y}-\mathbf{y}(t))^T(\mathbf{y}(t+1)-\mathbf{y}(t))$, particularly the term $(\mathbf{y}(t+1)-\mathbf{y}(t))$.

Given the definition of $\mathcal{Q}_i$ and $\overline{\mathcal{Q}}_i$ in equation 32 and equation 33, respectively, we have:

$$y_i(t+1) - y_i(t) = v_{1,i} + v_{2,i}, \tag{40}$$

where

$$v_{1,i} = \frac{1}{\sqrt{M}} \sum_{m \in \mathcal{Q}_i} b_m \left(\phi\big(\mathbf{x}_i^T(\widehat{\mathbf{u}}_m(t) + \Delta\widehat{\mathbf{u}}_m(t))\big) - \phi(\mathbf{x}_i^T\widehat{\mathbf{u}}_m(t))\right), \tag{41}$$

and

$$v_{2,i} = \frac{1}{\sqrt{M}} \sum_{m \in \overline{\mathcal{Q}}_i} b_m \left(\phi\big(\mathbf{x}_i^T(\widehat{\mathbf{u}}_m(t) + \Delta\widehat{\mathbf{u}}_m(t))\big) - \phi(\mathbf{x}_i^T\widehat{\mathbf{u}}_m(t))\right). \tag{42}$$

From equation 41, we have

$$v_{1,i} = \frac{1}{\sqrt{M}} \sum_{m \in \mathcal{Q}_i} b_m \Delta\widehat{\mathbf{u}}_m(t) \mathbf{1}_{\mathbf{x}_i^T\widehat{\mathbf{u}}_m(t)}.$$

Rewriting equation 9, we have

$$\frac{\partial \mathcal{L}_c(\mathbf{W}_c^k(t))}{\partial \mathbf{w}_{c,m}^k(t)} = \frac{1}{\sqrt{M}} \sum_{i=1}^{D_c} (y_{c,i}^k(t) - y_{c,i}) b_m \mathbf{x}_{c,i} \mathbf{1}_{\mathbf{x}_{c,i}^T \mathbf{w}_{c,m}^k}.$$

Applying equation 40, we obtain

$$v_{1,i} \leq (1+\epsilon)\frac{\eta}{MC} \sum_{k \in [K]} \sum_{c \in [C]} \sum_{j \in S_c} (y_j - y_{c,j}^{(k)}(t))\mathbf{x}_i^T\mathbf{x}_j \left(\sum_{m=1}^{M} \mathbf{1}_{\mathbf{x}_i^T\widehat{\mathbf{u}}_m(t), \mathbf{x}_j^T\mathbf{w}_{c,m}^k(t)} - \sum_{m \in \overline{\mathcal{Q}}_i} \mathbf{1}_{\mathbf{x}_i^T\widehat{\mathbf{u}}_m(t), \mathbf{x}_j^T\mathbf{w}_{c,m}^k(t)}\right)$$

$$\leq (1+\epsilon)\frac{\eta}{C} \sum_{k \in [K]} \sum_{c \in [C]} \sum_{j \in S_c} (y_j - y_{c,j}^{(k)}(t)) \left(G_{c,ij}^k(t) - G_{c,ij}^k(t)^\perp\right). \tag{43}$$

We introduce the vectors $\mathbf{v}_1 \in \mathbb{R}^D$ and $\mathbf{v}_2 \in \mathbb{R}^D$ such that $v_{1,i}$ and $v_{2,i}$ are the $i$-th elements of $\mathbf{v}_1$ and $\mathbf{v}_2$, respectively. Hence, analogous to equation 40, we have $\mathbf{y}(t+1) - \mathbf{y}(t) = \mathbf{v}_1 + \mathbf{v}_2$. In order to simplify equation 39, we use equation 43 to obtain the following:

$$
\begin{aligned}
2(\mathbf{y} - \mathbf{y}(t))^T(\mathbf{y}(t+1) - \mathbf{y}(t)) &= -2(\mathbf{y} - \mathbf{y}(t))^T(\mathbf{v}_1 + \mathbf{v}_2) \\
&\leq \frac{-2(1+\epsilon)\eta}{C} \sum_{i,k,c,j} (y_i - y_i(t))(y_j - y_{c,j}^{(k)}(t))(G_{c,ij}^k(t) - G_{c,ij}^{k,\perp}(t)) \\
&\quad - 2\sum_{i\in[D]} (y_i - y_i(t))v_{2,i}.
\end{aligned}
\tag{44}
$$

Substituting the above expression in equation 39, we obtain

$$
\|\mathbf{y} - \mathbf{y}(t+1)\|_2^2 \leq \|\mathbf{y} - \mathbf{y}(t)\|_2^2 + C_1 + C_2 + C_3 + C_4,
\tag{45}
$$

where

$$
C_1 = -\frac{2(1+\epsilon)\eta}{C} \sum_{i\in[D]} \sum_{k\in[K]} \sum_{c\in[C]} \sum_{j\in S_c} (y_i - y_i(t))(y_j - y_{c,j}^{(k)}(t))G_{c,ij}^k(t),
\tag{46}
$$

$$
C_2 = \frac{2(1+\epsilon)\eta}{C} \sum_{i\in[D]} \sum_{k\in[K]} \sum_{c\in[C]} \sum_{j\in S_c} (y_i - y_i(t))(y_j - y_{c,j}^{(k)}(t)_j)G_{c,ij}^k(t)^\perp,
\tag{47}
$$

$$
C_3 = -2\sum_{i\in[D]} (y_i - y_i(t))v_{2,i},
\tag{48}
$$

$$
C_4 = \|\mathbf{y}(t+1) - \mathbf{y}(t)\|_2^2.
\tag{49}
$$

Applying Lemma 10, Lemma 11, Lemma 12 and Lemma 13, we have

$$
\begin{aligned}
\|\mathbf{y} - \mathbf{y}(t+1)\|_2^2 &\leq \|\mathbf{y} - \mathbf{y}(t)\|_2^2 - \frac{\eta\lambda K(1+\epsilon)}{C}\|\mathbf{y} - \mathbf{y}(t)\|_2^2 \\
&\quad + 40(1+\epsilon)\frac{\eta KDR}{C}\|\mathbf{y} - \mathbf{y}(t)\|_2^2 \\
&\quad + 40(1+\epsilon)\frac{\eta KDR}{C}\|\mathbf{y} - \mathbf{y}(t)\|_2^2 \\
&\quad + (1+\epsilon)^2\frac{\eta^2 D^2 K^2}{C^2}\|\mathbf{y} - \mathbf{y}(t)\|_2^2.
\end{aligned}
$$

As mentioned earlier, $\epsilon \in (0,1)$. Substituting $\epsilon = 1$, and taking $\eta \leq \frac{\lambda}{1000\kappa D^2 K}$, $\eta\eta_g \leq \frac{\lambda}{1000\kappa D^2 K}$, and $R \leq \lambda/(1000D)$, we arrive at:

$$
\begin{aligned}
\|\mathbf{y} - \mathbf{y}(t+1)\|_2^2 &\leq \|\mathbf{y} - \mathbf{y}(t)\|_2^2 - (2 - 4/10)\frac{\eta\lambda K}{C}\|\mathbf{y} - \mathbf{y}(t)\|_2^2 + 160\frac{\eta KnR}{N}\|\mathbf{y} - \mathbf{y}(t)\|_2^2 \\
&\leq \|\mathbf{y} - \mathbf{y}(t)\|_2^2 - 0.44\Big(\frac{\eta\lambda K}{C}\|\mathbf{y} - \mathbf{y}(t)\|_2^2\Big).
\end{aligned}
\tag{50}
$$

Therefore, we have:

$$
\|\mathbf{y} - \mathbf{y}(t+1)\|_2^2 \leq \left(1 - 0.44\left(\frac{\eta\lambda K}{C}\|\mathbf{y} - \mathbf{y}(t)\|_2^2\right)\right)\|\mathbf{y} - \mathbf{y}(t)\|_2^2.
\tag{51}
$$

Next, substituting $\epsilon = 0$ in equation 50, we obtain

$$
\|\mathbf{y} - \mathbf{y}(t+1)\|_2^2 \leq \left(1 - 0.82\left(\frac{\eta\lambda K}{C}\|\mathbf{y} - \mathbf{y}(t)\|_2^2\right)\right)\|\mathbf{y} - \mathbf{y}(t)\|_2^2.
\tag{52}
$$

Alternatively, we can write

$$\|\mathbf{y}(t) - \mathbf{y}\|_2^2 \leq \left(1 - \gamma\left(\frac{\eta\lambda K}{C}\right)\right)^t \cdot \|\mathbf{y}(0) - \mathbf{y}\|_2^2, \tag{53}$$

where $\gamma = 0.82$ for uncompressed vanilla FL and $\gamma \in (0.44, 0.82)$ for COFL-MAC. Therefore, our convergence result shows that the convergence of COFL-MAC is similar to that of over-parameterized vanilla FL Huang et al. (2021a); albeit being slightly slower in COFL-MAC. We show via experiments that the parameter communication saved per communication round in COFL-MAC outweighs the slight convergence slowdown.

$\square$

**Lemma 10.** *We have the following with a probability at least* $1 - D^2 \exp\left(\frac{-MR_1}{4R'/9R_1 + 2/9}\right)$*, where* $R_1 = R - 2/3R'$ *over random initialization*

$$C_1 \leq (1 + \epsilon)\frac{2\eta_g\eta}{C}\|\mathbf{y} - \mathbf{y}(t)\|_2^2(-K\lambda + 4DRK(1 + 2\eta KD) + 2\eta\kappa\lambda K^2 D).$$

**Lemma 11.** *The following holds with probability at least* $1 - D\exp(-MR_2)$ *over random initialization*

$$C_2 \leq (1 + \epsilon)\frac{16\eta K}{C}(1 + 2\eta DK)DR\|\mathbf{y} - \mathbf{y}(t)\|_2^2.$$

**Lemma 12.** *With probability at least* $1 - D\exp(-MR_2)$ *over random initialization the following holds*

$$C_3 \leq \frac{(1 + \epsilon)16\eta_g\eta K}{C}(1 + 2\eta DK)DR\|\mathbf{y} - \mathbf{y}(t)\|_2^2.$$

**Lemma 13.** *We have*

$$C_4 \leq \frac{(1 + \epsilon)^2 4\eta^2\eta_g^2 D^2 K^2(1 + 2\eta DK)^2}{C^2}\|\mathbf{y} - \mathbf{y}(t)\|_2^2.$$

## 4 Discussions

### 4.1 Impact of MAC

Using MAC for uplink communication helps achieve a higher throughput and better communication efficiency. This impact becomes more conspicuous as more and more clients get added to the FL setup. For example, if we have 10 clients in the setup, in the absence of MAC uplink, we need 10 separate orthogonal channels; however, in a real-world FL setup, we could have millions of clients/ devices, making the use of MAC indispensable for making an FL setup communication efficient.

### 4.2 Impact of Compressive Sensing

Despite the use of MAC, modern-day ONNs can still choke the communication uplink when deployed in an FL setup, leading to bandwidth and latency concerns. The lazy training regime in ONNs renders sufficient sparsity in the transmitted client parameter gradients, making CS a viable solution. Moreover, the CS operator, i.e., the sensing matrix, $\mathbf{A}$, is a linear operator, which is a key requirement if we want to benefit from a MAC uplink. Therefore, CS makes deploying ONNs worthwhile in an FL-MAC setup as a scheme, which we call COFL-MAC. Moreover, as seen from Theorem 1, COFL-MAC exhibits exponential convergence similar to an over-parameterized uncompressed vanilla FL setup. However, the trade-off here will be a small reduction in speed with which we achieve this exponential convergence and the extra computation and memory requirement to perform CS and signal recovery.

### 4.3 Related Works

**ONNs and Lazy training**: It is natural to question the generalization capabilities of ONNs since the well-established classical ML theory emphasizes the curse of over-fitting in highly complex ML models. Surprisingly, ONNs generalize well, and this contradictory behavior arises because of the double descent bias-variance trade-off rule for NNs, unlike the single U-shaped curve for classical ML models Belkin et al. (2019). The second descent is for the over-parameterized (or interpolation) regime, where the loss starts to decline again upon adding more parameters to the NN model. Such ONNs trained with gradient-based optimizers are capable of converging exponentially fast to infinitesimal training loss, with sluggish variation in parameters across (S)GD iterations - a phenomenon known as lazy training. Some of the key works done related to ONNs, lazy training, and NTK theory are Du et al. (2018); Liu et al. (2022); Chizat et al. (2019); Jacot et al. (2018); Arora et al. (2019); Zou & Gu (2019a); Yang & Zhou (2024); Zou & Gu (2019b).

**Gradient sparsification and FL**: In the gradient sparsification approach, only a small subset of important elements of the gradient is communicated instead of the full gradient Li et al. (2020a). Gradient sparsification using linear encoders (e.g., CS) is particularly suited for analog transmission methods in FL-MAC. In contrast, digital transmission uses sparsification and quantization in tandem Oh et al. (2023; 2022). Analog transmission-based gradient sparsification has been studied in Amiri & Gündüz (2020), and the sparsification approach is based on compressive sensing (CS).

**CS and SSR**: There are a number of approaches to CS signal reconstruction in the literature and most of them belong to one of three main approaches Orović et al. (2016): convex optimization using algorithms like basis pursuit Candes & Tao (2005), Dantzig selector Boyd & Vandenberghe (2004), and gradient-based algorithms; greedy algorithms like matching pursuit Mallat & Zhang (1993) and OMP Tropp & Gilbert (2007), and hybrid methods such as compressive sampling matching pursuit (CoSamp) Needell & Tropp (2009) and stage-wise OMP Donoho et al. (2012). Convex programming methods have reported the best reconstruction accuracy; however, at the cost of high computational complexity Liu et al. (2012). The greedy algorithms bring about low computation complexity, while the hybrid methods try to provide a compromise between reconstruction accuracy and computational complexity Liu et al. (2012).

## 5 Experimental Evaluations and Discussions

In this section, we experimentally demonstrate the performance of the proposed COFL-MAC algorithm. The goal of the experiments is to (*a*) illustrate the relative accuracy performance after compression, (*b*) investigate if learning in the over-parametrized regime followed by compressed sensing is a better strategy than using under-parameterized NNs, and (*c*) investigate if COFL-MAC leads to improved communication efficiency. In the sequel, we describe the datasets, experimental settings, metrics, and numerical results.

### 5.1 Datasets, Experimental Settings, and NN Architecture

We have performed the experiments on two datasets, namely CIFAR-10 and Fashion MNIST (or FMNIST). The CIFAR-10 dataset consists of 60000, $32 \times 32$ colored images belonging to 10 classes, with 6000 images per class. There are 50000 training images and 10000 test images cif. The FMNIST dataset is a grayscale image dataset, which consists of 60000 training samples and 10000 test samples fmn. We perform non-IID data partitioning among the clients using the Dirichlet partitioning technique Lin (2016) with different values of the concentration parameter $\alpha$ to modulate the heterogeneity among the client data partitions. We have used $\alpha \in \{0.001, 0.01, 0.1, 0.5, 1\}$ for different experiments. Note that the lower the $\alpha$ value, the higher the heterogeneity across data among clients. The NN deployed at the clients in the COFL-MAC setup

for experiments involving the FMNIST dataset is an NN with around $79,510$ model weights, which correspond to 2.43 Megabits at each client before compression, assuming a 32-bit word implementation. The corresponding NN model for CIFAR10-based experiments consists of around $92,500$ weights, which correspond to 2.82 Megabits at each client before compression.

## 5.2 Benchmark Strategies and Performance Metrics

The performance of COFL-MAC is compared against the following benchmarking strategies:

1. Top-$k$ with correction: In this gradient sparsification strategy, the $k$ largest gradients are transmitted from the clients to the server in each communication round. The correction here refers to the process where each client adds its local error vector from the previous round into the gradient and then truncates this sum to its top $k$ components, sorted in decreasing order of magnitude, which is subsequently transmitted to the server. This was proposed in Alistarh et al. (2018).

2. SignSGD: This is a gradient quantization approach in which each client transmits only the sign (1 bit) of each coordinate of the gradient vector at the end of local training Bernstein et al. (2018).

3. MQAT: Multi-bit Quantization-Aware Training in FL is an extension of SignSGD, where the gradient is quantized and represented using multiple bits. MQAT incorporates the knowledge of quantization during the training phase, allowing the model to learn representations that maintain accuracy even when subjected to different quantization levels Gupta et al. (2022).

In order to compare the performance of COFL-MAC against the above-mentioned compression benchmarks, we employ NN, which has similar bit encoding at the clients before compression.

We employ three performance metrics, namely, test accuracy, communication cost, and the relative global reconstruction error. The communication cost per round in FL is calculated as the total number of bytes transmitted from the clients, which is averaged over the number of communication rounds. We define the relative global reconstruction error as $\frac{\|\mathbf{u}(t) - \widehat{\mathbf{u}}(t)\|_2}{\|\mathbf{u}(t)\|_2}$, where $\widehat{\mathbf{u}}(t) = SSR(\mathbf{A}\mathbf{u}(t))$, where SSR is as explained in 3.1.

## 5.3 Experimental Results

In this section, we illustrate the following: ($a$) the comparative performance of the proposed COFL-MAC framework with the benchmark strategies listed above, ($b$) ablation studies exploring: ($i$) performance improvement over under-parameterized FedAvg, ($ii$) the impact of varying hyper-parameters such as the number of clients, data heterogeneity and number of parameters transmitted; and ($c$) relative global reconstruction errors.

**Comparison with the Benchmark Strategies:** We illustrate the performance of the proposed COFL-MAC approach as compared to the SignSGD, MQAT, and Top-$k$ with correction gradient compression algorithms, which were described earlier. In particular, our goal is to investigate the communication cost of the proposed technique as compared to the benchmarks while maintaining the same value of test accuracy.

In the plots shown in Fig. 2, we compare the uplink communication cost of COFL-MAC with the benchmarks expressed as per round cost in kilobytes (KB), maintaining the same accuracy for the FMNIST and the CIFAR-10 datasets, respectively, for varying data heterogeneities. In SignSGD, the number of bits to be transmitted over the uplink is fixed (the sign bit) for any heterogeneity level. Therefore, for a fixed dimension of parameter update, the communication cost will be the same across the heterogeneity levels, though accuracy will vary. In MQAT, the communication cost increases with heterogeneity. This is because higher-bit quantization is essential to preserve gradient information in high-heterogeneity scenarios, ensuring that updates remain closer to the full-precision model.

We observe that Top-$k$ sparsification outperforms the other benchmarks and COFL-MAC for lower heterogeneity, i.e., $\alpha = 0.1$. However, as the heterogeneity increases, we observe a large reduction in the communication cost of COFL-MAC as compared to the baselines. This is mainly because the sensing matrix in COFL-MAC can be thought of as a dictionary, and dictionary-based learning is helpful in overcoming data heterogeneity in distributed settings like FL and COFL-MAC Huang et al. (2022) Gkillas et al. (2022). In COFL-MAC, a shared dictionary (the sensing matrix), $\mathbf{A}$, is used to represent local model updates in a structured manner. Instead of transmitting raw model updates, each client expresses its local updates as a sparse combination of dictionary elements. The dictionary in COFL-MAC helps mitigate heterogeneity by

enforcing a shared structure, aligning updates, and reducing model drift. Further note that, owing to the non-linear compression in the benchmark strategies, none of them leverage the MAC uplink the way COFL-MAC does. Simultaneous parameter transmission from all the clients reduces the uplink communication cost, thereby making the communication efficiency of our method very high compared to the benchmarks.

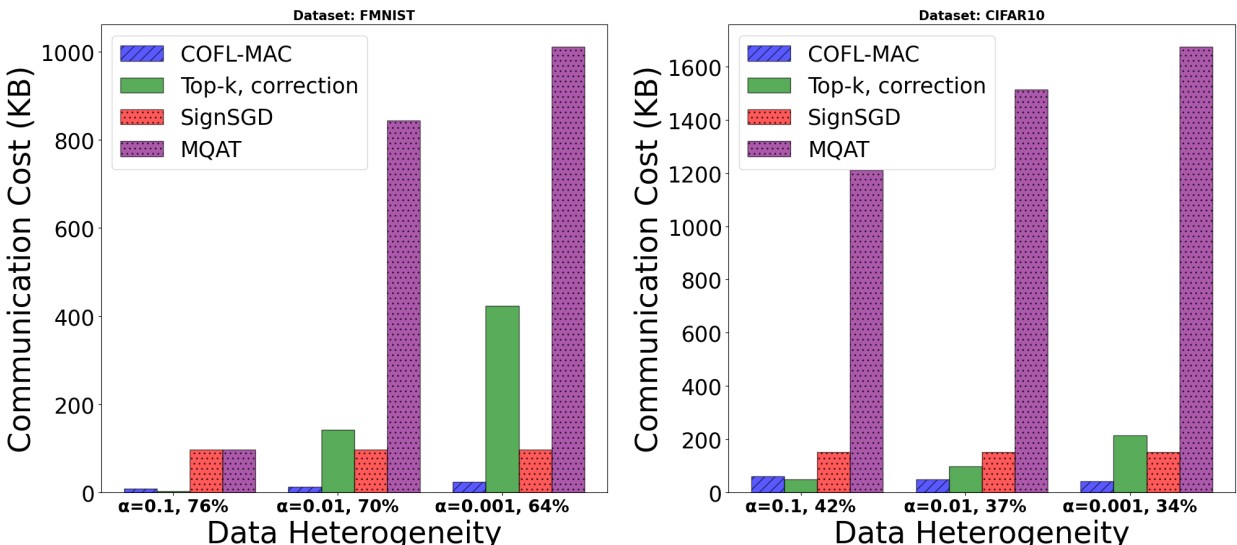

Figure 2: Comparing average communication cost of COFL-MAC with different compression baselines for FMNIST (left) and CIFAR-10 (right) datasets.

**Ablation Studies:** We perform a number of ablation studies. First, we explore the performance improvement of COFL-MAC over under-parameterized FedAvg. We also study the impact of varying the number of clients on performance, the impact of varying data heterogeneity, and the impact of varying the number of parameters transmitted over the uplink.

**Over-parameterize and Compress Vs Under-parameterized:** We measure the performance of COFL-MAC over under-parameterized FedAvg involving transmission of the same number of parameters over the uplink using *accuracy gain* as the metric as shown in Fig. 3, which is defined as the difference between test accuracy of COFL-MAC and under-parameterized FedAvg when the transmitted uplink parameters $\Delta \widetilde{\mathbf{w}}_{c,m}(t) \in \mathbb{R}^{n_r}$ are equal in number, i.e., $n_r$. In order to make a fair comparison for the FMNIST dataset, we choose $M = 5, 7, 10$ hidden neurons for the under-parameterized FL setup, and we consider the corresponding number of rows, $n_r = 3985, 5575, 7960$ in $\mathbf{A}$ for the COFL-MAC setup. This makes the number of parameters (and the number of bits) transmitted by the clients to the server nearly the same for both cases. Similarly, for the CIFAR-10 dataset, we chose $M = 3, 4, 5$ hidden neurons for the under-parameterized FL setup, and we consider the corresponding number of rows, $n_r = 9250, 12333, 15416$ in $\mathbf{A}$ for the COFL-MAC setup.

Essentially, we investigate if learning in the over-parametrized regime followed by CS is a better strategy than using under-parameterized NNs. It is evident from Fig. 3 that the former is the better strategy since we achieve high accuracy gains. Further, it can be observed that the improvement in accuracy (accuracy gain) is more prominent for higher data heterogeneity, i.e., $\alpha$. This happens because the advantage of heterogeneity mitigation in COFL-MAC due to the dictionary used for CS, i.e., $\mathbf{A}$, comes into the picture. It is known from the literature that dictionary-based learning helps overcome data heterogeneity in distributed settings like FL and COFL-MAC Huang et al. (2022) Gkillas et al. (2022).

**Varying the number of Clients ($C$):**

In Fig. 4, we depict the performance of COFL-MAC with varying numbers of clients in the FL setup. In general, adding more clients to an FL setup introduces wider data coverage in the training process, leading to better generalization and reduction variance in the averaged model parameters, leading to a more stable

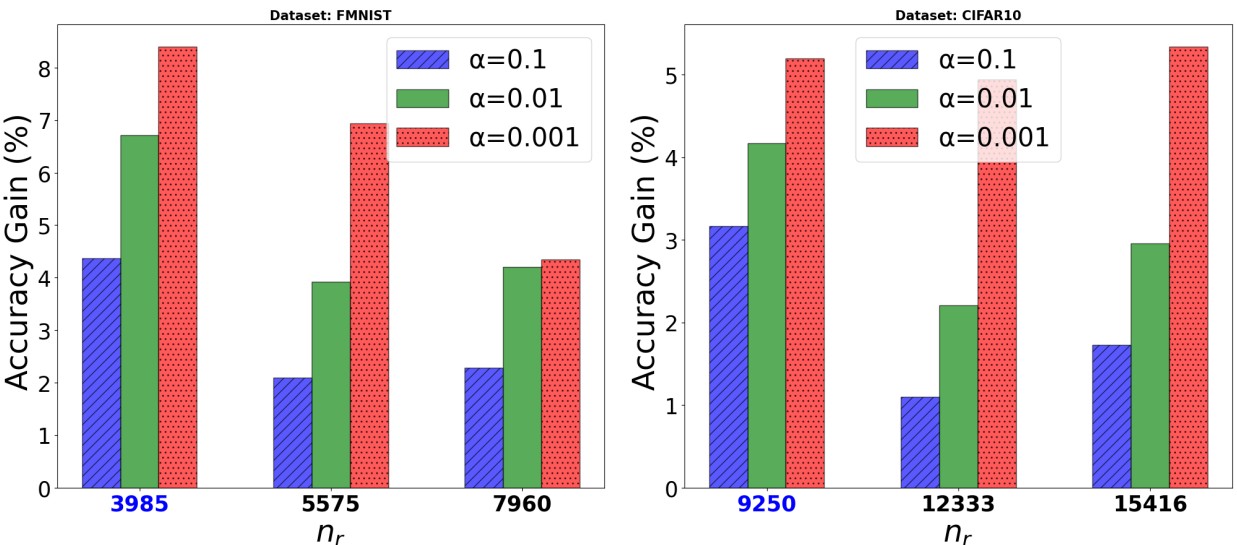

Figure 3: Accuracy gain defined as (COFL-MAC accuracy - FedAvg accuracy) vs number of parameters transmitted for FMNIST (left) and CIFAR-10 (right) datasets.

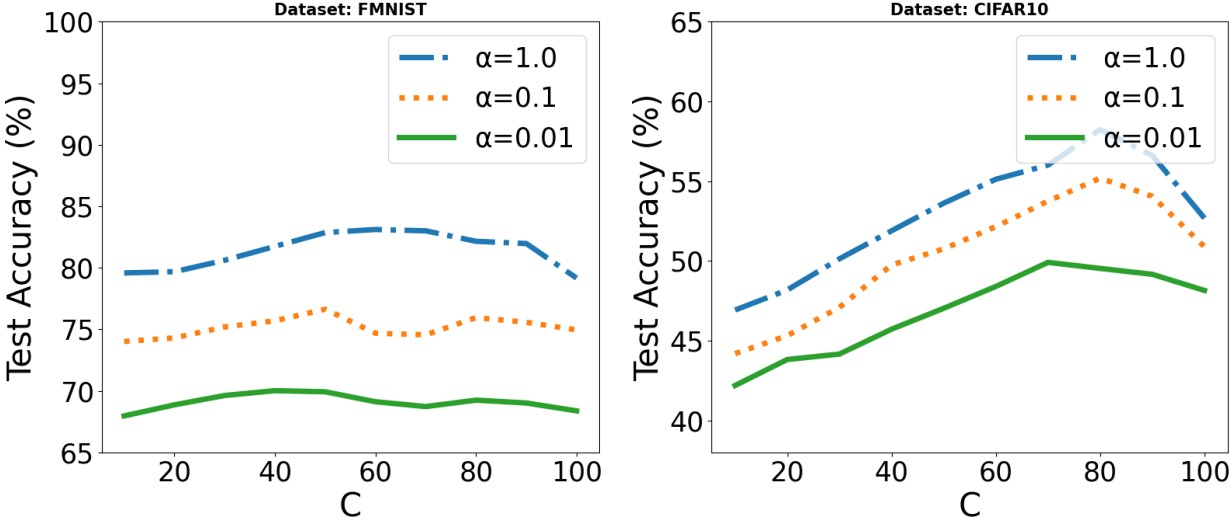

Figure 4: Effect of adding more clients to the COFL-MAC setup on the accuracy performance for FMNIST (left) and CIFAR10 (right).

and accurate global model Huang et al. (2021a); however, the convergence becomes slower. As it can be observed, initially, adding clients to the setup increases the performance in accordance with established FL theory. This can also be examined from the result of Theorem 1; however, if we go on adding more and more clients to our setup, after a certain point, the situation seems to be going antithetical, and we begin to see a decline in accuracy. The reason for this decrease in accuracy in exchange for higher communication efficiency is the reduction in sparsity in $\sum_{c=1}^{C} \Delta\mathbf{w}_{c,m}(t)$ on the addition of more clients impacting the SSR at the server, which starts to overshadow the increase in performance as per Theorem 1.

**Varying the number of parameters transmitted after compression ($n_r$):** In Fig. 5, we depict the performance of COFL-MAC with varying numbers of client parameters transmitted to the server after compression via the sensing matrix $\mathbf{A}$. In general, including more parameters from clients over the uplink leads to an increase in test accuracy in a COFL-MAC setup. For the CIFAR-10 dataset, we used a sparsity

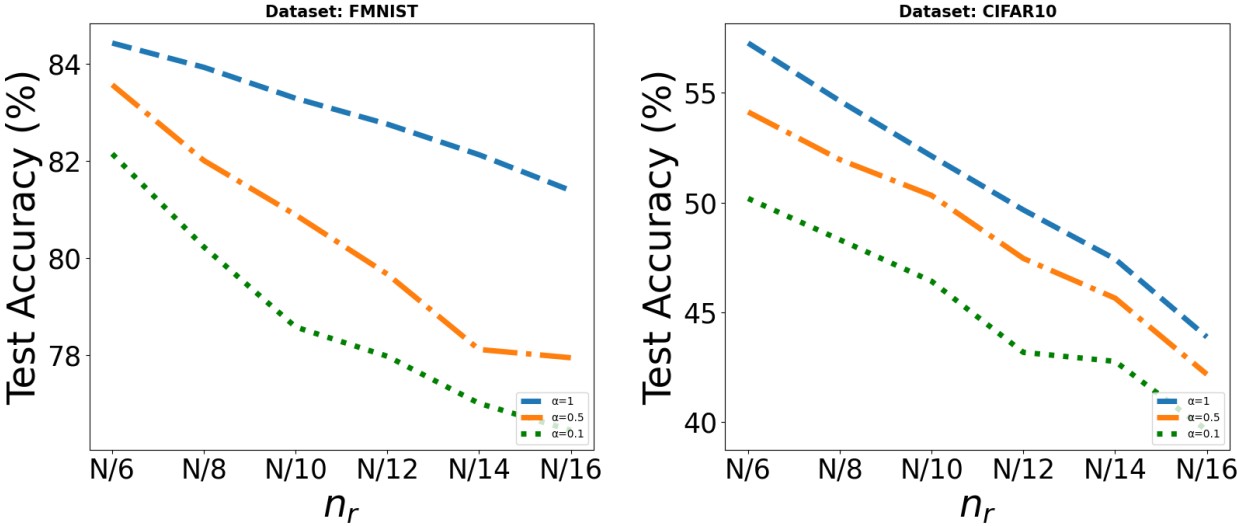

Figure 5: Impact of varying the number of client parameters transmitted to the server for different data heterogeneity extent among the clients on the test accuracy for FMNIST (left) and CIFAR10 (right) datasets.

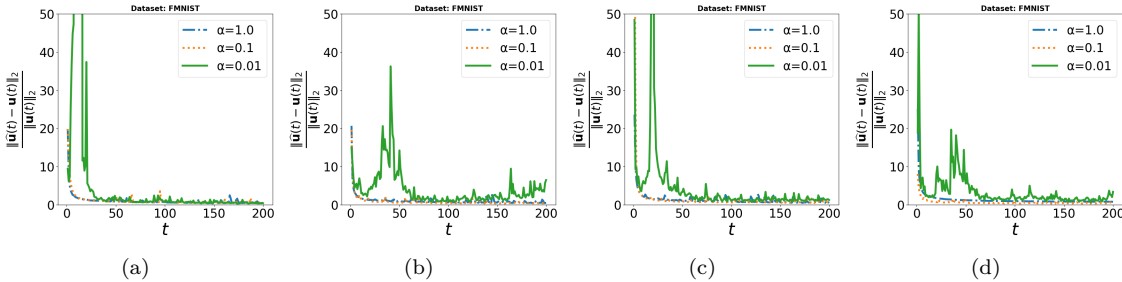

Figure 6: Global relative reconstruction error for FMNIST Dataset for different sparsity levels (a) 200 (b) 400 (c) 600 (d) 800

level of $S = 2000$, whereas for the FMNIST dataset, we used $S = 1200$. Note that a sparsity level of $S = 2000$ means that we recover 2000 non-zero elements in the SSR, and the rest will be zero.

**Relative Reconstruction Error in Sparse Signal Recovery:**

As shown in Fig. 6 and Fig. 7, we vary the heterogeneity levels to investigate their effect on the relative global reconstruction error as training proceeds. We observe that the relative reconstruction errors are lower for lower heterogeneity. Also, they fall off more rapidly when heterogeneity is low among the clients. Further, keeping the sparsity level the same, an increase in heterogeneity causes more oscillations in the reconstruction errors in addition to an increase in the errors.

At a broad level, we can say that as training proceeds, reconstruction errors fall off rapidly. This is an essential factor in the convergence of the COFL-MAC algorithm.

## 6 Conclusions and Future Work

In this work, we proposed a communication-efficient over-parameterized FL scheme based on gradient sparsification via CS, i.e., the COFL-MAC framework. We showed how CS helps deploy ONNs in an FL-MAC setup. By means of experiments, we showed that in our setup, by transmitting the same number of parameters as an under-parameterized NN, we achieve far higher accuracies. Further, using a dictionary, **A** (for

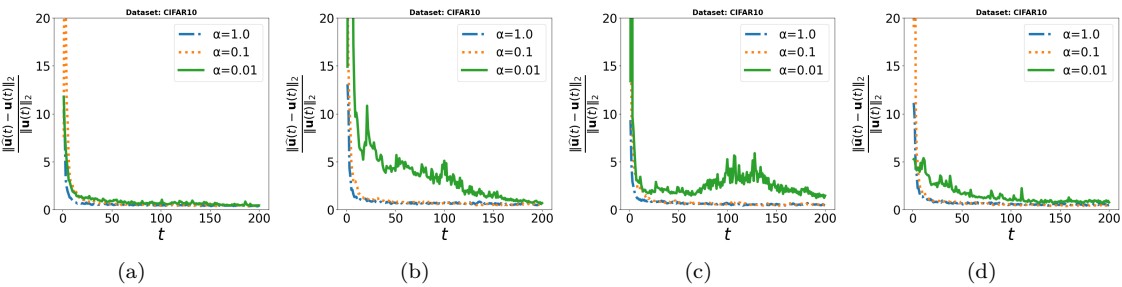

Figure 7: Global relative reconstruction error for CIFAR10 Dataset different sparsity levels (a) 200 (b) 400 (c) 600 (d) 800

compression), in the setup helped mitigate heterogeneity, making COFL-MAC a good choice for practical FL scenarios. Moreover, by employing NTK theory and analysis, we showed that the COFL-MAC setup exhibits exponential convergence. We demonstrated empirically that the parameter communication saved per communication round in COFL-MAC outweighs the slight convergence slowdown in COFL-MAC. We expect that the COFL-MAC algorithm will establish itself as a benchmark for examining the performance of future compression techniques in an FL or FL-MAC setup. As a future work, exploring the scalability of COFL-MAC in its capacity to include more and more clients will make it more powerful and enable the expansion of the COFL-MAC setup, which is important in practical applications like IoT.

## APPENDIX

**Roadmap:** In Appendix A, we list the results of two concentration inequalities. In Appendix B we prove the lemmas and claims used in convergence analysis (convergence theorem).

## A    Concentration Inequalities Followed

**Lemma 14** (Anti-concentration inequality of Gaussian Huang et al. (2021a)). *Let $X \sim \mathcal{N}(0, \sigma^2)$, then for any $0 < t \leq \sigma$*

$$P\left[|X| \leq t\right] \in \left(\frac{2t}{3\sigma}, \sqrt{\frac{2}{\pi}} \cdot \frac{t}{\sigma}\right).$$

**Lemma 15** (Bernstein inequality Bernstein (1924)). *Let $X_1, \ldots, X_n$ be i.i.d. random variables with mean $\mathbb{E}[X_i]$. Suppose that $|X_i - \mathbb{E}[X_i]| \leq \gamma$ almost surely, for all $i \in [n]$. Then, for all positive $t$,*

$$P\left[\frac{1}{n}\sum_{i=1}^{n}(X_i - \mathbb{E}[X_i]) \geq t\right] \leq \exp\left(-\frac{nt^2/2}{Var(X_i) + \gamma t/3}\right).$$

## B    Proofs of the Lemmas/Claims Stated Earlier

**Proof of Lemma 1**

*Proof.* As mentioned earlier, typically, a squared loss function is assumed while incorporating the NTK kernel, and hence, at each of the clients, we have the local loss function given by

$$\mathcal{L}_c(\mathbf{W}_c) := \frac{1}{2}\sum_{i=1}^{D_c}(y_{c,i}(t) - y_{c,i})^2. \tag{B1}$$

Hence, the gradient descent (SGD) update is given by

$$\mathbf{W}_c^{k+1}(t) \leftarrow \mathbf{W}_c^k(t) - \eta \frac{\partial \mathcal{L}_c(\mathbf{W}_c^k(t))}{\partial \mathbf{W}_c^k(t)}, \tag{B2}$$

where each column of $\dfrac{\partial \mathcal{L}_c(\mathbf{W}_c^k(t))}{\partial \mathbf{W}_c^k(t)}$ is obtained as :

$$\frac{\partial \mathcal{L}_c(\mathbf{W}_c^k(t))}{\partial \mathbf{w}_{c,m}^k(t)} = \frac{1}{\sqrt{M}} \sum_{i=1}^{D_c} (y_{c,i}^k(t) - y_{c,i}) b_m \mathbf{x}_{c,i} \mathbf{1}_{\mathbf{x}_{c,i}^T \mathbf{w}_{c,m}^k}. \tag{B3}$$

Taking $\ell_2$ norm, we obtain

$$\left\| \frac{\partial \mathcal{L}_c(\mathbf{W}_c^k(t))}{\partial \mathbf{w}_{c,m}^k(t)} \right\|_2 = \left\| \frac{1}{\sqrt{M}} \sum_{i=1}^{D_c} (y_{c,i}^k(t) - y_{c,i}) b_m \mathbf{x}_{c,i} \mathbf{1}_{\mathbf{x}_{c,i}^T \mathbf{w}_{c,m}^k} \right\|_2$$

Applying Cauchy Schwarz inequality followed by triangle inequality, we obtain

$$\left\| \frac{\partial \mathcal{L}_c(\mathbf{W}_c^k(t))}{\partial \mathbf{w}_{c,m}^k(t)} \right\|_2 \leq \frac{1}{\sqrt{M}} \sum_{i=1}^{D_c} \left\| (y_{c,i}^k(t) - y_{c,i}) b_m \mathbf{x}_{c,i} \mathbf{1}_{\mathbf{x}_{c,i}^T \mathbf{w}_{c,m}^k} \right\|_2.$$

We apply Cauchy Schwarz inequality again to arrive at the following:

$$\begin{aligned}
\left\| \frac{\partial \mathcal{L}_c(\mathbf{W}_c^k(t))}{\partial \mathbf{w}_{c,m}^k(t)} \right\|_2 &\leq \frac{1}{\sqrt{M}} \sum_{i=1}^{D_c} \left\| (y_{c,i}^k(t) - y_{c,i}) \right\|_2 \left\| b_m \right\|_2 \left\| \mathbf{x}_{c,i} \right\|_2 \left\| \mathbf{1}_{\mathbf{x}_{c,i}^T \mathbf{w}_{c,m}^k} \right\|_2 \\
&\leq \frac{1}{\sqrt{M}} \sum_{i=1}^{D_c} \left\| (y_{c,i}^k(t) - y_{c,i}) \right\|_2.
\end{aligned}$$

We have $\left\| (y_{c,i}^k(t) - y_{c,i}) \right\|_2 = \left| (y_{c,i}^k(t) - y_{c,i}) \right|$.
Further, $\left\| (\mathbf{y}_c^k(t) - \mathbf{y}_c) \right\|_2 \leq \left| (\mathbf{y}_c^k(t) - \mathbf{y}_c) \right| \leq \sqrt{D_c} \left\| (\mathbf{y}_c^k(t) - \mathbf{y}_c) \right\|_2$.

Therefore, we arrive at the following:

$$\left\| \frac{\partial \mathcal{L}_c(\mathbf{W}_c^k(t))}{\partial \mathbf{w}_{c,m}^k(t)} \right\|_2 \leq \frac{\sqrt{D}}{\sqrt{M}} \left\| \mathbf{y}_c^k(t) - \mathbf{y}_c \right\|_2. \tag{B4}$$

$\square$

**Proof of Lemma 2**

*Proof.* Gradient descent update rule for local training at the $c$-th FL client in the $t$-the FL round is given by

$$\mathbf{w}_{c,m}^k(t) = \mathbf{w}_{c,m}^{k-1}(t) - \eta \frac{\partial \mathcal{L}_c(\mathbf{W}_c^k(t))}{\partial \mathbf{w}_{c,m}^k(t)}. \tag{B5}$$

Re-arranging the terms and taking $\eta \to 0$, we obtain the following continuous time gradient flow dynamics as given in Lemma 10.1.1 in Arora et al. (2020). Therefore, the dynamics of the parameters at the $c$-th FL client $c$ under the gradient flow setting are as follows:

$$\frac{d}{dk} \mathbf{w}_{c,m}^k(t) = -\frac{\partial \mathcal{L}_c(\mathbf{W}_c^k(t))}{\partial \mathbf{w}_{c,m}^k(t)}. \tag{B6}$$

As discussed before, at the $c$-th client, given an input $\mathbf{x}_{c,i}$, we have the prediction, $y_{c,i}(t) = g(\mathbf{W}_c^k(t), \mathbf{x}_{c,i})$, given by:

$$y_{c,i}(t) = \frac{1}{\sqrt{M}} \sum_{m=1}^{M} b_m \phi(\mathbf{x}_{c,i}^T \mathbf{w}_{c,m}^k(t)), \tag{B7}$$

Using equation B7, and the chain rule of differentiation, the dynamics of prediction for the sample $i \in [D]$ at the $c$-th client across local epochs (indexed by $k \in [1, K]$) in the $t$-th FL round can be computed as follows Du et al. (2018):

$$\frac{d}{dk} y_{c,i}^k(t) = \sum_{m=1}^{M} \left\langle \frac{\partial g(\mathbf{W}_c^k(t))}{\partial \mathbf{w}_{c,m}^k(t)}, \frac{d\mathbf{w}_{c,m}^k(t)}{dk} \right\rangle.$$

Incorporating equation B6 in the above, we obtain:

$$\begin{aligned}
\frac{d}{dk} y_{c,i}^k(t) &= \sum_{m=1}^{M} \left\langle \frac{\partial g(\mathbf{W}_c^k(t))}{\partial \mathbf{w}_{c,m}^k(t)}, -\frac{\partial \mathcal{L}_c(\mathbf{W}_c^k(t))}{\partial \mathbf{w}_{c,m}^k(t)} \right\rangle \\
&= \sum_{r=1}^{m} \left\langle \frac{\partial g(\mathbf{W}_c^k(t))}{\partial \mathbf{w}_{c,m}^k(t)}, \sum_{i=1}^{D} -(y_{c,i} - y_{c,i}^k(t)) \frac{\partial g(\mathbf{W}_c^k(t))}{\partial \mathbf{w}_{c,m}^k(t)} \right\rangle \\
&= \sum_{j=1}^{D} (y_{c,i} - y_{c,i}^k(t)) \left\langle \frac{\partial g(\mathbf{W}_c^k(t))}{\partial \mathbf{w}_{c,m}^k(t)}, \frac{\partial g(\mathbf{W}_c^k(t))}{\partial \mathbf{w}_{c,m}^k(t)} \right\rangle.
\end{aligned}$$

The second step is derived from equation B1, and the third step is a re-arrangement of the second step. The $m$-th column of the local FL-NTK Gram matrix, $\mathbf{J}_c^k(t)$ is given by

$$\mathbf{J}_{c,m}^k(t) = \left\langle \frac{\partial g(\mathbf{W}_c^k(t))}{\partial \mathbf{w}_{c,m}^k(t)}, \frac{\partial g(\mathbf{W}_c^k(t))}{\partial \mathbf{w}_{c,m}^k(t)} \right\rangle. \tag{B8}$$

Therefore, we obtain

$$\frac{d}{dk} y_{c,i}^k(t) = \sum_{j=1}^{D} (y_{c,i} - y_{c,i}^k(t)) J_{c,ij}^k(t). \tag{B9}$$

The above equation can be written in terms of the NTK Gram matrix and the prediction vector as follows:

$$\frac{d}{dk} \mathbf{y}_c^k(t) = \mathbf{J}_c^k(t)(\mathbf{y}_c - \mathbf{y}_c^k(t)).$$

$\square$

**Proof of Claim 1**

*Proof.* From Assumption 2, the probability of exact recovery is given as

$$P\left[ \frac{\|\Delta \widehat{\mathbf{u}}_m(t) - \Delta \mathbf{u}_m(t)\|_2}{\|\Delta \mathbf{u}_m(t)\|_2} < \epsilon \right] > 1 - \delta', \tag{B10}$$

where $\epsilon \in (0, 1)$ and $\delta' \in (0, 0.5)$. As discussed before, $\epsilon$ is the maximum relative reconstruction error induced due to CS, i.e.,

$$\frac{\|\Delta \widehat{\mathbf{u}}_m(t) - \Delta \mathbf{u}_m(t)\|_2}{\|\Delta \mathbf{u}_m(t)\|_2} < \epsilon. \tag{B11}$$

Applying reverse triangle inequality to equation B11, we obtain

$$P\left[-\epsilon < \frac{\|\Delta\widehat{\mathbf{u}}_m(t)\|_2 - \|\Delta\mathbf{u}_m(t)\|_2}{\|\Delta\mathbf{u}_m(t)\|_2} < \epsilon\right] > 1 - \delta' \tag{B12}$$

Assuming the distribution of $\frac{\|\Delta\widehat{\mathbf{u}}_m(t)\|_2 - \|\Delta\mathbf{u}_m(t)\|_2}{\|\Delta\mathbf{u}_m(t)\|_2}$ is symmetrical, we obtain

$$P\left[-\epsilon < \frac{\|\Delta\widehat{\mathbf{u}}_m(t)\|_2 - \|\Delta\mathbf{u}_m(t)\|_2}{\|\Delta\mathbf{u}_m(t)\|_2}\right] = P\left[\frac{\|\Delta\widehat{\mathbf{u}}_m(t)\|_2 - \|\Delta\mathbf{u}_m(t)\|_2}{\|\Delta\mathbf{u}_m(t)\|_2} < \epsilon\right] > 1 - \delta, \tag{B13}$$

where $\delta \in (0, 0.25)$. The above can be re-written as follows

$$P\left[\|\Delta\widehat{\mathbf{u}}_m(t)\| > (1 - \epsilon)\|\Delta\mathbf{u}_m(t)\|\right] = P\left[\|\Delta\widehat{\mathbf{u}}_m(t)\| < (1 + \epsilon)\|\Delta\mathbf{u}_m(t)\|\right] > 1 - \delta. \tag{B14}$$

$\square$

**Proof of Lemma 3**

*Proof.* Local training at the clients in the $t$-th round ($1 \le t \le T$) is not affected by CS as compression is performed post local training just before transmission to the server. Therefore, the order of change in the magnitude of local model parameters in subsequent local epochs should be the same as in the absence of compressed sensing,i.e., $\mathcal{O}(1/\sqrt{M})$ as per Huang et al. (2021a). This lemma can be proved mathematically by adapting the proof from Du et al. (2018) (Lemma 3.3 and corollary 4.1) to local epochs at a client in the $t$-th FL round. Using the gradient descent update rule in equation B5, we bound $\|\mathbf{w}_{c,m}^{k+1}(t) - \mathbf{w}_{c,m}^0(t)\|_2$ as

$$\|\mathbf{w}_{c,m}^{k+1}(t) - \mathbf{w}_{c,m}^0(t)\|_2 \le \eta \sum_{k^{p}rime=0}^{k} \left\|\frac{\partial\mathcal{L}_c(\mathbf{w}_{c,m}^{k'}(t))}{\partial\mathbf{w}_{c,m}^{k'}(t)}\right\|_2. \tag{B15}$$

Using equation B3 and equation B4 that relate the loss function gradient and its norm with the difference in output function and the ground truth at the $c$-th client, we have

$$\begin{aligned}
\|\mathbf{w}_{c,m}^{k+1}(t) - \mathbf{w}_{c,m}^0(t)\|_2 &\le \eta \sum_{k'=0}^{k} \sqrt{\frac{D_c}{M}} \|\mathbf{y}_c - \mathbf{y}_c^{k'}(t)\|_2 \\
&\le \eta \sum_{k'=0}^{\infty} \sqrt{\frac{D_c}{M}} \left(1 - \frac{\eta\lambda_c(t)}{2}\right)^{k'/2} \|\mathbf{y}_c - \mathbf{y}_c^0(t)\|_2 \\
&\le \frac{4\sqrt{D_c}\|\mathbf{y}_c - \mathbf{y}_c^0(t)\|_2}{\sqrt{M}\lambda_c(t)},
\end{aligned} \tag{B16}$$

where $\lambda_c(t)$ is the minimum eigenvalue of $\mathbf{J}_c^0(t)$. In the above set of inequalities, the first inequality follows from the exponential decay in local training loss as per the following equation:

$$\|\mathbf{y}_c{}^k(t) - \mathbf{y}_c\|_2^2 \le \left(1 - \frac{\eta\lambda_c(t)}{2}\right)^k \|\mathbf{y}_c^0(t) - \mathbf{y}_c\|_2^2. \tag{B17}$$

The last inequality holds since we choose $\eta$ such that $\frac{\eta\lambda_c(t)}{2} < 1$, and apply the sum of infinite geometric progression followed by binomial expansion. $\square$

**Proof of Lemma 4**

*Proof.* The parameter aggregation at the server using the FedAvg algorithm is then given by

$$\Delta\mathbf{U}(t) = \frac{1}{C}\sum_{c=1}^{C} \Delta\mathbf{W}_c(t). \tag{B18}$$

Further, the $c$-th client transmits a compressed incremental weight update, $\Delta \widetilde{\mathbf{w}}_c(t)$, instead of $\Delta \mathbf{w}_c(t)$, obtained as follows:

$$\Delta \widetilde{\mathbf{w}}_c(t) = \mathbf{A} \Delta \mathbf{w}_c(t). \tag{B19}$$

From equation B18 and equation B19, we obtain:

$$\Delta \widetilde{\mathbf{u}}_m(t) = \frac{1}{C} \left( \mathbf{A} \sum_{c \in [C]} \Delta \mathbf{w}_{c,m}(t) \right).$$

From Algorithm 1, we have $\Delta \widehat{\mathbf{u}}_m(t) = SSR(\Delta \widetilde{\mathbf{u}}_m(t))$.
We have assumed that the relative reconstruction error in global weight gradients is upper bounded by $'\epsilon'$, i.e.,

$$\|\Delta \mathbf{u}_m(t) - \Delta \widehat{\mathbf{u}}_m(t)\|_2 < \epsilon \|\Delta \mathbf{u}_m(t)\|_2. \tag{B20}$$

Applying reverse triangle inequality, we obtain

$$-\epsilon \|\Delta \mathbf{u}_m(t)\|_2 < \|\Delta \widehat{\mathbf{u}}_m(t)\|_2 - \|\Delta \mathbf{u}_m(t)\|_2 < +\epsilon \|\Delta \mathbf{u}_m(t)\|_2. \tag{B21}$$

or,

$$\|\Delta \widehat{\mathbf{u}}_m(t)\|_2 < (1 + \epsilon) \|\Delta \mathbf{u}_m(t)\|_2. \tag{B22}$$

Applying the gradient descent update rule recursively for datapoints at all the clients for $K$ local epochs and incorporating equation B3, equation B4, and equation B22, we arrive at the following inequality:

$$\Delta \widehat{\mathbf{u}}_m(t) \leq (1 + \epsilon) \frac{b_m \eta}{C\sqrt{M}} \left( \sum_{c,k} \sum_{j \in S_c} (y_j - y_{c,j}^k(t)) \mathbf{x}_j \mathbf{1}_{\mathbf{x}_j^T \mathbf{w}_{c,m}^k(t)} \right), \tag{B23}$$

where $\sum_{c,k} \equiv \sum_{c \in [C]} \sum_{k \in [K]}$. Taking $\ell_2$-norm on both sides of the above expression, we arrive at the following:

$$\|\Delta \widehat{\mathbf{u}}_m(t)\|_2 \leq (1 + \epsilon) \frac{b_m \eta}{C\sqrt{M}} \left\| \sum_{c,k} \sum_{j \in S_c} (y_j - y_{c,j}^k(t)) \mathbf{x}_j \mathbf{1}_{\mathbf{x}_j^T \mathbf{w}_{c,m}^k(t)} \right\|_2.$$

Following Lemma 1 proof, we arrive at the following:

$$\|\Delta \widehat{\mathbf{u}}_m(t)\|_2 \leq \frac{\eta \sqrt{D} (1 + \epsilon)}{C\sqrt{M}} \sum_{k \in [K]} \|\mathbf{y} - \mathbf{y}^k(t)\|_2. \tag{B24}$$

From FL-NTK theory in Huang et al. (2021a), we have

$$\|\mathbf{y} - \mathbf{y}^k(t)\|_2^2 \leq 2(1 + 2\eta DK)^2 \|\mathbf{y} - \mathbf{y}(t)\|_2^2. \tag{B25}$$

Applying the above bound to equation B24, we obtain the following bound.

$$\|\Delta \widehat{\mathbf{u}}_m(t)\|_2 \leq (1 + \epsilon) \frac{2\eta K (1 + 2\eta DK)\sqrt{D}}{C\sqrt{M}} \|\mathbf{y} - \mathbf{y}(t)\|_2, \tag{B26}$$

$$\square$$

**Proof of Lemma 5**

*Proof.* In COFL-MAC, the global weight update step (step 16 in Algorithm 1 calculates the newly reconstructed model weights from the previous ones. Recursively applying the global update step, we arrive at the following:

$$\|\widehat{\mathbf{u}}_m(t+1) - \mathbf{u}_m(0)\|_2 \le \sum_{\tau=0}^{t} \|\Delta\widehat{\mathbf{u}}_m(\tau)\|_2.$$

Incorporating the reconstructed global weight movement in subsequent rounds as per Lemma 4, we obtain

$$\le (1+\epsilon)\sum_{\tau=0}^{t} \frac{2\eta K(1+2\eta DK)\sqrt{D}}{C\sqrt{M}}\|\mathbf{y}-\mathbf{y}(\tau)\|_2$$

$$\le (1+\epsilon)\left(\frac{2\eta K(1+2\eta DK)\sqrt{D}}{C\sqrt{M}}\right)\sum_{\tau=0}^{t}\left(1-\frac{\eta\lambda K}{2C}\right)^{\tau}\|\mathbf{y}-\mathbf{y}(0)\|_2,$$

where $\mathbf{y}$ represents the ground truth and $\lambda = \lambda_{min}(\mathbf{G}(0))$. The above step follows from Lemma 1 and Lemma 3. Next, from the choice of $\eta \le \dfrac{\lambda}{1000\kappa D^2 K}$, where $\kappa$ is the condition number of $\mathbf{G}(0)$, we obtain the following result

$$\|\widehat{\mathbf{u}}_m(t+1) - \mathbf{u}_m(0)\|_2 \le (1+\epsilon)\left(\frac{8\sqrt{D}\|\mathbf{y}-\mathbf{y}(0)\|_2}{\sqrt{M}\lambda}\right). \tag{B27}$$

□

**Proof of Lemma 6**

*Proof.* The global step of COFL-MAC (step 16 of Algorithm 1) can be upper-bounded using triangular inequality as follows.

$$\|\widehat{\mathbf{u}}_m(t) - \mathbf{u}_m(0)\|_2 \le \sum_{\tau=0}^{t-1} \|\Delta\widehat{\mathbf{u}}_m(\tau)\|_2. \tag{B28}$$

Using equation B22, we obtain

$$\|\widehat{\mathbf{u}}_m(t) - \mathbf{u}_m(0)\|_2 \le (1+\epsilon)\sum_{\tau=0}^{t-1} \|\Delta\mathbf{u}_m(\tau)\|_2. \tag{B29}$$

Considering the maximum value of both sides of the above expression, we have

$$\max_{m\in[M]} \|\widehat{\mathbf{u}}_m(t) - \mathbf{u}_m(0)\|_2 \le \max_{m\in[M]}\left((1+\epsilon)\sum_{\tau=0}^{t-1} \|\Delta\mathbf{u}_m(\tau)\|_2\right). \tag{B30}$$

Using the definition of $R$ from the lemma statement, we obtain

$$\max_{m\in[M]} \|\widehat{\mathbf{u}}_m(t) - \mathbf{u}_m(0)\|_2 \le (1+\epsilon)R. \tag{B31}$$

□

Therefore, Lemma 6 states that if, in vanilla FL, the movement of global weights in the $t$-th communication round w.r.t. initialization is within a ball of radius $R$, then the movement of reconstructed global weights w.r.t. initialization in COFL-MAC is within a ball of radius $(1+\epsilon)R$. Let us define

$$\widehat{R} \triangleq (1+\epsilon)R. \tag{B32}$$

With this definition, we will define events $\mathcal{B}_{i,m}$ and $\mathcal{E}_{i,m}$ in the sequel, which will help in proving the convergence by preventing excessive client drift and maintaining NTK approximation theory.

**Proof of Lemma 7**

*Proof.* Let us redefine the events $\mathcal{B}_{i,m}$, $\mathcal{E}_{i,m}$, and $\mathcal{E}_{i,m} \mid \mathcal{B}_{i,m}$. The event $\mathcal{B}_{i,m}$ for vanilla FL is defined as

$$\mathcal{B}_{i,m} = \left\{ \mathbf{z} : \|\mathbf{z} - \mathbf{u}_m(0)\|_2 \leq R, \mathbf{1}_{\mathbf{x}_i^T \mathbf{u}_m(0)} \neq \mathbf{1}_{\mathbf{x}_i^T \mathbf{z}} \right\} \tag{B33}$$

The event $\mathcal{E}_{i,m}$ for COFL-MAC is defined as

$$\mathcal{E}_{i,m} = \left\{ \widehat{\mathbf{z}} : \|\widehat{\mathbf{z}} - \mathbf{u}_m(0)\|_2 \leq \widehat{R}, \mathbf{1}_{\mathbf{x}_i^T \mathbf{u}_m(0)} \neq \mathbf{1}_{\mathbf{x}_i^T \widehat{\mathbf{z}}} \right\}. \tag{B34}$$

Further $\mathcal{E}_{i,m} \mid \mathcal{B}_{i,m}$ has been defined as follows:

$$\begin{aligned} \mathcal{E}_{i,m} \mid \mathcal{B}_{i,m} = \\ \{\widehat{\mathbf{z}} : \|\widehat{\mathbf{z}} - \mathbf{u}_m(0)\|_2 \leq \widehat{R}, \mathbf{1}_{\mathbf{x}_i^T \mathbf{u}_m(0)} \neq \mathbf{1}_{\mathbf{x}_i^T \widehat{\mathbf{z}}} \mid \\ \mathbf{z} : \|\mathbf{z} - \mathbf{u}_m(0)\|_2 \leq R, \mathbf{1}_{\mathbf{x}_i^T \mathbf{u}_m(0)} \neq \mathbf{1}_{\mathbf{x}_i^T \mathbf{z}} \}, \end{aligned} \tag{B35}$$

In the context of COFL-MAC, we assume that the reconstructed global weight vector in a particular round $t$ is $\widehat{\mathbf{z}}$. Recursively applying the global step of FL, we have

$$\|\mathbf{z} - \mathbf{u}_m(0)\|_2 \leq \sum_{\tau=1}^{t-1} \|\Delta \mathbf{u}_m(\tau)\|_2. \tag{B36}$$

Using the definition of $R$ from Lemma 6 we obtain

$$\|\mathbf{z} - \mathbf{u}_m(0)\|_2 \leq R. \tag{B37}$$

Analogously, the global step of COFL-MAC is given by

$$\|\widehat{\mathbf{z}} - \mathbf{u}_m(0)\|_2 \leq \sum_{\tau=1}^{t-1} \|\Delta \widehat{\mathbf{u}}_m(t)\|_2 \leq (1+\epsilon) \sum_{\tau=1}^{t-1} \|\Delta \mathbf{u}_m(\tau)\|_2,$$

where the last inequality is obtained using equation B22. Therefore, we have

$$\|\widehat{\mathbf{z}} - \mathbf{u}_m(0)\|_2 \leq \widehat{R}, \tag{B38}$$

where $\widehat{R} = (1+\epsilon)R$, as required for the result in equation B35. In other words, $\|\Delta \widehat{\mathbf{u}}_m(t)\|_2 < (1+\epsilon)\|\Delta \mathbf{u}_m(t)\|_2$ implies the event $\mathcal{E}_{i,m} \mid \mathcal{B}_{i,m}$, i.e., equation B22 along with $\mathcal{B}_{i,m}$ implies $\mathcal{E}_{i,m}$. □

**Proof of Proposition 1**

*Proof.* In Lemma 7, we proved that the event $\|\Delta \widehat{\mathbf{u}}_m(t)\|_2 \leq (1+\epsilon)\|\Delta \mathbf{u}_m(t)\|_2$ implies the occurrence of the event $\mathcal{E}_{i,m} \mid \mathcal{B}_{i,m}$. If the occurrence of an event $\mathcal{A}$ implies the occurrence of another event $\mathcal{B}$, then $\mathcal{A} \subseteq \mathcal{B}$ and therefore, $P[\mathcal{B}] \geq P[\mathcal{A}]$ pro. Substituting $\mathcal{A}$ as $\|\Delta \widehat{\mathbf{u}}_m(t)\|_2 \leq (1+\epsilon)\|\mathbf{u}_m(t)\|_2$ and $\mathcal{B}$ as $\mathcal{E}_{i,m} \mid \mathcal{B}_{i,m}$, we obtain

$$\begin{aligned} P[\mathcal{E}_{i,m} \mid \mathcal{B}_{i,m}] &\geq P\big[\|\Delta \widehat{\mathbf{u}}_m(t)\|_2 \leq (1+\epsilon)\|\mathbf{u}_m(t)\|_2\big] \\ &\geq 1 - \delta. \end{aligned} \tag{B39}$$

The last step comes from equation B14. □

**Proof of Claim 2**

*Proof.* From Proposition 1, we have

$$P[\mathcal{E}_{i,m} \mid \mathcal{B}_{i,m}] > 1 - \delta.$$

By the law of total probability, we have

$$P[\mathcal{E}_{i,m}] = P[\mathcal{E}_{i,m} \mid \mathcal{B}_{i,m}]P[\mathcal{B}_{i,m}] + P[\mathcal{E}_{i,m} \mid \overline{\mathcal{B}}_{i,m}]P[\overline{\mathcal{B}}_{i,m}]. \tag{B40}$$

From equation B35, we have

$$P\left[\mathcal{E}_{i,m} \mid \overline{\mathcal{B}}_{i,m}\right] = P\left[\|\widehat{\mathbf{z}} - \mathbf{u}_m(0)\|_2 < (1+\epsilon)R \mid \|\mathbf{z} - \mathbf{u}_m(0)\|_2 > R\right].$$

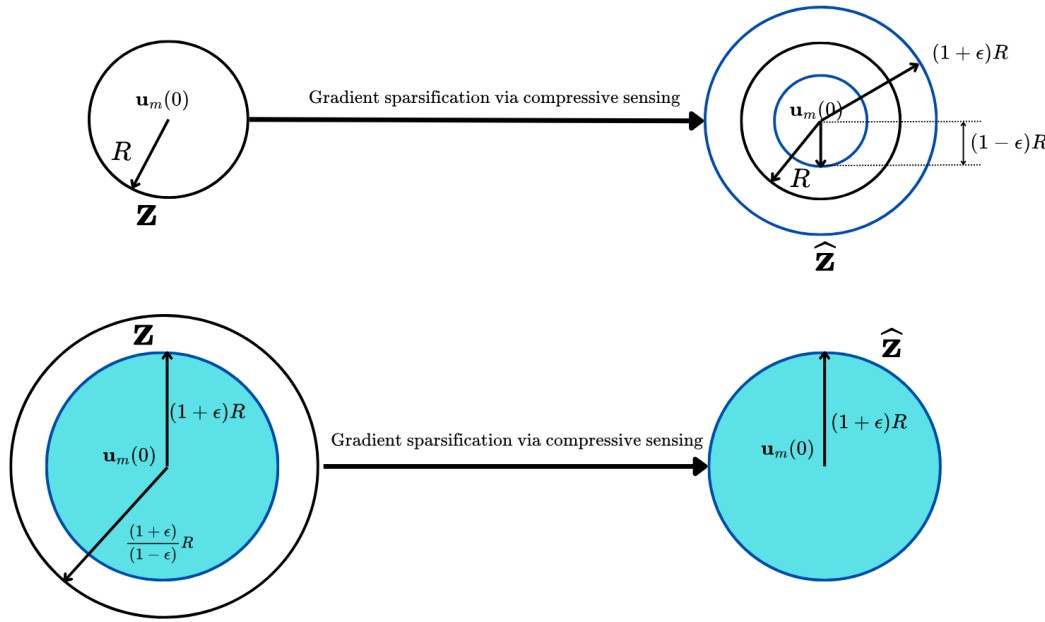

Figure 8

The event $\|\widehat{\mathbf{z}} - \mathbf{u}_m(0)\|_2 < (1+\epsilon)R \mid \|\mathbf{z} - \mathbf{u}_m(0)\|_2 > R$ happens when the model weights in FL-NTK lie in a shell between radii $(1+\epsilon)R$ and $\frac{(1+\epsilon)}{(1-\epsilon)}R$ as shown in Fig. 8, such that

$$(1+\epsilon)R \leq \|\mathbf{z} - \mathbf{u}_m(0)\|_2 \leq \frac{(1+\epsilon)}{(1-\epsilon)}R. \tag{B41}$$

From equation B14, we have

$$P\left[\|\Delta\widehat{\mathbf{u}}_m(t)\|_2 > (1-\epsilon)\|\Delta\mathbf{u}_m(t)\|_2\right] > 1 - \delta.$$

The above can be re-written as

$$P\left[\|\Delta\widehat{\mathbf{u}}_m(t)\|_2 < (1-\epsilon)\|\Delta\mathbf{u}_m(t)\|_2\right] \leq \delta. \tag{B42}$$

From the global step of COFL-MAC, we have

$$\|\widehat{\mathbf{z}} - \mathbf{u}_m(0)\|_2 \leq \sum_{\tau=0}^{t-1} \|\Delta\widehat{\mathbf{u}}_m(\tau)\|_2.$$

Therefore, we have the following with a probability at most $\delta$

$$\|\widehat{\mathbf{z}} - \mathbf{u}_m(0)\|_2 \leq (1-\epsilon) \sum_{\tau=0}^{t-1} \|\Delta\widehat{\mathbf{u}}_m(\tau)\|_2.$$

From equation B41, we have the outer radius of the shell as

$$\|\mathbf{z} - \mathbf{u}_m(0)\|_2 \leq \left(\frac{1+\epsilon}{1-\epsilon}\right) R. \tag{B43}$$

Applying equation B43, we obtain

$$\|\widehat{\mathbf{z}} - \mathbf{u}_m(0)\|_2 \leq (1-\epsilon) \left(\frac{1+\epsilon}{1-\epsilon}\right) R.$$

Therefore, we have

$$\|\widehat{\mathbf{z}} - \mathbf{u}_m(0)\|_2 \leq (1+\epsilon)R. \tag{B44}$$

with a probability at most $\delta$. The event $\|\Delta\widehat{\mathbf{u}}_m(t)\|_2 < (1-\epsilon)\|\Delta\mathbf{u}_m(t)\|_2$ implies the event $\mathcal{E}_{i,m}$ given equation B41. In other words, the event $\|\Delta\widehat{\mathbf{u}}_m(t)\|_2 < (1-\epsilon)\|\Delta\mathbf{u}_m(t)\|_2$ implies the event $\mathcal{E}_{i,m}$ given the occurrence of event $\overline{\mathcal{B}}_{i,m}$. Therefore, we have

$$P[\mathcal{E}_{i,m} \mid \overline{\mathcal{B}}_{i,m}] \geq P[\|\Delta\widehat{\mathbf{u}}_m(t)\|_2 < (1-\epsilon)\|\Delta\mathbf{u}_m(t)\|_2]. \tag{B45}$$

We have deduced earlier that

$$P[\|\Delta\widehat{\mathbf{u}}_m(t)\|_2 < (1-\epsilon)\|\Delta\mathbf{u}_m(t)\|_2] < \delta. \tag{B46}$$

By the anti-concentration inequality of Gaussian random variables (Lemma 14), we have

$$P[\mathcal{B}_{i,m}] = P[|\mathbf{x}_j^T \mathbf{u}_m(0)| < R] \leq \frac{2R}{\sqrt{2\pi}}. \tag{B47}$$

Therefore, we obtain

$$P[\overline{\mathcal{B}}_{i,m}] \geq 1 - \frac{2R}{\sqrt{2\pi}}. \tag{B48}$$

Applying the law of total probability, we obtain

$$\begin{aligned} P[\mathcal{E}_{i,m}] &\leq (1-\delta) \times \frac{2R}{\sqrt{2\pi}} + 1 \cdot (1 - \frac{2R}{\sqrt{2\pi}}), \\ &\leq R - \delta R + 1 - R \\ &\leq 1 - \delta R \tag{B49} \\ &\leq R'. \tag{B50} \end{aligned}$$

$\square$

**Proof of Lemma 8**

*Proof.* In this lemma, we provide a bound for the discrepancy, $\|\mathbf{G}^k(t) - \mathbf{G}(0)\|_F$. We rewrite the definition of COFL-MAC Gram matrix $(\mathbf{G}_c^k(t))$ elements as follows:

$$G_{c,ij}^k(t) = \frac{1}{M}\mathbf{x}_i^T \mathbf{x}_j \sum_{m=1}^M \mathbf{1}_{\mathbf{x}_i^T \widehat{\mathbf{u}}_m(t), \mathbf{x}_j^T \mathbf{w}_{c,m}^k(t)}. \tag{B51}$$

Further, the global COFL-MAC Gram matrix, $\mathbf{G}^k(t)$ is given by

$$\mathbf{G}^k(t) = \frac{1}{C}\sum_{c\in C} \mathbf{G}_c^k(t). \tag{B52}$$

Further, $\|\mathbf{G}^k(t) - \mathbf{G}(0)\|_F$ can be bounded as follows

$$\|\mathbf{G}^k(t) - \mathbf{G}(0)\|_F^2 = \sum_{i=1}^D \sum_{j=1}^D |G_{ij}^k(t) - G_{ij}(0)|^2.$$

Applying equation B51, we obtain

$$\|\mathbf{G}^k(t) - \mathbf{G}(0)\|_F^2 \leq \frac{1}{M^2 C^2}\sum_{i=1}^D \sum_{c\in[C]} \sum_{j\in\mathcal{S}_c} \left(\sum_{m=1}^M \mathbf{1}_{\mathbf{x}_i^T \widehat{\mathbf{u}}_m(t), \mathbf{x}_j^T \mathbf{w}_{c,m}^k(t)} - \mathbf{1}_{\mathbf{x}_i^T \mathbf{u}_m(0), \mathbf{x}_j^T \mathbf{u}_m(0)}\right)^2$$

$$\leq \frac{1}{M^2 C^2}\sum_{i=1}^D \sum_{c\in[C]} \sum_{j\in\mathcal{S}_c} \left(\sum_{m=1}^M s_{c,m,ij}\right)^2, \tag{B53}$$

where the random variable $s_{c,m,ij}$ is defined as follows

$$s_{c,m,ij} := \mathbf{1}_{\mathbf{x}_i^T \widehat{\mathbf{u}}_m(t), \mathbf{x}_j^T \mathbf{w}_{c,m}^k(t)} - \mathbf{1}_{\mathbf{x}_i^T \mathbf{u}_m(0), \mathbf{x}_j^T \mathbf{u}_m(0)}. \tag{B54}$$

We fix the subscripts $c, i$ and $j$, and denote $s_{c,m,ij}$ as $\psi_{ij}^m$. Here, $\{\psi_{ij}^m\}_{m=1}^M$ are mutually independent random variables.

From equation B34, we see that the occurrence of $\mathcal{E}_{i,m}$ entails $\mathbf{1}_{\mathbf{x}_i^T \mathbf{u}_m(0)} \neq \mathbf{1}_{\mathbf{x}_i^T \widehat{\mathbf{z}}}$ and the occurrence of $\mathcal{E}_{j,m}$ entails $\mathbf{1}_{\mathbf{x}_j^T \mathbf{u}_m(0)} \neq \mathbf{1}_{\mathbf{x}_j^T \widehat{\mathbf{z}}}$. This implies that $\neg\mathcal{E}_{i,m}$ occurs if $\mathbf{1}_{\mathbf{x}_i^T \mathbf{u}_m(0)} = \mathbf{1}_{\mathbf{x}_i^T \widehat{\mathbf{z}}}$ and $\neg\mathcal{E}_{j,m}$ occurs if $\mathbf{1}_{\mathbf{x}_j^T \mathbf{u}_m(0)} = \mathbf{1}_{\mathbf{x}_j^T \widehat{\mathbf{z}}}$. Hence, if $\neg\mathcal{E}_{i,m}$ and $\neg\mathcal{E}_{j,m}$ jointly occur, then

$$\left|\mathbf{1}_{\mathbf{x}_i^T \widehat{\mathbf{u}}_m(t), \mathbf{x}_j^T \mathbf{w}_{c,m}^k(t)} - \mathbf{1}_{\mathbf{x}_i^T \mathbf{u}_m(0), \mathbf{x}_j^T \mathbf{u}_m(0)}\right| = 0. \tag{B55}$$

If $\mathcal{E}_{i,m}$ or $\mathcal{E}_{j,m}$ happen, then

$$\left|\mathbf{1}_{\mathbf{x}_i^T \widehat{\mathbf{u}}_m(t), \mathbf{x}_j^T \mathbf{w}_{c,m}^k(t)} - \mathbf{1}_{\mathbf{x}_i^T \mathbf{u}_m(0), \mathbf{x}_j^T \mathbf{u}_m(0)}\right| \leq 1. \tag{B56}$$

Analogous to the event $\mathcal{E}_{i,m}$ for COFL-MAC weights, we have the event $\mathcal{B}_{i,m}$ for vanilla FL weights. From the definition of event $\mathcal{B}_{i,m}$ in equation B33 and by the anti-concentration inequality of Gaussian random variables (refer Lemma 14), we arrive at the following:

$$P[\mathcal{B}_{i,m}] = P[|\mathbf{x}_j^T \mathbf{u}_m(0)| < R] \leq \frac{2R}{\sqrt{2\pi}}. \tag{B57}$$

From Claim 2, we have

$$P[\mathcal{E}_{i,m}] \leq R', \tag{B58}$$

where $R' = 1 - \delta R$. Therefore, we obtain

$$\mathbb{E}[\psi_{ij}^m] \leq \mathbb{E}\left[\mathbf{1}_{\mathcal{E}_{i,m} \vee \mathcal{E}_{j,m}}\right] \leq 2R'. \tag{B59}$$

Also, the variance of $\psi_{ij}^m$, $Var(\psi_{ij}^m)$ can be obtained as follows:

$$
\begin{aligned}
Var(\psi_{ij}^m) = \mathbb{E}\left[\left(\psi_{ij}^m - \mathbb{E}[\psi_{ij}^m]\right)^2\right] &= \mathbb{E}[\psi_{ij}^{m\,2}] - \mathbb{E}_{\mathbf{w}_m}[\psi_{ij}^m]^2 \\
&\leq \mathbb{E}[\psi_{ij}^{m\,2}] \\
&\leq \mathbb{E}\left[\left(\mathbf{1}_{\mathcal{E}_{i,m} \vee E_{j,m}}\right)^2\right] \\
&\leq 2R'. \tag{B60}
\end{aligned}
$$

Also, $|\psi_{ij}^m| \leq 1$.

Therefore, from Bernstein's inequality (Lemma 15), we have for all $\xi > 0$,

$$P\left[\frac{1}{M}\sum_{m=1}^{M}\left(\psi_{ij}^m - \mathbb{E}\left[\psi_{ij}^m\right]\right) \geq \xi\right] \leq \exp\left(-\frac{M\xi^2/2}{Var(\psi_{ij}^m) + \xi/3}\right).$$

Applying equation B60, the above can be written as

$$P\left[\frac{1}{M}\sum_{m=1}^{M}\psi_{ij}^m - \mathbb{E}\left[\psi_{ij}^m\right] \geq \xi\right] \leq \exp\left(-\frac{M\xi^2/2}{2R' + \xi/3}\right).$$

The above can be written as follows

$$1 - P\left[\frac{1}{M}\sum_{m=1}^{M}\psi_{ij}^m - \mathbb{E}\left[\psi_{ij}^m\right] < \xi\right] \leq \exp\left(-\frac{M\xi^2/2}{2R' + \xi/3}\right).$$

Applying equation B59, we obtain

$$1 - P\left[\sum_{m=1}^{M}\psi_{ij}^m < 2MR' + M\xi\right] \leq \exp\left(-\frac{Mt^2/2}{2R' + \xi/3}\right),$$

which can further be written as follows

$$P\left[\sum_{m=1}^{M}\psi_{ij}^m \geq 2MR' + M\xi\right] \leq \exp\left(-\frac{M\xi^2/2}{2R' + \xi/3}\right).$$

By setting $\xi = 3R - 2R'$, we obtain

$$
\begin{aligned}
P\left[\sum_{m=1}^{M}\psi_{ij}^m \geq 3MR\right] &\leq \exp\left(-\frac{M(3R - 2R')^2/2}{2R' + (3R - 2R')/3}\right) \\
&\leq \exp\left(\frac{-MR_1}{4R'/9R_1 + 2/9}\right),
\end{aligned}
$$

where $R_1 = R - 2/3R'$. It follows that

$$P\left[\frac{1}{M}\sum_{m=1}^{M}\psi_{ij}^m \geq 3R\right] \leq \exp\left(\frac{-MR_1}{4R'/9R_1 + 2/9}\right). \tag{B61}$$

Alternatively,

$$P\left[\frac{1}{M}\sum_{m=1}^{M}\psi_{ij}^m < 3R\right] \geq 1 - \exp\left(\frac{-MR_1}{4R'/9R_1 + 2/9}\right). \tag{B62}$$

From B53 and B62, we obtain

$$\|\mathbf{G}^k(t) - \mathbf{G}(0)\|_F^2 = \sum_{i=1}^{D} \sum_{j=1}^{D} |G_{ij}^k(t) - G_{ij}(0)|^2$$

$$\leq \frac{1}{C^2} \sum_{i=1}^{D} \sum_{j=1}^{D} 9R^2. \qquad (B63)$$

Therefore, we have

$$\|\mathbf{G}^k(t) - \mathbf{G}(0)\|_F < \frac{2DR}{C},$$

with a probability at least $1 - \exp\left(\frac{-MR_1}{4R'/9R_1 + 2/9}\right)$. Here $R_1 = R - (2/3)R'$.

$\square$

**Proof of Claim 3**

*Proof.* In this claim, we provide an upper bound on the value of $\zeta_i$ and show that this value is small. Since the set $\overline{\mathcal{Q}}_i$ defines the index set of the event $\mathcal{E}_{i,m}$ as given in equation B34. As noted earlier, this event is analogous to the event $\mathcal{B}_{i,m}$ as given in equation B33. Note that the event $\mathcal{B}_{i,m}$ occurs if and only if $|\mathbf{x}_i^T \mathbf{u}_m(0)| \leq R$, as shown in Huang et al. (2021a); Song & Yang (2019); Du et al. (2018). Analogously, we conjecture that the event $\mathcal{E}_{i,m}$ happens if and only if $|\mathbf{x}_i^T \mathbf{u}_m(0)| \leq (1 + \epsilon)R$. Furthermore, since, $\mathbf{u}_m(0) \sim \mathcal{N}(0, \mathbf{I}_D)$ and $\|\mathbf{x}_i\|_2 = 1$, the linear combination given by $\mathbf{x}_i^T \mathbf{u}_m(0) \sim \mathcal{N}(0, 1)$.

The above conjecture implies that for all $m \in \overline{\mathcal{Q}}_i$, we have

$$P[\mathbf{1}_{m \in \overline{Q}_i}] = P[\mathcal{E}_{i,m}] = P[|\mathbf{x}_i^T \mathbf{u}_m(0)| \leq (1 + \epsilon)R]. \qquad (B64)$$

Interchanging probability and the expected value for the indicator variable and using equation B50, we have

$$\mathbb{E}[\mathbf{1}_{m \in \overline{Q}_i}] = P[\mathcal{E}_{i,m}] \leq R'. \qquad (B65)$$

Further, the variance, $Var(\mathbf{1}_{m \in \overline{Q}_i})$ is bounded as follows:

$$Var(\mathbf{1}_{m \in \overline{Q}_i}) = \mathbb{E}\left[(\mathbf{1}_{m \in \overline{Q}_i} - \mathbb{E}[\mathbf{1}_{m \in \overline{Q}_i}])^2\right]$$

$$= \mathbb{E}[\mathbf{1}_{m \in \overline{Q}_i}^2] - \mathbb{E}[\mathbf{1}_{m \in \overline{Q}_i}]^2$$

$$\leq \mathbb{E}[\mathbf{1}_{m \in \overline{Q}_i}^2]$$

$$\leq R'. \qquad (B66)$$

Therefore, $|\mathbf{1}_{m \in \overline{Q}_i} - \mathbb{E}[\mathbf{1}_{m \in \overline{Q}_i}]| \leq 1$.

Moreover, random variables, $\{\mathbf{1}_{m \in \overline{Q}_i}\}_{m=1}^M$ are i.i.d. Therefore, from Bernstein's inequality (Lemma 15), we have for all $\xi > 0$,

$$P\left[\frac{1}{M} \sum_{m=1}^{M} \left(\mathbf{1}_{m \in \overline{Q}_i} - \mathbb{E}\left[\mathbf{1}_{m \in \overline{Q}_i}\right]\right) \geq t\right] \leq \exp\left(-\frac{M\xi^2/2}{Var(\mathbf{1}_{m \in \overline{Q}_i}) + \xi/3}\right).$$

Substituting $\zeta_i = \sum_{m=1}^{M} \mathbf{1}_{m \in \overline{Q}_i}$, and applying equation B66, the above can be written as

$$P\left[\frac{1}{M}\zeta_i - \mathbb{E}\left[\mathbf{1}_{m \in \overline{Q}_i}\right] \geq \xi\right] \leq \exp\left(-\frac{M\xi^2/2}{R' + \xi/3}\right).$$

The above can be written as follows

$$1 - P\left[\frac{1}{M}\zeta_i - \mathbb{E}\left[\mathbf{1}_{m \in \overline{Q}_i}\right] < \xi\right] \leq \exp\left(-\frac{M\xi^2/2}{R' + \xi/3}\right).$$

Applying equation B65, we arrive at

$$1 - P\left[\zeta_i < MR' + M\xi\right] \leq \exp\left(-\frac{M\xi^2/2}{R' + \xi/3}\right).$$

Which can further be written as follows

$$P\left[\zeta_i \geq MR' + M\xi\right] \leq \exp\left(-\frac{M\xi^2/2}{R' + \xi/3}\right).$$

By setting $\xi = 4R - R'$, we have

$$P\left[\zeta_i \geq 4MR\right] \leq \exp\left(\frac{-MR_2}{\frac{R'}{8R_2} + 1/6}\right), \tag{B67}$$

where, $R_2 = R - R'/4$. Alternatively,

$$P\left[\zeta_i < 4MR\right] \leq 1 - \exp\left(\frac{-MR_2}{\frac{R'}{8R_2} + 1/6}\right). \tag{B68}$$

$\square$

Next, we propose the following lemma regarding the complementary Gram matrix of COFL-MAC.

**Proof of Lemma 9**

*Proof.* In this lemma, we obtain an upper bound on the norm of the complementary Gram matrix, $\mathbf{G}^k(t)^\perp$. We rewrite the definition of the elements of the complementary COFL-MAC-NTK matrix, $\mathbf{G}_c^k(t)^\perp$ as follows

$$G_{c,ij}^k(t)^\perp = \frac{1}{M}\mathbf{x}_i^T\mathbf{x}_j \sum_{m \in \overline{\mathcal{Q}}_i} \mathbf{1}_{\mathbf{x}_i^T\widehat{\mathbf{u}}_m(t), \mathbf{x}_j^T\mathbf{w}_{c,m}^k(t)}. \tag{B69}$$

Further, the global complementary COFL-MAC-NTK matrix, $\mathbf{G}^k(t)^\perp$ is obtained as follows:

$$\mathbf{G}^k(t)^\perp = \frac{1}{C}\sum_{c \in C} \mathbf{G}_c^k(t)^\perp. \tag{B70}$$

From the definition of COFL-MAC Gram matrix elements in equation B69, we have

$$\|\mathbf{G}^k(t)^\perp\|_F^2 = \sum_{i=1}^{D}\sum_{j=1}^{D}(G_{ij}^k(t)^\perp)^2. \tag{B71}$$

From the definition of COFL-MAC Gram matrix elements in equation B69, we arrive at

$$\|\mathbf{G}^k(t)^\perp\|_F^2 = \frac{1}{C^2}\sum_{i=1}^{D}\sum_{c \in [C]}\sum_{j \in S_c}\left(\frac{1}{M}\mathbf{x}_i^T\mathbf{x}_j \sum_{m \in \overline{\mathcal{Q}}_i} \mathbf{1}_{\mathbf{x}_i^T\widehat{\mathbf{u}}_m(t), \mathbf{x}_j^T\mathbf{w}_{c,m}^k(t)}\right)^2$$

$$= \frac{1}{C^2}\sum_{i=1}^{D}\sum_{c \in [C]}\sum_{j \in S_c}\left(\frac{\mathbf{x}_i^T\mathbf{x}_j}{M}\right)^2\left(\sum_{m=1}^{M} \mathbf{1}_{\mathbf{x}_i^T\widehat{\mathbf{u}}_m(t), \mathbf{x}_j^T\mathbf{w}_{c,m}^k(t)}\mathbf{1}_{m \in \overline{\mathcal{Q}}_i}\right)^2, \tag{B72}$$

where we replace $\sum_{m \in \overline{Q}_i}$ by $\mathbf{1}_{m \in \overline{Q}_i}$ within the summation. From Assumption 2, we have $\|\mathbf{x}_i\|_2 = 1$ and $\|\mathbf{x}_j\|_2 = 1$ which leads to $\mathbf{x}_i^T \mathbf{x}_j < 1$. Thus, we arrive at the following:

$$\|\mathbf{G}^k(t)^{\perp}\|_F^2 \leq \frac{1}{M^2 C^2} \sum_{i=1}^{D} \sum_{c \in [C]} \sum_{j \in S_c} \Big( \sum_{m=1}^{M} \mathbf{1}_{\mathbf{x}_i^T \hat{\mathbf{u}}_m(t), \mathbf{x}_j^T \mathbf{w}_{c,m}^k(t)} \mathbf{1}_{m \in \overline{Q}_i} \Big)^2. \tag{B73}$$

Since the value of the indicator variable is either zero or one, we arrive at the following

$$\|\mathbf{G}^k(t)^{\perp}\|_F^2 \leq \frac{1}{M^2 C^2} \sum_{i=1}^{D} \sum_{c \in [C]} \sum_{j \in S_c} \Big( \sum_{m=1}^{M} \mathbf{1}_{m \in \overline{Q}_i} \Big)^2. \tag{B74}$$

$$\|\mathbf{G}^k(t)^{\perp}\|_F^2 \leq \frac{1}{M^2 C^2} \sum_{i=1}^{D} \sum_{c \in [C]} D_c \Big( \sum_{m=1}^{M} \mathbf{1}_{m \in \overline{Q}_i} \Big)^2. \tag{B75}$$

Since $\sum_{c \in [C]} D_c = D$, we obtain

$$\|\mathbf{G}^k(t)^{\perp}\|_F^2 = \leq \frac{D}{M^2 C^2} \sum_{i=1}^{D} \Big( \sum_{m=1}^{M} \mathbf{1}_{m \in \overline{Q}_i} \Big)^2. \tag{B76}$$

We rewrite the definition of $\zeta_i$ as follows:

$$\zeta_i = \sum_{m=1}^{M} \mathbf{1}_{m \in \overline{Q}_i}. \tag{B77}$$

From the above definition, we arrive at

$$\|\mathbf{G}^k(t)^{\perp}\|_F^2 \leq \frac{D}{M^2} \sum_{i=1}^{D} \zeta_i^2.$$

Applying Claim 3, we obtain

$$\|\mathbf{G}^k(t)^{\perp}\|_F^2 \leq \frac{D}{M^2} \cdot D \cdot (4MR)^2 = 16D^2 R^2$$

with probability at least $1 - D \exp \left( \frac{-MR_2}{\frac{R'}{8R_2} + 1/6} \right)$.

Therefore, we conclude this lemma with the following:

$$\|\mathbf{G}^k(t)^{\perp}\|_F < \frac{4DR}{C} \tag{B78}$$

with a probability of at least $1 - D \exp \left( \frac{-MR_2}{\frac{R'}{8R_2} + 1/6} \right)$ over random initialization, where $R_2 = R - R'/4$. $\qquad \square$

**Proof of Lemma 10**

*Proof.* In this lemma, we will establish an upper bound for the term $C_1$. We rewrite the expression for $C_1$ as follows:

$$C_1 = -(1 + \epsilon) \frac{2\eta}{C} \sum_{i,k,c,j} (y_i - y_i(t))(y_j - y_{c,j}^{(k)}(t)) G_{c,ij}^k(t), \tag{B79}$$

where

$$\sum_{i,k,c,j} \equiv \sum_{i \in [D]} \sum_{k \in [K]} \sum_{c \in [C]} \sum_{j \in S_c} . \tag{B80}$$

Adding and subtracting the term $G_{ij}(0))$ from $G_{c,ij}^k(t)$, we obtain

$$\begin{aligned} C_1 &= -\frac{2(1+\epsilon)\eta}{C} \sum_{i,k,c,j} (y_i - y_i(t))(y_j - y_{c,j}^k(t))(G_{c,ij}^k(t) - G_{ij}(0)) \\ &\quad - \frac{2(1+\epsilon)\eta}{C} \sum_{i,k,c,j} (y_i - y_i(t))(y_j - y_{c,j}^k(t))G_{ij}(0), \end{aligned} \tag{B81}$$

Adding and subtracting the term $y_j(t)$ from the term $(y_j - y_c^{(k)}(t)_j)$, we arrive at

$$C_1 = C_{11} + C_{12} + C_{13}, \tag{B82}$$

where

$$C_{11} = -(1+\epsilon)\frac{2\eta}{C} \sum_{i,k,c,j} (y_i - y_i(t))(y_j - y_{c,j}^k(t))(G_{c,ij}^k(t) - G_{ij}(0)), \tag{B83}$$

$$C_{12} = -(1+\epsilon)\frac{2\eta}{C} \sum_{i,k,c,j} (y_i - y_i(t))(y_j(t) - y_{c,j}^k(t))G_{ij}(0), \tag{B84}$$

$$C_{13} = -(1+\epsilon)\frac{2\eta K}{C} \sum_{i,j} (y_i - y_i(t))(y_j - y_j(t))G_{ij}(0), \tag{B85}$$

where

$$\sum_{i,j} \equiv \sum_{i \in [D]} \sum_{j \in [D]} . \tag{B86}$$

Therefore, $C_{11}$ can be bounded as per as follows:

$$C_{11} = -(1+\epsilon)2\eta \sum_{i,k,c,j} (y_i - y_i(t))(y_j - y_{c,j}^{(k)}(t))(G_{c,ij}^k(t) - G_{ij}(0)). \tag{B87}$$

The above can be rewritten in the vector notation using COFL-MAC Gram matrices $\mathbf{G}^k(t)$, and $\mathbf{G}(0)$, the global and local predictions for the $t$-th round given by $\mathbf{y}(t)$ and $\mathbf{y}^k(t)$, respectively, and the ground truth, $\mathbf{y}$, as follows

$$C_{11} = -(1+\epsilon)2\eta \sum_{k \in [K]} (\mathbf{y} - \mathbf{y}(t))^T (\mathbf{G}^k(t) - \mathbf{G}(0))(\mathbf{y} - \mathbf{y}^k(t)), \tag{B88}$$

where the Gram matrix $\mathbf{G}^k(t) \in \mathbb{R}^{D \times D}$ is obtained by combining the $S_c$ columns of the Gram matrices $\mathbf{G}_c^k(t)$ for all $c \in [C]$. Using triangle inequality followed by Cauchy Schwarz inequality, we obtain an upper bound on the term $C_{11}$ as follows

$$C_{11} \le -(1+\epsilon)2\eta \sum_{k \in [K]} \|\mathbf{y} - \mathbf{y}(t)\|_2 \|\mathbf{y} - \mathbf{y}^k(t)\|_2 \|\mathbf{G}^k(t) - \mathbf{G}(0)\|_F. \tag{B89}$$

From Lemma 8 we have

$$\|\mathbf{G}^k(t) - \mathbf{G}(0)\|_F \le \frac{2DR}{C}. \tag{B90}$$

From equation 15 from Huang et al. (2021b), we have

$$
\begin{aligned}
\|\mathbf{y} - \mathbf{y}^k(t)\|_2 \; &\le 2(1 + 2\eta K^2 D^2)^2 \|\mathbf{y} - \mathbf{y}(t)\|_2 \\
&\le 2(1 + 2\eta K D)^2 \|\mathbf{y} - \mathbf{y}(t)\|_2
\end{aligned}
\tag{B91}
$$

Applying Lemma 8 and equation B91, we arrive at

$$
C_{11} \le -\frac{2\eta(1+\epsilon)}{C} 4DRK(1 + 2\eta K^2 D)\|\mathbf{y} - \mathbf{y}(t)\|_2^2,
\tag{B92}
$$

with probability at least $1 - D^2 \cdot \exp\left(\frac{-MR_1}{4R'/9R_1 + 2/9}\right)$.

Next, $C_{12}$ is bounded as follows:

$$
C_{12} = -(1+\epsilon)\frac{2\eta}{C} \sum_{i,k,c,j} (y_i - y_i(t))(y_j(t) - y_{c,j}^k(t))G_{ij}(0).
$$

The above can be written in terms of the initial COFL-MAC Gram matrix, $\mathbf{G}(0)$ as follows.

$$
C_{12} = -(1+\epsilon)\frac{2\eta}{C} \sum_{k \in [K]} (\mathbf{y} - \mathbf{y}(t))^T \mathbf{G}(0)(\mathbf{y} - \mathbf{y}^k(t)).
\tag{B93}
$$

Applying equation B91, and assuming $\|\mathbf{G}(0)\|_F = \kappa\lambda$, where $\kappa$ is the condition number and $\lambda$ is the minimum eigenvalue of $\mathbf{G}(0)$, we obtain

$$
C_{12} \le -(1+\epsilon)\frac{4\eta^2\kappa\lambda K^2 D}{C}\|\mathbf{y} - \mathbf{y}(t)\|_2^2.
\tag{B94}
$$

Next, we bound $C_{13}$ as follows.

$$
C_{13} = -(1+\epsilon)\frac{2\eta K}{C} \sum_{i,j}(y_i - y_i(t))(y_j - y_j(t))G_{ij}(0).
$$

The above can be simplified as follows.

$$
C_{13} = -(1+\epsilon)\frac{2\eta K}{C}(\mathbf{y} - \mathbf{y}(t))^T \mathbf{G}(0)(\mathbf{y} - \mathbf{y}(t)).
\tag{B95}
$$

Alternatively,

$$
C_{13} \le (1+\epsilon)\frac{2\eta K\lambda}{C}\|\mathbf{y} - \mathbf{y}(t)\|_2^2.
\tag{B96}
$$

Therefore, from equation B82, equation B92, equation B94, equation B96, we have

$$
\begin{aligned}
C_1 \; &\le -(1+\epsilon)\frac{2\eta}{C}(-4DRK(1 + 2\eta K^2 D)\|\mathbf{y} - \mathbf{y}(t)\|_2^2 + K\lambda\|\mathbf{y} - \mathbf{y}(t)\|_2^2 - 2\eta\kappa\lambda K^2 D\|\mathbf{y} - \mathbf{y}(t)\|_2^2) \\
&\le (1+\epsilon)\frac{2\eta}{C}\|\mathbf{y} - \mathbf{y}(t)\|_2^2(-K\lambda + 4DRK(1 + 2\eta K^2 D) + 2\eta\kappa\lambda K^2 D).
\end{aligned}
$$

Therefore,

$$
C_1 \le (1+\epsilon)\frac{2\eta}{C}\|\mathbf{y} - \mathbf{y}(t)\|_2^2(-K\lambda + 4DRK(1 + 2\eta K^2 D) + 2\eta\kappa\lambda K^2 D)
\tag{B97}
$$

with a probability at least $1 - D^2\exp\left(\frac{-MR_1}{4R'/9R_1 + 2/9}\right)$. Here $R_1 = R - (2/3)R'$. $\qquad\square$

**Proof of Lemma 11**

*Proof.* We have the complementary COFL-MAC Gram matrix $\mathbf{G}^k(t)^\perp \in \mathbb{R}^{D \times D}$ such that $G^k(t)_{ij}^\perp \equiv G_{c,ij}^k(t)^\perp, j \in S_c$. We rewrite the expression for $C_2$ as follows:

$$C_2 = (1+\epsilon)\frac{2\eta}{C} \sum_{i,k,c,j} (y_i - y_i(t))(y_j - y_{c,j}^k(t))G_{c,ij}^k(t)^\perp.$$

The above can be rewritten in the vector notation using COFL-MAC Gram matrix $\mathbf{G}^k(t)^\perp$, the global and local predictions for the $t$-th round given by $\mathbf{y}(t)$ and $\mathbf{y}^k(t)$, respectively, and the ground truth, $\mathbf{y}$, as follows

$$C_2 = (1+\epsilon)2\eta \sum_{k \in [K]} (\mathbf{y} - \mathbf{y}(t))^T \mathbf{G}^k(t)^\perp (\mathbf{y} - \mathbf{y}^k(t)). \tag{B98}$$

Therefore, from equation B91, and using triangle inequality followed by Cauchy-Schwarz inequality, we obtain an upper bound on $C_2$ as follows:

$$C_2 \leq (1+\epsilon)4\eta K(1 + 2\eta DK)\|\mathbf{y} - \mathbf{y}(t)\|_2^2 \|\mathbf{G}^k(t)^\perp\|_F. \tag{B99}$$

From Lemma 9, we have:

$$\|\mathbf{G}^k(t)^\perp\|_F \leq \frac{4DR}{C}$$

with a probability of at least $1 - \exp\left(\frac{-MR_2}{\frac{R'}{8R_2}+1/6}\right)$ over random initialization, where $R_2 = R - R'/4$. Incorporating Lemma 9 into equation B99 gives us the required result. $\square$

**Proof of Lemma 12**

*Proof.* Let us rewrite the expression for $C_3$ as follows:

$$\begin{aligned} C_3 &= -2\mathbf{v}_2^T(\mathbf{y} - \mathbf{y}(t)) \\ &\leq 2|\mathbf{v}_2^T(\mathbf{y} - \mathbf{y}(t))| \\ &\leq 2\|\mathbf{y} - \mathbf{y}(t)\|_2\|\mathbf{v}_2\|_2, \end{aligned} \tag{B100}$$

where the last step was obtained using Cauchy-Schwarz inequality. Let us rewrite the expression for $v_{2,i}$ as follows:

$$v_{2,i} = \frac{1}{\sqrt{M}} \sum_{m \in \overline{\mathcal{Q}}_i} b_m \left(\phi(\mathbf{x}_i^T(\widehat{\mathbf{u}}_m(t) + \Delta\widehat{\mathbf{u}}_m(t))) - \phi(\mathbf{x}_i^T \widehat{\mathbf{u}}_m(t))\right). \tag{B101}$$

We start from the definition of $v_{2,i}$ in equation B101, and upper bound $\|\mathbf{v}_2\|_2$ as follows:

$$\|\mathbf{v}_2\|_2^2 \leq \sum_{i=1}^n \left(\frac{1}{\sqrt{M}} \sum_{m \in \overline{\mathcal{Q}}_i} |\mathbf{x}_i^T \Delta\widehat{\mathbf{u}}_m(t)|\right)^2 \tag{B102}$$

$$= \frac{1}{M} \sum_{i=1}^D \left(\sum_{m=1}^M \mathbf{1}_{m \in \overline{\mathcal{Q}}_i} |\mathbf{x}_i^T \Delta\widehat{\mathbf{u}}_m(t)|\right)^2. \tag{B103}$$

Applying Cauchy Schwarz inequality, we arrive at

$$\|\mathbf{v}_2\|_2^2 \leq \frac{1}{M} \sum_{i=1}^D \left(\sum_{m=1}^M \mathbf{1}_{m \in \overline{\mathcal{Q}}_i} \|\mathbf{x}_i\|_2 \|\Delta\widehat{\mathbf{u}}_m(t)\|_2\right)^2. \tag{B104}$$

Applying Lemma 4, and re-arranging the above, we obtain

$$\|\mathbf{v}_2\|_2^2 \leq \frac{\eta^2}{M}\left((1+\epsilon)\frac{2\eta K(1+2\eta DK)\sqrt{D}}{C\sqrt{M}}\|\mathbf{y} - \mathbf{y}(t)\|_2,\right)^2 \sum_{i=1}^D \left(\sum_{m=1}^M \mathbf{1}_{m \in \overline{\mathcal{Q}}_i}\right)^2. \tag{B105}$$

By claim 3, $\sum_{r=1}^{m} \mathbf{1}_{m \in \overline{\mathcal{Q}}_i} \leq 4MR$ holds with a probability of at least $1 - \exp\left(\frac{-MR_2}{\frac{R'}{8R_2} + 1/6}\right)$, where $R_2 = R - R'/4$, which leads to the following:

$$\|\mathbf{v}_2\|_2^2 \leq \frac{\eta^2}{M}(1+\epsilon)^2 \frac{4K^2(1+2\eta DK)^2 D}{C^2 M} \|\mathbf{y} - \mathbf{y}(t)\|_2^2 D(4MR)^2 \tag{B106}$$

$$\leq \left(\frac{(1+\epsilon)\, 8\eta K(1+2\eta DK)}{C} DR \|\mathbf{y} - \mathbf{y}(t)\|\right)^2. \tag{B107}$$

Therefore, we arrive at the following:

$$\|\mathbf{v}_2\|_2 \leq \frac{(1+\epsilon)\, 8\eta K(1+2\eta DK)DR}{C} \|\mathbf{y} - \mathbf{y}(t)\|. \tag{B108}$$

Using equation B105 in equation B100, we obtain the required result. $\qquad\square$

**Proof of Lemma 13**

*Proof.* Let us rewrite the definition of $C_4$ as follows:

$$C_4 = \|\mathbf{y}(t+1) - \mathbf{y}(t)\|_2^2. \tag{B109}$$

Let us also rewrite the definition of $v1_{i,1}$ as follows:

$$v_{1,i} = \frac{1}{\sqrt{M}} \sum_{m \in \mathcal{Q}_i} b_m \left(\phi\big(\mathbf{x}_i^T(\widehat{\mathbf{u}}_m(t) + \Delta\widehat{\mathbf{u}}_m(t))\big) - \phi\big(\mathbf{x}_i^T \widehat{\mathbf{u}}_m(t)\big)\right), \tag{B110}$$

Further, $\mathbf{y}(t+1) - \mathbf{y}(t) = \mathbf{v}_1 + \mathbf{v}_2$. From equation B110 and equation B101, we have

$$\mathbf{y}(t+1) - \mathbf{y}(t) \leq \sum_{i=1}^{D} \frac{1}{\sqrt{M}} \sum_{m=1}^{M} \mathbf{x}_i^T \Delta\widehat{\mathbf{u}}_m(t). \tag{B111}$$

Taking the square of the $\ell_2$ norm followed by triangle inequality, we obtain

$$\|\mathbf{y}(t+1) - \mathbf{y}(t)\|_2^2 \leq \sum_{i=1}^{D} \left(\frac{1}{\sqrt{M}} \sum_{m=1}^{M} |\mathbf{x}_i^T \Delta\widehat{\mathbf{u}}_m(t)|\right)^2 \tag{B112}$$

$$= \frac{1}{M} \sum_{i=1}^{D} \left(\sum_{m=1}^{M} |\mathbf{x}_i^T \Delta\widehat{\mathbf{u}}_m(t)|\right)^2. \tag{B113}$$

From Lemma 6 and Cauchy-Schwarz, we have the required bound on the term $C_4$.

$\qquad\square$

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
