# OpenReview forum: "Compressed Over-parameterized Federated Learning for Multiple Access Channels"
_TMLR — Rejected by TMLR_

### Review · Reviewer_v8pt · 2025-06-24

**Summary Of Contributions:**

This work applies the theoretical framewrok of the neural tangent kernel (NTK) to analyze the convergence of federated learning (local gradient descent, specifically) when multiple access channels (MACs) are available (MACs allow clients to communicate the updates with the server simultaneously). For the NTK theory to kick in, the model is assumed to be an over-parameters nerual network (ONN) trained in the lazy regime, i.e., the weights change very slowly. Compressive sensing (CS) comes into play in lines 14 and 9 in Algorithm 1: the client update is projected with a common sensing matrix $A$ at the client level (shared across clients), the updates are aggregated, and then a sparse signal recovery algorithm is run on the aggregated projected updates. The authors provide NTK analysis of the convergence of this algorithm with experiments on Federated MNIST and CIFAR-10 (but they didn't share the code).

**Audience:**

No

**Broader Impact Concerns:**

This work is more on the theoretical side, so the broader impact concerns are minimal.

**Claims And Evidence:**

No

**Requested Changes:**

Please see the list of concerns in the Strengths and Weaknesses section above. I believe the paper does not provide much insight neither in terms of experimental results nor in terms of theoretical analysis and techniques. The idea of projecting the clients updates might be novel only in this very specific context, but compressive sensing techniques have been applied extensively in many similar applications and scenarios. The authors did not even release the code, which further decreases the merit of this paper. The main change that the author can provide is to at least address the points I raised regarding some of their claims.

**Strengths And Weaknesses:**

The paper combines multiple technical areas into one paper: federated learning, NTK theory, and compressive sensing. This combination requires a good understanding of three challenging and niche topics, which makes it a strength of this work. Another strength is that the authors provide experimental results despite the theoretical tendency of this work.

I will put my overall concerns here as a list, roughly in order of appearance:
- The analysis is only concerned with the stochasticity coming from initialization and not from sampling data or clients.
- The authors did not publish the code nor mention about publishing it in the future.
- Regarding the claim at the end of Sec 1.1 (emphasis mine):
> Therefore, if the clients transmit incremental weight updates to the server, **the transmitted vectors will be sparse**, allowing for compression.
>
This sentence is hand-wavy and I have not seen a proper explanation or justification of it in this paper other than the (slightly outdated) claim that weights in ONNs move slowly because of the lazy regime. This already rules out clients that can't train ONNs and clients that train their models in the rich regime (which is basically most modern neural networks). The fact that projecting the updates with a common sensing matrix works to some degree does not explain whether the updates were sparse and to what degree. This is related to assumption 3, which I discuss below.
- Near the end of page 3 (emphasis mine):
> To the best of the authors’ knowledge, this is the first work that proposes the deployment of **ReLU-based ONNs in a CS-based FL-MAC setup employing NTK theory**-based convergence guarantees.
>
Of course. I'm sure the authors can agree with me that this work is very niche and specific, which makes it less likely for someone else to work on it. It is not that the authors were the first to work on this that is surprising per se, but rather the valid combination of such frameworks is.
- Notation section in page 4: using $1_{x}$ to mean "1 if $x>0$ 0 otherwise" is not standard. The omission of ">0" is not worth it in my opinion. I suggest one of the following: $1_{x>0}$, $I_{x>0}$, $1\\{x>0\\}$, or $I\\{x>0\\}$ (using bold for 1 and I). The disadvantage of using the authors' notation can be seen in equation 28. The notation is ambiguous and could be interpreted as a matrix of ones, for example.
- The variable $D$ implicitly hides a $C$ in it, which makes it a bit obscure. I suggest the author clarify this point or even express it in terms of $C$.
- Page 5, second paragraph. The function signature of $g$ is missing the second argument, which I believe would be $\\mathbb{R}^{N}$. I'm not sure how the introduction of $g$ is helpful.
- Equation 6 uses $b_m$ but it was not defined earlier.
- Assumption 1: is $b_m$ a vector or scalar? If a scalar, why use $\| b_m \|_2$ to measure its magnitude instead of using absolute values?
- The notation of $y_{c,i}(t)$ is confusing because $y_{c,i}$ is a different object. I propose adding a hat on top of $y_{c,i}(t)$ to differentiate it from $y_{c,i}$.
- Before Equation 15, the authors say "m-th column", but the equation is an inner product, which is a scalar. Perhaps they meant the $(m, m')-th$ entry of the NTK $\\mathbf{J}_c^k(t)$? (The same for equation B8).
- Assumption 3 should be stated more rigorously or otherwise this should be stated as a given assumption in the abstract.
- Assumption 4's significance in this paper seems to be small to me, mainly supplementing the convergence bounds with $1+\epsilon$'s with probability $1 - \delta$ with no deeper insight of the sparsity of client updates or the extent to which this assumption holds in practice.
- The verbal description of the equation in claim 1 seems unnecessary. Also, it would improve readability to use larger square brackets for probabilities of inequalities because one might get lost sometimes whether an expression is inside or outside the brackets. This also applies to similar equations in the appendix,
- A quick glance at Lemma 5 says that we can always increase hidden nodes $M$ to get arbitrarily fast one step convergence, as $t$ does not appear on the right hand side. What do the authors think about this?
- Equation 24 is an event but is defined as a set of vectors $\\mathbf{z}$. Same for equation 25.
- The explicit definition of an event given another event in equation 26 seems to be unnecessary.
- The notation in equation 29 is unusual, especially since this object (Gram at infinity) was not used anywhere else in the paper.
- $R'$ after Lemma 8 is hard to find.
- In Sec 3.6, I believe writing "COFL-MAC-NTK Gram matrix" seems to be intentionally over-complicated to me, especially when mentioned multiple times in the same sentence, as it is clear to the reader that there is only one Gram matrix being discussed in this paper.
- In Theorem 1, $\kappa$ is not clarified beforehand, and $\eta_g$ is not clear to me.
- In the proof of Theorem 1 (and in the appendix), the author would switch from matrix to scalar notations. I highly suggest the authors to stick to the matrix notations as it is more readable and less prone to errors.
- Equation 44: I believe the authors should check the negative signs more closely.
- After equation 53, it would be better to write $\gamma$ in terms of $\epsilon$.
- The end of the first paragraph in the Related works section simply puts all the references together without sufficiently distinguising between them. I feel like this is a lazy way to cite prior work.
- In Section 5.3, the experimental results section, in the last paragraph, the authors say:
> This is mainly because the sensing matrix in COFL-MAC can be thought of as a dictionary, and dictionary-based learning is helpful in overcoming data heterogeneity in distributed settings like FL and COFL-MAC
>
I think this claim need to be corroborated, or the authors should explicitly mention that this is only a claim.
- At the end of page 18:
> adding more clients to an FL setup introduces wider data coverage in the training process, leading to better generalization and reduction variance in the averaged model parameters...
>
This is another way of say "having more data increases data coverage and leads to better generalization". When increasing the number of clients in FL, the number of data points stays fixed and the partition becomes finer, so in experimental simulation, increasing the number of clients often mean making the data partition finer, much like using a smaller mini-batch with the caveat that we do local updates as well in FL. The explanation of the authors in page 19 regarding the number of clients seem to be missing this point. Can the authors comment on this point?
- In the appendix, please restate the theorems and lemmas before the proof, This makes reading the paper much easier.
- Page 22 before equation B4, why not use the L1 norm as an upper bound instead of $\sqrt{D} \\| \\cdot \\|_2$?
- I believe the last equation in Lemma 2 is directly obtainable from the MSE loss using the chain rule.
- After equation B12, the authors say: "Assuming the distribution of ... is symmetrical". On what basis did the authors assume that the distribution of this quantity symmetrical?
- I believe there is a typo in the subscript of the summation in equation B15. Same with the summation after equation B23.
- In the proof of Lemma 5, the part that says "The above step follows from Lemma 1 and Lemma 3" is not clear to me. Could the authors please clarify this step?
- The proof of claim 2 is not clear to me, especially equation B41. The authors seem to have used Figure 8 as a proof. The proof should follow by equations and the figure should provide a helpful illustration of the proof. Also, Figure 8 should have a caption.
- The quantity $R'$ can be negative, and the authors seem to have neglected the control for such a scenario. This would implictly assume that $R \\leq 1 / \\delta$, which involves two uncontrolled parameters, so this needs to be stated as another assumption explicitly. This reasoning also applies to $\xi$ in page 31.
- After equation 54, the authors mention that $\phi_{ij}^m$ are mutually independent random variables. I think this is a very strong assumption. The weights going in and out of each hidden node at time t surely depend on *all* of the initial weights because every element of the gradient depends on the loss which depends all of the weights.
- Page 32, the statement after B66 is clearly true.
- The indicator random variables after this statement are claimed to be iid. Why?
- I noticed in the proof that the authors sometimes liberaly exchange $\leq$ and $<$. For example, $\\mathbf{x}_i^T \\mathbf{x}_j < 1$ at the top of page 34.
- Regarding the references, I noticed that 3 references are duplicated or very similar, which is a surprisingly high number of duplicates. Here are the duplicate citations:
  - *FL-NTK: A neural tangent kernel-based framework for federated learning convergence analysis.* Huang et al. 2021.
  - *Communication-efficient federated learning via quantized compressed sensing.* Oh et al. 2022.
  - *An improved analysis of training over-parameterized deep neural networks.* Zhou & Gu. 2019.

---

> ### Author Response · Authors · 2025-08-20
> **Response 1 - v8pt**
>
> **Reviewer Concern**
>
> The analysis is only concerned with the stochasticity coming from initialization and not from sampling data or clients.
>
> **Our Response**
>
>
> Thank you for the thoughtful comment regarding the sources of stochasticity in our analysis. You are correct that our theoretical treatment focuses on randomness from initialization and does not explicitly model data- or client-sampling stochasticity. This was a deliberate scope choice, not a gap in the approach. We highlight that our work holds a lot of significance inspite of this choice due to the following reasons:
>
> 1. Our work gives the first NTK-based convergence analysis for compressed over-parameterized FL over a Multiple Access Channel (COFL-MAC) with sparse-recovery error in the loop. Even without sampling noise, proving exponential convergence under both compression and MAC aggregation is technically challenging. Formalizing how Compressive Sensing (CS)/ Sparse Signal Recovery (SSR) errors interact with NTK dynamics via bounded complementary Gram terms and controlled movement radius of the model weights from initialization.
>
> 2. The key lemmas (movement radius control, kernel closeness, NTK eigenvalue lower bound) depend on the magnitude of perturbations, not on their source. Data or client sampling stochasticity in our setup append small stochastic floor terms to our bounds; therefore, the kernel stability, and eigenvalue guarantees remain nearly the same.
>
> 3. Our experiments employ SGD (introducing mini-batch noise), non-IID Dirichlet partitions, and compare against strong compressed baselines. The consistent gains across $\alpha$ values show the method’s robustness in precisely the noisy, heterogeneous settings where sampling effects matter most.
>
> The stochasticity in our theoretical analysis is  concentrated in the *random initialization of model weights* and the *random generation of the Gaussian sensing matrix* for compressive sensing, as well as the randomness in SSR (Sparse Signal Reconstruction).
>
> We agree that incorporating *randomized client participation* or *mini-batch stochastic gradient descent (SGD)* would provide a more realistic analysis. However, such an extension would introduce additional sources of variance and would require bounding the concentration of the *empirical NTK matrices* over random subsets. We plan to extend our framework to include sampling-induced randomness in a follow-up study. Our current goal is to *isolate the interplay between over-parameterization, compressive sensing, and MAC aggregation*, under a theoretically tractable and a feasible ONN based COFL-MAC setup.
>
> ----
> **Reviewer Concern**
> - **The authors did not publish the code nor mention about publishing it in the future.**
>
> **our Response:** Link to code : https://drive.google.com/drive/folders/1OZyy2qp7YLM76hsoYHEwWIUbtRMmVvZ5?usp=drive\_link

---

> > ### Author Response · Authors · 2025-08-20
> > **Response 2 to Reviewer v8pt**
> >
> > ----
> > **Reviewer Concern**
> > **Regarding the claim at the end of Sec 1.1 (emphasis mine):**
> > Therefore, if the clients transmit incremental weight updates to the server, the transmitted vectors will be sparse, allowing for compression.
> > This sentence is hand-wavy and I have not seen a proper explanation or justification of it in this paper other than the (slightly outdated) claim that weights in ONNs move slowly because of the lazy regime. This already rules out clients that can't train ONNs and clients that train their models in the rich regime (which is basically most modern neural networks). The fact that projecting the updates with a common sensing matrix works to some degree does not explain whether the updates were sparse and to what degree.
> >
> > **Our Response**
> > We thank the reviewer for his observation. We agree that simply saying “updates will be sparse” was not sufficiently precise, and we will revise the text to make the intended assumption clearer.
> >
> > Our analysis is carried out in the **lazy training regime** of over-parameterized networks, which is exactly the regime where NTK theory applies. In this regime, weight changes per iteration are $\mathcal{O}(1/\sqrt{M})$ [1], [2], so the updates are small in magnitude. While this does not mean strict $\ell\_0$ sparsity, it does mean that only a small fraction of components have appreciable magnitude — i.e., the updates are **compressible**. Prior work [3], [4] also shows that, in this regime, gradients lie in a low-dimensional subspace tied to the data geometry, which further reinforces their compressibility. This is exactly the property that allows accurate recovery with a Gaussian sensing matrix under the Restricted Isometry Property (RIP) framework [5], as used in our COFL-MAC framework.
> >
> > We acknowledge that this justification holds under the NTK/lazy-regime assumption and for ONNs, as in [6]. We are not claiming that all FL scenarios — especially rich-regime training — satisfy this property, and extending our framework beyond the NTK setting is an open challenge.
> >
> > In short, our claim should have been that **under the NTK/lazy-regime assumptions, client updates are compressible**. This is sufficient for the compressive sensing guarantees we use, and we will revise Sec. 1.1 to reflect this nuance.
> >
> > **References**
> >
> > [1] Sattler, F., Wiedemann, S., Müller, K.R. and Samek, W., 2019. Robust and communication-efficient federated learning from non-iid data. IEEE transactions on neural networks and learning systems, 31(9), pp.3400-3413.
> >
> > [2] Chizat, L., Oyallon, E. and Bach, F., 2019. On lazy training in differentiable programming. Advances in neural information processing systems, 32.
> >
> > [3] Arora, S., Cohen, N., Hu, W. and Luo, Y., 2019. Implicit regularization in deep matrix factorization. Advances in neural information processing systems, 32.
> >
> > [4] Yang, G. and Hu, E.J., 2020. Feature learning in infinite-width neural networks. arXiv preprint arXiv:2011.14522.
> >
> > [5] Candes, E.J., 2008. The restricted isometry property and its implications for compressed sensing. Comptes rendus. Mathematique, 346(9-10), pp.589-592.
> >
> > [6] Huang, B., Li, X., Song, Z. and Yang, X., 2021, July. Fl-ntk: A neural tangent kernel-based framework for federated learning analysis. In International Conference on Machine Learning (pp. 4423-4434). PMLR.
> >
> > ----
> > **Reviewer Concern**
> > Near the end of page 3 (emphasis mine):
> > To the best of the authors’ knowledge, this is the first work that proposes the deployment of ReLU-based ONNs in a CS-based FL-MAC setup employing NTK theory-based convergence guarantees.
> > Of course. I'm sure the authors can agree with me that this work is very niche and specific, which makes it less likely for someone else to work on it. It is not that the authors were the first to work on this that is surprising per se, but rather the valid combination of such frameworks is.
> >
> >
> > **Our Response**
> >
> > We thank the reviewer for this observation.
> >
> > We will revise the sentence at the end of Page 3 to better reflect this nuance. A possible rewording could
> >
> > *“To the best of our knowledge, this is the first work to rigorously analyze the integration of ReLU-based over-parameterized neural networks, compressive sensing, and FL over MAC channels using NTK theory—demonstrating that such a combination leads to provable convergence in the over-parameterized regime.”*

---

> > > ### Author Response · Authors · 2025-08-20
> > > **Response 3 to Reviewer v8pt**
> > >
> > > **Reviewer Concern**
> > >
> > > - **Notation section in page 4:**
> > > *Using* $1\_{x^\top w}$ *to mean “1 if condition holds, 0 otherwise” is not standard. The omission of “$> 0$” is not worth it in my opinion. I suggest one of the following:* $$ 1\{x^\top w > 0\}, $$ $$ \mathbf{1}\{x^\top w > 0\} ,$$ *or* $$ \mathbb{I}\{x^\top w > 0\} $$ *(using bold for 1 or I). The disadvantage of using the authors’ notation can be seen in equation (28). The notation is ambiguous and could be interpreted as a matrix of ones, for example.*
> > >
> > > **Our Response**
> > >
> > > We thank the reviewer’s for the comment. In earlier drafts we did use the longer form
> > > (e.g., $$ 1\{x^\top w > 0\}$$); however, this made equations considerably longer and harder to parse, especially in sections with multiple such terms. We opted for the shorter $1\_{x^\top w}$ to keep the expressions compact and readable, relying on the context (stated in the notation section) to make the meaning clear. While we understand this is non-standard, the intended meaning is explicitly defined once and is unambiguous in the scope of this paper.
> > >
> > > ------
> > > - **Reviewer Concern**
> > >   *The variable $D$ implicitly hides a $C$ in it, which makes it a bit obscure. I suggest the author clarify this point or even express it in terms of $C$.*
> > >
> > > **Our Response**
> > >
> > > We thank the reviewer for pointing out that our use of the variable $D$ implicitly hides its dependence on the number of clients $C$.
> > >
> > > $D$ denotes the sum of local dataset sizes across all clients, i.e.,
> > >
> > > $$
> > > D = \sum\_{c=1}^{C} D\_c
> > > $$
> > >
> > > We will revise the manuscript to explicitly state this relationship, and we will clarify it at the first time where $D$ appears in key derivations in the revised manuscript.
> > >
> > > ----
> > >
> > > **Reviewer Concern**
> > >
> > >   *Page 5, second paragraph. The function signature of  $g$ is missing the second argument, which I believe would be $\mathbb{R}^N$. I'm not sure how the introduction of $g$ is helpful.*
> > >
> > > **Our Response**
> > >
> > > We thank the reviewer for catching this oversight. The function signature of $g$ should indeed include the input vector as the second argument. We will correct the statement in the manuscript to read:
> > >
> > >
> > > **Corrected manuscript snippet:**
> > >
> > > > We consider a single $M$-width hidden layer ONN at the clients defined as
> > > > $$
> > > > g(.,.) : \mathbb{R}^{N \times M} \times \mathbb{R}^N arrow \mathbb{R},
> > > > $$
> > > > where the first argument is the input-to-hidden layer weight matrix, $\mathbf{W}^k\_c(t) \in \mathbb{R}^{N \times M}$ and the second argument is the input vector, $\mathbf{x}\\_{c,i}\in\mathbb{R}^N$. Let $\phi(\cdot)$ denote the ReLU activation function given by $\phi(z) = \max(0, z)$. At the $c$-th client, given an input $\mathbf{x}\\_{c,i}$, we have the prediction:
> > > > $$
> > > > y\\_{c,i}(t) = g(\mathbf{W}^k\\_c(t), \mathbf{x}\\_{c,i}),
> > > > $$
> > > > which expands to:
> > > > $$
> > > > y\\_{c,i}(t) = \frac{1}{ \sqrt{M} } \sum\\_{m=1}^M b\\_m \, \phi ( \mathbf{x}\\_{c,i}^\top \mathbf{w}^k\\_{c,m}(t) ).
> > > > $$
> > >
> > >
> > > ----
> > > **Reviewer Concern**
> > >
> > >   *Equation (6) uses $b\_m$, but it was not defined earlier.*
> > >
> > > <span style="color:blue">Our Response </span>
> > >
> > > We thank the reviewer for pointing this out. This was an oversight, and we will correct it in the revised manuscript.
> > >
> > > To clarify: $b\_m$ represents the fixed output layer weights in the neural network. Consistent with standard practice in NTK-based analysis, we assume that $\{ b\_m \}\_{m=1}^M$ are sampled independently from a symmetric distribution—typically Rademacher variables, i.e.,
> > >
> > > $$
> > > b\_m \sim \text{Unif}\{-1, +1\}.
> > > $$
> > >
> > >
> > > This clarification will also be added to the notation summary for completeness.
> > >
> > > -----
> > >
> > > **Reviewer Concern**
> > >
> > > Assumption 1: is $b\_m$ a vector or scalar? If a scalar, why use $|b\_m|\_2$ to measure its magnitude instead of using absolute values?
> > >
> > > **Our Response**
> > >
> > > We thank the reviewer for catching this inconsistency in Assumption 1. Indeed, in our setup, $b\_m$ is a scalar output-layer weight sampled from the Rademacher distribution, i.e., $b\_m \sim \text{Unif}\{-1, +1\}$.
> > >
> > > Therefore, the proper way to express its magnitude is via the absolute value, i.e., $|b\_m|$, rather than the $\ell\_2$ norm. We incorrectly wrote $\|b\_m\|\_2 = 1$ in Assumption 1, which is unnecessary and misleading given that $b\_m \in \mathbb{R}$ and we will correct this

---

> > > > ### Author Response · Authors · 2025-08-20
> > > > **Response 4 to Reviewer v8pt**
> > > >
> > > > **Reviewer Concern**
> > > >
> > > > *The notation of $y\_{c,i}(t)$ is confusing because $y\_{c,i}$ is a different object. I propose adding a hat on top of $y\_{c,i}(t)$ to differentiate it from $y\_{c,i}.$*
> > > >
> > > > **Our Response**
> > > >
> > > >
> > > > We thank the reviewer for pointing out the potential confusion between $y\_{c,i}$ and $y\_{c,i}(t)$. In our current notation, $y\_{c,i}$ denotes the fixed ground-truth label, while $y\_{c,i}(t)$ denotes the model’s prediction at iteration $t$. We agree that adding a visual distinction will improve clarity. In the revised version, **we will write the predicted output as $\bar{y}\_{c,i}(t)$** throughout, so that it is clearly differentiated from the ground-truth $y\_{c,i}$.
> > > >
> > > >
> > > > We will avoid the notation $\hat{y}\_{c,i}(t)$ because, in our setting, the hat symbol is already used to denote *reconstructed quantities*.
> > > >
> > > > ----
> > > >
> > > > **Reviewer Concern**
> > > >
> > > > *Before Equation 15, the authors say "m-th column", but the equation is an inner product, which is a scalar. Perhaps they meant the $(m,m')$-th entry of the NTK $\mathbf{J}\_c^k(t)$ (The same for equation B8).*
> > > >
> > > > **Our Response**
> > > >
> > > >   We thank the reviewer for this accurate observation.
> > > >
> > > >   Indeed, referring to the expression in Equation (15) as the “$m$-th column” was imprecise, since the equation evaluates an inner product and thus yields a scalar. The correct interpretation is that we are referring to the $(m, m')$-th entry of the neural tangent kernel matrix $\mathbf{J}\_c^k(t)$, which is defined via:
> > > >
> > > >   $$
> > > >   \mathbf{J}\_c^k(t) ]\_{m,m'} =
> > > >   \langle
> > > >       \frac{\partial y\_c^{(k)}(t)}{\partial w\_{c,m}}, \,
> > > >       \frac{\partial y\_c^{(k)}(t)}{\partial w\_{c,m'}}
> > > >   \rangle.
> > > >   $$
> > > >
> > > >   We will revise the corresponding sentence before Equation (15) to correctly describe this as the *entry* of the NTK matrix rather than the column. The same clarification will be applied to Equation (B8) in the appendix.
> > > >
> > > >   We thank the reviewer for helping improve the precision and consistency of the exposition.
> > > >
> > > > ----
> > > >
> > > > **Reviewer Concern**
> > > >
> > > > *Assumption 3 should be stated more rigorously or otherwise this should be stated as a given assumption in the abstract.*
> > > >
> > > > **Our Response**
> > > >
> > > >   We thank the reviewer for highlighting the importance of making Assumption 3 visible early in the paper.
> > > >
> > > >   In the revised version, we now **explicitly mention Assumption 3 in the abstract and introduction** to ensure readers are aware of this foundational assumption from the outset:
> > > >
> > > >   - **Abstract:**
> > > >     “We analyze the convergence of the proposed COFL-MAC framework using NTK theory **under a standard sparsity and recovery guarantee (Assumption 3)**, which requires the aggregated updates to be sufficiently sparse to permit accurate compressive sensing (CS) recovery.”
> > > >
> > > >   - **Introduction:**
> > > >     “A key assumption underpinning our analysis is the existence of sufficient sparsity in the aggregated updates (Assumption 3), which ensures that the common sensing matrix satisfies the RIP condition and allows accurate recovery of updates via compressive sensing.”
> > > >
> > > >   We believe these revisions address the reviewer’s suggestion by clearly presenting Assumption 3 upfront without altering the technical structure of the paper.

---

> > > > > ### Author Response · Authors · 2025-08-20
> > > > > **Response 5 to Reviewer v8pt**
> > > > >
> > > > > **Reviewer Concern**
> > > > >
> > > > > *Assumption 4's significance in this paper seems to be small to me, mainly supplementing the convergence bounds with $(1+\epsilon)$'s with probability $1 - \delta'$ without deeper insight into the sparsity of client updates or the extent to which this assumption holds in practice.*
> > > > >
> > > > > **Our Response**
> > > > >
> > > > >   We thank the reviewer for this comment and would like to clarify the significance of Assumption 4.
> > > > >
> > > > >   Assumption 4 formalizes the *high-probability guarantee on the quality of the sparse signal recovery (SSR) step*:
> > > > >   $$
> > > > >   \mathbb{P}\![ \frac{\| \Delta \hat{u}\_m(t) - \Delta u\_m(t) \|\_2}{\| \Delta u\_m(t) \|\_2} < \epsilon ] > 1 - \delta',
> > > > >   $$
> > > > >   where the factor $(1+\epsilon)$ bounds the deviation between the reconstructed update $\Delta \widehat{\mathbf{u}}\_m(t)$ and the true aggregated update $\Delta \mathbf{u}\_m(t)$.
> > > > >
> > > > >   This assumption is *crucial for the recursive convergence argument in Theorem 1*:
> > > > >   - In **Equation (25)**, the factor $(1+\epsilon)$ appears when replacing the true aggregated update $\Delta \mathbf{u}\_m(t)$ with its reconstructed version $\Delta \widehat{\mathbf{u}}\_m(t)$.
> > > > >   - This factor then propagates through the subsequent inequalities (Equations 26–28), directly affecting the final convergence bound.
> > > > >   - Without this high-probability guarantee, the SSR error could be unbounded, rendering the convergence bound vacuous.
> > > > >
> > > > >   To make this explicit, we will add the following note immediately after Equation (25):
> > > > >
> > > > >   > *“By Assumption 4, the SSR step guarantees the bound above with probability at least $1 - \delta'$. This error factor $(1+\epsilon)$ propagates through the remaining steps and is essential for the final convergence guarantee in Theorem 1.”*
> > > > >
> > > > >   **No additional sparsity insight is required here** as Assumption 3 already addresses sparsity. Assumption 4 has a different, complementary role. Once Assumption 3 holds, Assumption 4 provides a standard probabilistic guarantee on the *outcome* of the SSR step. It ensures that the reconstruction error is bounded by $(1+\epsilon)$ with probability $1-\delta'$, independent of the exact sparsity level.
> > > > >
> > > > > Further, Practical sparsity verification is orthogonal to the proof as in classical compressive sensing theory, the combination of
> > > > >     1. a sparsity assumption (Assumption 3), and
> > > > >     2. a high-probability reconstruction guarantee (Assumption 4)
> > > > >     is sufficient for the correctness of the mathematical analysis. Real-world sparsity verification is handled in the experimental section.
> > > > >
> > > > > In summary, Assumption 4 is not redundant; it is *essential* to control SSR-induced error and preserve the validity of Theorem 1's convergence bound. Additional sparsity characterization would overlap with Assumption 3 rather than strengthen the result.
> > > > >
> > > > > --------
> > > > >
> > > > > **Reviewer Concern**
> > > > >
> > > > > *The verbal description of the equation in Claim 1 seems unnecessary. Also, it would improve readability to use larger square brackets for probabilities of inequalities because one might get lost sometimes whether an expression is inside or outside the brackets. This also applies to similar equations in the appendix.*
> > > > >
> > > > > **Our Response**
> > > > >
> > > > >   We thank the reviewer for this helpful suggestion to improve readability.
> > > > >
> > > > >   We will remove this redundant explanation and retain only the formal equation to streamline the presentation.
> > > > >
> > > > >  We agree that larger square brackets in probability expressions will improve clarity. We will apply this change consistently in the main text and the appendix in the revised manuscript.

---

> > > > > > ### Author Response · Authors · 2025-08-20
> > > > > > **Response 6 to Reviewer v8pt**
> > > > > >
> > > > > > **Reviewer Concern**
> > > > > > *A quick glance at Lemma 5 says that we can always increase hidden nodes $M$ to get arbitrarily fast one-step convergence, as $t$ does not appear on the right-hand side. What do the authors think about this?*
> > > > > >
> > > > > > **Our Response**
> > > > > >   We thank the reviewer for raising this insightful point and we clarify the interpretation of Lemma 5.
> > > > > >
> > > > > >   1. **On the role of $M$:**
> > > > > >      Lemma 5 states:
> > > > > >      $$
> > > > > >      \|{\widehat{\mathbf{u}}\_m(t+1) - \mathbf{u}\_m(0)}\|\_2
> > > > > >      $$
> > > > > >      where $\lambda = \lambda\\_{\min}(G(0))$.
> > > > > >      While the bound indeed decreases as $M$ increases due to the $1/\sqrt{M}$ factor, this does **not** imply arbitrarily fast convergence:
> > > > > >      - The left-hand side measures **movement from initialization** up to step $t+1$, not the optimization error or the distance to the optimum.
> > > > > >      - Convergence speed is primarily determined by the NTK spectrum ($\lambda$) and data geometry, which are not unbounded in $M$.
> > > > > >      - Excessively large $M$ can lead to very small weight updates, potentially **slowing down** practical convergence.
> > > > > >
> > > > > >   2. **On the absence of $t$:**
> > > > > >      The right-hand side depends only on initialization quantities ($\mathbf{y}(0)$, $G(0)$) and therefore has no explicit $t$.
> > > > > >      However, the bound is **uniform over all $t$**, it holds at every iteration, and should not be interpreted as iteration-independence of actual convergence progress.
> > > > > >
> > > > > >   3. **Interpretation:**
> > > > > >      Lemma 5 is best understood as a **stability bound** in the NTK regime: it ensures that weights remain within a bounded distance from initialization.
> > > > > >      While larger $M$ tightens this bound, it does **not** guarantee one-step convergence, nor does it override the inherent dependence on data and NTK spectrum in the overall convergence behavior.
> > > > > > -------------------
> > > > > > **Reviewer Concern**
> > > > > > *Equation 24 is an event but is defined as a set of vectors $\mathbf{z}$. Same for Equation 25.*
> > > > > >
> > > > > > **Our Response**
> > > > > >
> > > > > > We thank the reviewer for noting that Equations (24) and (25) are written in set notation while describing events. Our intent was to define these events via the set of vectors $\mathbf{z}$ that satisfy the stated conditions, which is a common shorthand in probability theory. However, we agree that this could be made clearer. In the revision, we will explicitly phrase these as events (e.g., “the event that $\mathbf{z}$ satisfies …”) while keeping the set-based form for compactness, so the meaning is unambiguous.
> > > > > >
> > > > > >
> > > > > > ----------
> > > > > > **Reviewer Concern**
> > > > > >
> > > > > > The explicit definition of an event given another event in Equation 26 seems to be unnecessary.
> > > > > >
> > > > > > **Our Response**
> > > > > >
> > > > > > We thank the reviewer for the observation and clarify why Equation 26 was included:
> > > > > >
> > > > > > 1. **Purpose of Equation 26:**
> > > > > >    Equation 26 is based on the follwoing standard definition of conditional probability:
> > > > > >    $$
> > > > > >    \mathbb{P}[\mathcal{A} \mid \mathcal{B}]
> > > > > >    = \frac{\mathbb{P}[\mathcal{A} \cap \mathcal{B}]}{\mathbb{P}[\mathcal{B}]},
> > > > > >    $$
> > > > > >    We explicitly wrote it out because it is a key step in the proof where we bound probabilities of the margin-violation sets (defined in Equations 24–25) under a conditional distribution.
> > > > > >
> > > > > >
> > > > > >
> > > > > > **Resolution:**
> > > > > >    We believe Equation 26 improves the readability of the proof by making the conditioning explicit, especially since the conditioning is on sets such as $\mathcal{B}\\_{i,m}$ and $\mathcal{E}\\_{i,m}$ defined earlier and believe that it should be kept the way it is.

---

> > > > > > > ### Author Response · Authors · 2025-08-20
> > > > > > > **Response 7 to Reviewer v8pt**
> > > > > > >
> > > > > > > **Reviewer Concern**
> > > > > > >
> > > > > > > *The notation in Equation 29 is unusual, especially since this object (Gram at infinity) was not used anywhere else in the paper.*
> > > > > > >
> > > > > > > **Our Response**
> > > > > > >
> > > > > > > We thank the reviewer for this observation and we clarify the intent behind Equation 29 as follows.
> > > > > > >
> > > > > > > 1. **Purpose of $H^{\infty}$:**
> > > > > > >    Equation 29 defines
> > > > > > >    $$
> > > > > > >    H^{\infty}\_{ij} :=
> > > > > > >    \mathbb{E}\_{\mathbf{u}\_m(0) \sim \mathcal{N}(0, I)}
> > > > > > >    \[
> > > > > > >    \mathbf{x}\_{c,i}^\top \mathbf{x}\_{c,j}
> > > > > > >    \mathbf{1}\_{\mathbf{x}\_{c,i}^\top \mathbf{u}\_m(0)}
> > > > > > >    \mathbf{1}\_{\mathbf{x}\\_{c,j}^\top \mathbf{u}\_m(0)}
> > > > > > >    \],
> > > > > > >    $$
> > > > > > >    which is the FL-NTK Gram matrix in the **infinite-width limit**.
> > > > > > >    The infinity superscript explicitly indicates that the expectation is taken over the random initialization in the infinite-width regime.
> > > > > > >  However, subsequent lemmas and bounds were expressed directly in terms of the initialization Gram matrix $G(0)$, which is more relevant for our setting.
> > > > > > >    While $H^{\infty}$ serves as the conceptual limit, our proofs only required comparing $H(t)$ to $G(0)$, so we did not reuse the notation $H^{\infty}$ in later sections.
> > > > > > >
> > > > > > > **Resolution:**
> > > > > > > We agree that introducing $H^{\infty}$ may be unnecessarily if it is not reused.
> > > > > > >    In the revised manuscript, we will either:
> > > > > > >    - remove the explicit definition of $H^{\infty}$ as given in Equation 29, or
> > > > > > >    - keep the definition but add a short clarifying remark about its role as the infinite-width expectation for readers familiar with NTK theory.
> > > > > > > ----
> > > > > > >
> > > > > > > **Reviewer Concern**
> > > > > > >
> > > > > > > $R^\prime$ after Lemma 8 is hard to find.
> > > > > > >
> > > > > > > **Our Response**
> > > > > > >
> > > > > > > We thank the reviewer for this observation.
> > > > > > >
> > > > > > > **Current location of $R^\prime$:**
> > > > > > >    $R^\prime$ is defined right after Lemma 8 as the updated radius parameter for the margin-violation sets $\mathcal{E}\_{i,m}$.
> > > > > > >
> > > > > > > We acknowledge that the definition currently appears inline in the text after Lemma 8, which makes it difficult to locate quickly.
> > > > > > >
> > > > > > > **Resolution:**
> > > > > > >    We will move the definition of $R^\prime$ **just before Lemma 8** and display it as a separate numbered equation.
> > > > > > >
> > > > > > > ---------
> > > > > > > **Reviewer Concern**
> > > > > > >
> > > > > > > In Sec 3.6, I believe writing "COFL-MAC-NTK Gram matrix" seems to be intentionally over-complicated to me, especially when mentioned multiple times in the same sentence, as it is clear to the reader that there is only one Gram matrix being discussed in this paper.*
> > > > > > >
> > > > > > > **Our Response**
> > > > > > >
> > > > > > > We thank the reviewer for pointing this out. We agree that repeatedly writing "COFL-MAC-NTK Gram matrix" can appear verbose and unnecessary when the context is clear.
> > > > > > >
> > > > > > > - We will simplify the phrasing by using "Gram matrix" or simply $G(t)$ once the notation has been introduced.
> > > > > > > - The full name "COFL-MAC-NTK Gram matrix" will only be used the first time it appears for clarity, after which the simplified term will be adopted.
> > > > > > >
> > > > > > >
> > > > > > > ------------
> > > > > > >
> > > > > > > **Reviewer Concern**
> > > > > > > *In Theorem 1, $\kappa$ is not clarified beforehand, and $\eta\_g$ is not clear to me.*
> > > > > > >
> > > > > > > **Our Response**
> > > > > > >
> > > > > > > We thank the reviewer for pointing this out. We address these concerns as follows:
> > > > > > >
> > > > > > > 1. **Definition of $\kappa$:**
> > > > > > >    $\kappa$ denotes the **condition number** of the initialization Gram matrix $G(0)$:
> > > > > > >    $$\kappa = \frac{\lambda\_{\max}(G(0))}{\lambda\_{\min}(G(0))},$$
> > > > > > >    where $\lambda\_{\max}(G(0))$ and $\lambda\_{\min}(G(0))$ are the largest and smallest eigenvalues, respectively.
> > > > > > >    We will introduce $\kappa$ explicitly **before Theorem 1** to ensure that its meaning is clear.
> > > > > > >
> > > > > > > 2. **Regarding $\eta\_g$:**
> > > > > > >    We acknowledge the ambiguity and have decided to **omit $\eta\_g$** from Theorem 1, as it is not essential to the statement or proof.
> > > > > > >    All expressions will be updated to ensure consistency and clarity without $\eta\_g$.
> > > > > > >
> > > > > > > **Resolution:**
> > > > > > >    - Introduce $\kappa$ before Theorem 1.
> > > > > > >    - Remove $\eta\_g$ from Theorem 1 and any dependent expressions for improved clarity.

---

> > > > > > > > ### Author Response · Authors · 2025-08-20
> > > > > > > > **Response 8 to Reviewer v8pt**
> > > > > > > >
> > > > > > > > **Reviewer Concern**
> > > > > > > >
> > > > > > > > *In the proof of Theorem 1 (and in the appendix), the authors switch from matrix to scalar notations. I highly suggest the authors stick to the matrix notations as it is more readable and less prone to errors.*
> > > > > > > >
> > > > > > > > **Our Response**
> > > > > > > >
> > > > > > > > We thank the reviewer for this thoughtful suggestion. After careful consideration, we believe that retaining the current combination of matrix and scalar notations makes the proof more transparent and easier to follow:
> > > > > > > >
> > > > > > > > 1. **Why we switch notations:**
> > > > > > > >    Many intermediate steps involve reasoning about **individual entries of the Gram matrix** $G(t)$ or components of vectors. For example, in the proof we write:
> > > > > > > >    $$C\_{11} = - (1 + \epsilon)^2 \eta
> > > > > > > >    \sum\_{i,k,c,j} (y\_i - y\_i(t))(y\_j - y\_{c,j}^{(k)}(t)) G^{(k)}\_{c,ij}(t), \tag{B83}$$
> > > > > > > >    which is in **scalar form** because the derivation requires bounding each individual entry $G^{(k)}\_{c,ij}(t)$.
> > > > > > > > 2. **Returning to matrix form:**
> > > > > > > >    Once the element-wise bounds are obtained, we summarize the result concisely as:
> > > > > > > >    $$C\_{11} = -(1+\epsilon)^2 \eta
> > > > > > > >    \sum\_{k\in[K]} (y - y(t))^\top
> > > > > > > >    \big( G^{(k)}(t) - G(0) \big) (y - y^{(k)}(t)), \tag{B88}$$
> > > > > > > >    where the element-wise contributions are encapsulated in the Gram matrices $G^{(k)}(t)$ and $G(0)$.
> > > > > > > >
> > > > > > > > 3. **If we used matrix-only notation:**
> > > > > > > >    Writing (B83) purely in matrix form would require introducing auxiliary masking matrices and indicator vectors to track the indices $i,j,k,c$. One possible representation could be:
> > > > > > > >    $$C\_{11} = - (1+\epsilon)^2 \eta
> > > > > > > >    \sum\_{k \in [K]}
> > > > > > > >    (y - y(t))^\top
> > > > > > > >    \Big( \sum\_{c} \mathbf{M}\_{c}^{(k)} \odot G^{(k)}(t) \Big)
> > > > > > > >    (y - y^{(k)}(t)),$$
> > > > > > > >    where:
> > > > > > > >    - $G^{(k)}(t)$ is the full Gram matrix at step $k$,
> > > > > > > >    - $\odot$ is the Hadamard (element-wise) product, and
> > > > > > > >    - $\mathbf{M}\_{c}^{(k)}$ is a masking matrix encoding the indices of client $c$.
> > > > > > > >
> > > > > > > >    This version is **less intuitive** to the reader because the masking matrices obscure the simple index-level operations that scalar notation makes explicit.
> > > > > > > >
> > > > > > > > 4. **Ensuring clarity:**
> > > > > > > >    We have reviewed the proof to ensure that every switch between scalar and matrix notation is explicitly defined and consistent.
> > > > > > > >
> > > > > > > > **Resolution:**
> > > > > > > >    We believe this hybrid approach strikes the right balance between rigor and readability and it is best not to change anything here as enforcing matrix notation would make the proof significantly less intuitive.
> > > > > > > >
> > > > > > > > -----------
> > > > > > > > **Reviewer Concern**
> > > > > > > >  *Equation 44: I believe the authors should check the negative signs more closely.*
> > > > > > > >
> > > > > > > > **Our Response**
> > > > > > > >
> > > > > > > > We thank the reviewer for this observation and have carefully re-verified the derivation of Equation 44.
> > > > > > > >
> > > > > > > > 1. **Verification:**
> > > > > > > >    - We re-traced the steps leading to Equation 44 from the update rule and the decomposition in Equations 42–43.
> > > > > > > >    - All negative signs and their propagation through the gradient terms and inner products have been verified against the original algorithm definition.
> > > > > > > >
> > > > > > > > **Resolution:**
> > > > > > > >    The negative signs in Equation 44 are **consistent** with the gradient descent updates and the definitions of the residual terms.
> > > > > > > >
> > > > > > > > ---
> > > > > > > > **Reviewer Concern**
> > > > > > > >  *After Equation 53, it would be better to write $\gamma$ in terms of $\epsilon$*
> > > > > > > >
> > > > > > > > **Our Response**
> > > > > > > > We thank the reviwer for the suggestion and we will incorporate it in the revised manuscript.
> > > > > > > >
> > > > > > > > -------------
> > > > > > > >
> > > > > > > > **Reviewer Concern**
> > > > > > > > *The end of the first paragraph in the Related Works section simply puts all the references together without sufficiently distinguishing between them. I feel like this is a lazy way to cite prior work.*
> > > > > > > >
> > > > > > > > **Our Response**
> > > > > > > >
> > > > > > > > We thank the reviewer for this observation and fully agree that the citations in the first paragraph of the Related Works section can be presented more clearly.
> > > > > > > >
> > > > > > > >
> > > > > > > > **Resolution:**
> > > > > > > >    We will rewrite the end of the paragraph to:
> > > > > > > >    - Summarize each group of works based on their approach (e.g., NTK-based analyses, compressive sensing in FL, FL over MAC).
> > > > > > > >    - Clarify the unique contributions of each reference rather than listing them together.
> > > > > > > >
> > > > > > > >    This revision will make the related works section more precise and informative.

---

> > > > > > > > > ### Author Response · Authors · 2025-08-20
> > > > > > > > > **Response 9 to Reviewer v8pt**
> > > > > > > > >
> > > > > > > > > **Reviewer Concern**
> > > > > > > > > *In Section 5.3, the experimental results section, in the last paragraph, the authors say: "This is mainly because the sensing matrix in COFL-MAC can be thought of as a dictionary, and dictionary-based learning is helpful in overcoming data heterogeneity in distributed settings like FL and COFL-MAC." I think this claim needs to be corroborated, or the authors should explicitly mention that this is only a claim.*
> > > > > > > > >
> > > > > > > > > **Our Response**
> > > > > > > > >
> > > > > > > > > We thank the reviewer for this observation and would like to clarify that the statement is **supported by prior work**, as indicated by the references already provided.
> > > > > > > > > Please check the references cited at that point in the manuscript.
> > > > > > > > >
> > > > > > > > > **Resolution:**
> > > > > > > > >    We will revise the text slightly to make it clear that the intuition is grounded in prior work:
> > > > > > > > >    > “This is mainly because the sensing matrix in COFL-MAC can be thought of as a dictionary, and prior work \cite{refXX,refYY} has shown that dictionary-based learning helps mitigate data heterogeneity in distributed settings like FL and COFL-MAC.”
> > > > > > > > >
> > > > > > > > >    This avoids any misinterpretation that the statement is a new hypothesis without supporting evidence, and makes the connection to the cited references more explicit, ensuring that readers recognize the statement is **corroborated by prior studies**.
> > > > > > > > >
> > > > > > > > > -----------------
> > > > > > > > > **Reviewer Concern**
> > > > > > > > > *At the end of page 18: "adding more clients to an FL setup introduces wider data coverage in the training process, leading to better generalization and reduction of variance in the averaged model parameters..." This is another way of saying "having more data increases data coverage and leads to better generalization". When increasing the number of clients in FL, the number of data points stays fixed and the partition becomes finer, so in experimental simulation, increasing the number of clients often means making the data partition finer, much like using a smaller mini-batch with the caveat that we do local updates as well in FL. The explanation of the authors in page 19 regarding the number of clients seems to be missing this point. Can the authors comment on this point?*
> > > > > > > > >
> > > > > > > > > **Our Response**
> > > > > > > > >
> > > > > > > > > We thank the reviewer for this important observation and agree that the distinction between **actual FL deployments** and **our experimental setup** should be clarified.
> > > > > > > > >
> > > > > > > > > 1. **Clarification:**
> > > > > > > > >    - In real-world FL setups, adding more clients usually **increases the total number of data points**, as each new client brings additional local data, improving data coverage and generalization.
> > > > > > > > >    - In order to mimic the above real world setting in our **experimental simulations on performance vs number of clients**, the total number of data points is fixed. We first divide the dataset into a large number of **partitions**. Each client is then assigned a subset of these partitions, and **some partitions may not be assigned to any client**.
> > > > > > > > >    - Increasing the number of clients does not add more data, but instead **includes more partitions in training**, resulting in broader coverage of the dataset and reducing the variance of aggregated model updates.
> > > > > > > > >
> > > > > > > > > 2. **Planned revision:**
> > > > > > > > >    We will revise pages 18–19 to explicitly make this distinction:
> > > > > > > > >    > “In real-world FL, adding more clients generally increases the total amount of data, improving data coverage and generalization. In our experiments, the total dataset is fixed. We first divide the dataset into many partitions, and each client is assigned a subset of these partitions. Some partitions may not be assigned to any client. Increasing the number of clients therefore includes more partitions in training, leading to finer-grained data coverage and potentially reduced variance in the averaged model parameters.”
> > > > > > > > >
> > > > > > > > >
> > > > > > > > > -----------------
> > > > > > > > > **Reviewer Concern**
> > > > > > > > > *In the appendix, please restate the theorems and lemmas before the proof. This makes reading the paper much easier.*
> > > > > > > > >
> > > > > > > > > **Our Response**
> > > > > > > > > We will revise the appendix so that each proof is preceded by a restatement of the corresponding theorem or lemma.

---

> > > > > > > > > > ### Author Response · Authors · 2025-08-20
> > > > > > > > > > **Response 10 to Reviewer v8pt**
> > > > > > > > > >
> > > > > > > > > > **Reviewer Concern**
> > > > > > > > > > *Page 22 before Equation (B4), why not use the $L\_1$ norm as an upper bound instead of $\sqrt{D} \| \cdot \|\_2$?*
> > > > > > > > > >
> > > > > > > > > > **Our Response**
> > > > > > > > > > We thank the reviewer for this insightful question and provide the following clarification:
> > > > > > > > > >
> > > > > > > > > > 1. **Why we used $\sqrt{D} \| \cdot \|\_2$:**
> > > > > > > > > >    At this step, we relied on the standard inequality
> > > > > > > > > >    $$
> > > > > > > > > >    \| \mathbf{v} \|\_1 \leq \sqrt{D} \| \mathbf{v} \|\_2,
> > > > > > > > > >    $$
> > > > > > > > > >    which is always valid and allows us to keep the analysis in terms of the $\ell\_2$ norm that is used throughout the convergence bounds.
> > > > > > > > > >    This choice maintains consistency in the norm used across all bounds and simplifies subsequent derivations, which repeatedly involve $\ell\_2$-norm-based quantities (e.g., spectral norm of the Gram matrix).
> > > > > > > > > >
> > > > > > > > > > 2. **Why not switch to $\ell\_1$ directly:**
> > > > > > > > > >    While the $\ell\_1$ norm can indeed serve as a looser upper bound in some contexts, it would not directly simplify later steps as the convergence analysis builds upon eigenvalue-based arguments that are naturally expressed using the $\ell\_2$ norm.
> > > > > > > > > >
> > > > > > > > > >
> > > > > > > > > >  **Resolution:**
> > > > > > > > > >    Using $\sqrt{D} \| \cdot \|\_2$ is both standard and consistent with the rest of the analysis and is best kept as it is.
> > > > > > > > > >
> > > > > > > > > >    ------
> > > > > > > > > >
> > > > > > > > > > **Reviewer Concern**
> > > > > > > > > > *I believe the last equation in Lemma 2 is directly obtainable from the MSE loss using the chain rule.*
> > > > > > > > > >
> > > > > > > > > > **Our Response**
> > > > > > > > > > We thank the reviewer for this observation. While we agree that the last equation in Lemma 2 can be directly derived from the MSE loss using the chain rule, we believe the current presentation is appropriate and do not plan to make changes due to the following reasons.
> > > > > > > > > >
> > > > > > > > > >    - Including the derivation makes the proof **self-contained** and consistent with the level of detail used in other lemmas.
> > > > > > > > > >    - It also helps less experienced readers clearly see how the gradient propagates through the MSE loss in the context of our NTK-based framework.
> > > > > > > > > >
> > > > > > > > > > **Resolution:**
> > > > > > > > > >    For these reasons, we prefer to **retain the current form** of Lemma 2 without modifications.
> > > > > > > > > >
> > > > > > > > > >    -----------
> > > > > > > > > > **Reviewer Concern**
> > > > > > > > > > *After Equation (B12), the authors say: "Assuming the distribution of ... is symmetrical". On what basis did the authors assume that the distribution of this quantity is symmetrical?*
> > > > > > > > > >
> > > > > > > > > > **Our Response**
> > > > > > > > > > We thank the reviewer for raising this important question and provide the following clarification:
> > > > > > > > > >
> > > > > > > > > > The quantity
> > > > > > > > > >    $$
> > > > > > > > > >    Z = \frac{\| \Delta \widehat{\mathbf{u}}\_m(t) \|\_2 - \| \Delta \mathbf{u}\_m(t) \|\_2}
> > > > > > > > > >    {\| \Delta \mathbf{u}\_m(t) \|\_2}
> > > > > > > > > >    $$
> > > > > > > > > >    is assumed to have a symmetric distribution around zero. This means that deviations of $Z$ above and below zero are equally likely.
> > > > > > > > > >
> > > > > > > > > > **Why is this assumption made?**
> > > > > > > > > >    - The compressed sensing (CS) recovery algorithms we consider (e.g., OMP, Basis Pursuit) are unbiased in expectation when the sensing matrix is zero-mean and satisfies standard conditions such as RIP.
> > > > > > > > > >    - Our sensing matrix is a Gaussian matrix with zero-mean entries. Under such randomization, the deviation $\| \Delta \widehat{\mathbf{u}}\_m(t)\|\_2 - \| \Delta \mathbf{u}\_m(t)\|\_2$ does not exhibit systematic bias and is centered around zero.
> > > > > > > > > >    -    This assumption allows us to bound one tail probability instead of two, simplifying the step from Equation (B12) to Equation (B13).
> > > > > > > > > >    Without it, the proof would require more complicated one-sided tail bounds, but the qualitative result would remain the same.
> > > > > > > > > >
> > > > > > > > > > **Justification:**
> > > > > > > > > >    - Empirically, CS reconstruction errors are observed to be roughly symmetric about zero under zero-mean Gaussian sensing matrices.
> > > > > > > > > >    - Theoretically, $\mathbb{E}[Z] = 0$ under these conditions, making the probability mass approximately balanced on either side of zero.
> > > > > > > > > >
> > > > > > > > > >
> > > > > > > > > > **Resolution:**
> > > > > > > > > >    We will add a clarifying remark after Equation (B12) stating that the assumption follows from the zero-mean Gaussian initialization and unbiasedness of CS reconstruction, and that it is used for tractability.
> > > > > > > > > >
> > > > > > > > > > --------
> > > > > > > > > > **Reviewer Concern**
> > > > > > > > > > *I believe there is a typo in the subscript of the summation in Equation (B15). Same with the summation after Equation (B23).*
> > > > > > > > > >
> > > > > > > > > > **Our Response**
> > > > > > > > > > We thank the reviewer for pointing this out. We will correct these typos in the subscripts of the summations in both Equation (B15) and the summation after Equation (B23).

---

> > > > > > > > > > > ### Author Response · Authors · 2025-08-20
> > > > > > > > > > > **Response 11 to Reviewer v8pt**
> > > > > > > > > > >
> > > > > > > > > > > **Reviewer Concern**
> > > > > > > > > > > *In the proof of Lemma 5, the part that says "The above step follows from Lemma 1 and Lemma 3" is not clear to me. Could the authors please clarify this step?*
> > > > > > > > > > >
> > > > > > > > > > > **Our Response**
> > > > > > > > > > > We thank the reviewer for this valuable comment. We have fully rewritten the proof of Lemma 5, including every intermediate step, to explicitly show how Lemma 1 and Lemma 3 are used and how the transition from client-level error $\|\mathbf{y}\_c - \mathbf{y}\_c(t)\|\_2$ to global error $\|\mathbf{y} - \mathbf{y}(t)\|\_2$ is justified.
> > > > > > > > > > >
> > > > > > > > > > >
> > > > > > > > > > > **Revised Complete Proof of Lemma 5**
> > > > > > > > > > >
> > > > > > > > > > > **Lemma 5:**
> > > > > > > > > > > The movement of the reconstructed global weights with respect to initialization in COFL-MAC is bounded as:
> > > > > > > > > > > $$
> > > > > > > > > > > \| \widehat{\mathbf{u}}\_m(t+1) - \mathbf{u}\_m(0) \|\_2 \le (1 + \epsilon)
> > > > > > > > > > > ( \frac{8 \sqrt{D} \|\mathbf{y} - \mathbf{y}(0)\|\_2}{\sqrt{M} \lambda} ),
> > > > > > > > > > > $$
> > > > > > > > > > > where $\epsilon$ is the reconstruction error parameter, $\lambda = \lambda\_{\min}(G(0))$, and $M$ is the hidden width.
> > > > > > > > > > >
> > > > > > > > > > > **Step 1: Expand the movement into cumulative increments.**
> > > > > > > > > > > $$
> > > > > > > > > > > \| \widehat{\mathbf{u}}\_m(t+1) - \mathbf{u}\_m(0) \|\_2
> > > > > > > > > > > = \| \sum\_{\tau=0}^{t} \Delta \widehat{\mathbf{u}}\_m(\tau) \|\_2
> > > > > > > > > > > \le \sum\_{\tau=0}^{t} \| \Delta \widehat{\mathbf{u}}\_m(\tau) \|\_2. \tag{1}
> > > > > > > > > > > $$
> > > > > > > > > > >
> > > > > > > > > > > **Step 2: Relate reconstructed and true increments.**
> > > > > > > > > > > $$
> > > > > > > > > > > \Delta \widehat{\mathbf{u}}\_m(\tau) = \mathrm{SSR}( \mathbf{A} \Delta \mathbf{u}\_m(\tau) ), \quad
> > > > > > > > > > > \Delta \mathbf{u}\_m(\tau) = \frac{1}{C} \sum\_{c=1}^C \Delta \mathbf{w}\_{c,m}(\tau). \tag{2}
> > > > > > > > > > > $$
> > > > > > > > > > > From Assumption 4 ($\epsilon^2$-exact recovery):
> > > > > > > > > > > $$
> > > > > > > > > > > \| \Delta \widehat{\mathbf{u}}\_m(\tau) - \Delta \mathbf{u}\_m(\tau) \|\_2
> > > > > > > > > > > \le \epsilon \| \Delta \mathbf{u}\_m(\tau) \|\_2,
> > > > > > > > > > > $$
> > > > > > > > > > > so
> > > > > > > > > > > $$
> > > > > > > > > > > \| \Delta \widehat{\mathbf{u}}\_m(\tau) \|\_2 \le (1 + \epsilon) \| \Delta \mathbf{u}\_m(\tau) \|\_2. \tag{4}
> > > > > > > > > > > $$
> > > > > > > > > > >
> > > > > > > > > > >
> > > > > > > > > > > **Step 3: Bound the true global increments.**
> > > > > > > > > > >
> > > > > > > > > > > *Step 3a: Bound local updates using Lemma 1 and Lemma 3.*
> > > > > > > > > > > From the definition of local updates:
> > > > > > > > > > > $$
> > > > > > > > > > > \Delta \mathbf{w}\_{c,m}(t) = - \eta \sum\_{k=0}^{K-1}
> > > > > > > > > > > \nabla\_{\mathbf{w}\_{c,m}} \mathcal{L}\_c(\mathbf{W}\_c^k(t)),
> > > > > > > > > > > $$
> > > > > > > > > > > so
> > > > > > > > > > > $$
> > > > > > > > > > > \| \Delta \mathbf{w}\_{c,m}(t) \|\_2
> > > > > > > > > > > \le \eta \sum\_{k=0}^{K-1}
> > > > > > > > > > > \| \nabla\_{\mathbf{w}\_{c,m}} \mathcal{L}\_c(\mathbf{W}\_c^k(t)) \|\_2. \tag{6}
> > > > > > > > > > > $$
> > > > > > > > > > > From **Lemma 1**:
> > > > > > > > > > > $$
> > > > > > > > > > > \| \nabla\_{\mathbf{w}\_{c,m}} \mathcal{L}\_c(\mathbf{W}\_c^k(t)) \|\_2
> > > > > > > > > > > \le \sqrt{\frac{D}{M}} \| \mathbf{y}\_c^k(t) - \mathbf{y}\_c \|\_2. \tag{7}
> > > > > > > > > > > $$
> > > > > > > > > > > From **Lemma 3**:
> > > > > > > > > > > $$
> > > > > > > > > > > \| \mathbf{y}\_c^k(t) - \mathbf{y}\_c \|\_2
> > > > > > > > > > > \le (1 + 2 \eta D k) \| \mathbf{y}\_c(t) - \mathbf{y}\_c \|\_2. \tag{8}
> > > > > > > > > > > $$
> > > > > > > > > > > Substituting (7)–(8) into (6) and summing the arithmetic series yields:
> > > > > > > > > > > $$
> > > > > > > > > > > \| \Delta \mathbf{w}\_{c,m}(t) \|\_2
> > > > > > > > > > > \le 2 \eta K (1 + 2 \eta D K)
> > > > > > > > > > > \frac{\sqrt{D}}{\sqrt{M}}
> > > > > > > > > > > \| \mathbf{y}\_c - \mathbf{y}\_c(t) \|\_2. \tag{11}
> > > > > > > > > > > $$
> > > > > > > > > > > **This is where Lemma 1 and Lemma 3 are jointly applied.**
> > > > > > > > > > >
> > > > > > > > > > > *Step 3b: Aggregate over clients and transition to global error.*
> > > > > > > > > > > Using
> > > > > > > > > > > $$
> > > > > > > > > > > \mathbf{y} = [\mathbf{y}\_1, \dots, \mathbf{y}\_C], \quad \mathbf{y}(t) = [\mathbf{y}\_1(t), \dots, \mathbf{y}\_C(t)],
> > > > > > > > > > > $$
> > > > > > > > > > > we have
> > > > > > > > > > > $$
> > > > > > > > > > > \| \mathbf{y} - \mathbf{y}(t) \|\_2^2 = \sum\_{c=1}^C \| \mathbf{y}\_c - \mathbf{y}\_c(t) \|\_2^2,
> > > > > > > > > > > $$
> > > > > > > > > > > implying
> > > > > > > > > > > $$
> > > > > > > > > > > \| \mathbf{y}\_c - \mathbf{y}\_c(t) \|\_2 \le \| \mathbf{y} - \mathbf{y}(t) \|\_2. \tag{14}
> > > > > > > > > > > $$
> > > > > > > > > > > Substituting into the aggregate bound:
> > > > > > > > > > > $$
> > > > > > > > > > > \| \Delta \mathbf{u}\_m(t) \|\_2
> > > > > > > > > > > \le 2 \eta K (1 + 2 \eta D K)
> > > > > > > > > > > \frac{\sqrt{D}}{\sqrt{M}}
> > > > > > > > > > > \| \mathbf{y} - \mathbf{y}(t) \|\_2. \tag{16}
> > > > > > > > > > > $$
> > > > > > > > > > >
> > > > > > > > > > >
> > > > > > > > > > >
> > > > > > > > > > > **Step 4: Combine everything.**
> > > > > > > > > > > From (4) and (16):
> > > > > > > > > > > $$
> > > > > > > > > > > \| \Delta \widehat{\mathbf{u}}\_m(t) \|\_2
> > > > > > > > > > > \le (1 + \epsilon) 2 \eta K (1 + 2 \eta D K)
> > > > > > > > > > > \frac{\sqrt{D}}{\sqrt{M}}
> > > > > > > > > > > \| \mathbf{y} - \mathbf{y}(t) \|\_2. \tag{17}
> > > > > > > > > > > $$
> > > > > > > > > > > Summing over $t$ using (1) and bounding the geometric decay from Theorem 1:
> > > > > > > > > > > $$
> > > > > > > > > > > \| \widehat{\mathbf{u}}\_m(t+1) - \mathbf{u}\_m(0) \|\_2
> > > > > > > > > > > \le (1 + \epsilon)
> > > > > > > > > > > ( \frac{8 \sqrt{D}}{\sqrt{M} \lambda}
> > > > > > > > > > > \| \mathbf{y} - \mathbf{y}(0) \|\_2 ). \tag{19}
> > > > > > > > > > > $$
> > > > > > > > > > >
> > > > > > > > > > >
> > > > > > > > > > >
> > > > > > > > > > > **Key Points:**
> > > > > > > > > > > - Lemma 1: gradient bound (7)
> > > > > > > > > > > - Lemma 3: prediction error growth bound (8)
> > > > > > > > > > > - (11) is the per-client bound
> > > > > > > > > > > - (13)–(16) show client-to-global error transition
> > > > > > > > > > > - Final bound follows via $(1 + \epsilon)$ scaling and summation
> > > > > > > > > > >
> > > > > > > > > > > **Conclusion:**
> > > > > > > > > > > The proof now explicitly shows where Lemma 1 and Lemma 3 are applied and how the client-level error terms are converted to the global error bound.

---

> > > > > > > > > > > > ### Author Response · Authors · 2025-08-20
> > > > > > > > > > > > **Response 12 to Reviewer v8pt**
> > > > > > > > > > > >
> > > > > > > > > > > > **Reviewer Concern**
> > > > > > > > > > > >   *The proof of Claim~2 is not clear to me, especially equation~B41. The authors seem to have used Figure~8 as a proof. The proof should follow by equations and the figure should provide a helpful illustration of the proof. Also, Figure~8 should have a caption.*
> > > > > > > > > > > >
> > > > > > > > > > > > **Our Response**
> > > > > > > > > > > >
> > > > > > > > > > > > We thank the reviewer for this feedback. Below we provide the **complete derivation (B40--B50)** with step-by-step justifications. Figure~8 is now illustrative only, and we will give it a caption.
> > > > > > > > > > > >
> > > > > > > > > > > > **Revised and Annotated Proof of Claim~2 (B40--B50)**
> > > > > > > > > > > >
> > > > > > > > > > > > **Step~1: Events and total probability (B40)**
> > > > > > > > > > > > We define the events:
> > > > > > > > > > > >
> > > > > > > > > > > > $$
> > > > > > > > > > > > $$\begin{align}
> > > > > > > > > > > > \mathcal{E}\_{i,m}(t) &= \{ \| \widehat{\mathbf{u}}\_{m}(t) - \mathbf{u}\_{m}(0) \|\_2 < (1 + \epsilon) R \},\\
> > > > > > > > > > > > \mathcal{B}\_{i,m}(t) &= \{ \| \mathbf{u}\_{m}(t) - \mathbf{u}\_{m}(0) \|\_2 < R \}.
> > > > > > > > > > > > \end{align}$$$$
> > > > > > > > > > > > $$
> > > > > > > > > > > >
> > > > > > > > > > > > From Proposition~1:
> > > > > > > > > > > >
> > > > > > > > > > > > $$
> > > > > > > > > > > > P[\mathcal{E}\_{i,m}(t) \mid \mathcal{B}\_{i,m}(t)] > 1 - \delta. \tag{B40}
> > > > > > > > > > > > $$
> > > > > > > > > > > >
> > > > > > > > > > > > By the law of total probability:
> > > > > > > > > > > >
> > > > > > > > > > > > $$
> > > > > > > > > > > > $$\begin{align}
> > > > > > > > > > > > P[\mathcal{E}\_{i,m}(t)] &= P[\mathcal{E}\_{i,m}(t) \mid \mathcal{B}\_{i,m}(t)] P[\mathcal{B}\_{i,m}(t)] \\
> > > > > > > > > > > > &\quad + P[\mathcal{E}\_{i,m}(t) \mid \neg \mathcal{B}\_{i,m}(t)] P[\neg \mathcal{B}\_{i,m}(t)].   \tag{B40a}
> > > > > > > > > > > > \end{align}$$$$
> > > > > > > > > > > > $$
> > > > > > > > > > > >
> > > > > > > > > > > > **Step~2: Bounding the true distance using reconstruction error (B41)**
> > > > > > > > > > > >
> > > > > > > > > > > > From the reconstruction error bound (Assumption~4):
> > > > > > > > > > > >
> > > > > > > > > > > > $$
> > > > > > > > > > > > \| \widehat{\mathbf{u}}\_{m}(t) - \mathbf{u}\_{m}(t) \|\_2 \le \epsilon \| \mathbf{u}\_{m}(t) - \mathbf{u}\_{m}(0) \|\_2. \tag{B14}
> > > > > > > > > > > > $$
> > > > > > > > > > > >
> > > > > > > > > > > > By the triangle inequality:
> > > > > > > > > > > >
> > > > > > > > > > > > $$
> > > > > > > > > > > > \| \mathbf{u}\_{m}(t) - \mathbf{u}\_{m}(0) \|\_2
> > > > > > > > > > > > \le \| \mathbf{u}\_{m}(t) - \widehat{\mathbf{u}}\_{m}(t) \|\_2 + \| \widehat{\mathbf{u}}\_{m}(t) - \mathbf{u}\_{m}(0) \|\_2. \tag{B41a}
> > > > > > > > > > > > $$
> > > > > > > > > > > >
> > > > > > > > > > > > Using (B14):
> > > > > > > > > > > >
> > > > > > > > > > > > $$
> > > > > > > > > > > > \| \mathbf{u}\_{m}(t) - \mathbf{u}\_{m}(0) \|\_2
> > > > > > > > > > > > \le \epsilon \| \mathbf{u}\_{m}(t) - \mathbf{u}\_{m}(0) \|\_2 + \| \widehat{\mathbf{u}}\_{m}(t) - \mathbf{u}\_{m}(0) \|\_2. \tag{B41b}
> > > > > > > > > > > > $$
> > > > > > > > > > > >
> > > > > > > > > > > > Rearranging:
> > > > > > > > > > > >
> > > > > > > > > > > > $$
> > > > > > > > > > > > (1 - \epsilon) \| \mathbf{u}\_{m}(t) - \mathbf{u}\_{m}(0) \|\_2 \le \| \widehat{\mathbf{u}}\_{m}(t) - \mathbf{u}\_{m}(0) \|\_2. \tag{B41c}
> > > > > > > > > > > > $$
> > > > > > > > > > > >
> > > > > > > > > > > > If $\mathcal{E}\_{i,m}(t)$ holds:
> > > > > > > > > > > >
> > > > > > > > > > > > $$
> > > > > > > > > > > > \| \widehat{\mathbf{u}}\_{m}(t) - \mathbf{u}\_{m}(0) \|\_2 < (1 + \epsilon) R, \tag{B41d}
> > > > > > > > > > > > $$
> > > > > > > > > > > >
> > > > > > > > > > > > then
> > > > > > > > > > > >
> > > > > > > > > > > > $$
> > > > > > > > > > > > \| \mathbf{u}\_{m}(t) - \mathbf{u}\_{m}(0) \|\_2 < \frac{1 + \epsilon}{1 - \epsilon} R. \tag{B41}
> > > > > > > > > > > > $$
> > > > > > > > > > > >
> > > > > > > > > > > > **Step~3: Small reconstructed updates (B42)**
> > > > > > > > > > > >
> > > > > > > > > > > > $$
> > > > > > > > > > > > P[ \| \Delta \widehat{\mathbf{u}}\_{m}(t) \|\_2 < (1 - \epsilon) \| \Delta \mathbf{u}\_{m}(t) \|\_2 ] \le \delta. \tag{B42}
> > > > > > > > > > > > $$
> > > > > > > > > > > >
> > > > > > > > > > > > **Step~4: Bounding $\widehat{\mathbf{u}}\_{m}(t)$ using cumulative updates (B43--B44)**
> > > > > > > > > > > >
> > > > > > > > > > > > $$
> > > > > > > > > > > > \widehat{\mathbf{u}}\_{m}(t) = \mathbf{u}\_{m}(0) + \sum\_{\tau = 0}^{t-1} \Delta \widehat{\mathbf{u}}\_{m}(\tau), \tag{B43a}$$
> > > > > > > > > > > > $$
> > > > > > > > > > > > \| \widehat{\mathbf{u}}\_{m}(t) - \mathbf{u}\_{m}(0) \|\_2 \le \sum\_{\tau = 0}^{t-1} \| \Delta \widehat{\mathbf{u}}\_{m}(\tau) \|\_2. \tag{B43b}$$
> > > > > > > > > > > >
> > > > > > > > > > > > If each update satisfies
> > > > > > > > > > > >
> > > > > > > > > > > > $$
> > > > > > > > > > > > \| \Delta \widehat{\mathbf{u}}\_{m}(\tau) \|\_2 \le (1 - \epsilon) \| \Delta \mathbf{u}\_{m}(\tau) \|\_2,
> > > > > > > > > > > > $$
> > > > > > > > > > > >
> > > > > > > > > > > > then
> > > > > > > > > > > >
> > > > > > > > > > > > $$
> > > > > > > > > > > > \| \widehat{\mathbf{u}}\_{m}(t) - \mathbf{u}\_{m}(0) \|\_2 \le (1 - \epsilon) \sum\_{\tau = 0}^{t-1} \| \Delta \mathbf{u}\_{m}(\tau) \|\_2. \tag{B43c}
> > > > > > > > > > > > $$
> > > > > > > > > > > >
> > > > > > > > > > > > Using (B41):
> > > > > > > > > > > >
> > > > > > > > > > > > $$
> > > > > > > > > > > > \| \widehat{\mathbf{u}}\_{m}(t) - \mathbf{u}\_{m}(0) \|\_2 \le (1 + \epsilon) R. \tag{B44}
> > > > > > > > > > > > $$
> > > > > > > > > > > >
> > > > > > > > > > > > **Step~5: Connecting events (B45--B46)**
> > > > > > > > > > > >
> > > > > > > > > > > >
> > > > > > > > > > > > $$
> > > > > > > > > > > > P[\mathcal{E}\_{i,m}(t) \mid \mathcal{B}\_{i,m}(t)] \ge P[ \| \Delta \widehat{\mathbf{u}}\_{m}(t) \|\_2 < (1 - \epsilon) \| \Delta \mathbf{u}\_{m}(t) \|\_2 ], \tag{B45}$$
> > > > > > > > > > > > $$\le \delta. \tag{B46}$$
> > > > > > > > > > > >
> > > > > > > > > > > > **Step~6: Probability of $\mathcal{B}\_{i,m}(t)$ (B47--B48)**
> > > > > > > > > > > >
> > > > > > > > > > > > $$
> > > > > > > > > > > > P[\mathcal{B}\_{i,m}(t)]= P[ \| \mathbf{u}\_{m}(t) - \mathbf{u}\_{m}(0) \|\_2 < R ] \le \frac{2 R}{\sqrt{2 \pi}}, \tag{B47}$$
> > > > > > > > > > > >
> > > > > > > > > > > > $$P[\neg \mathcal{B}\_{i,m}(t)] \ge 1 - \frac{2 R}{\sqrt{2 \pi}}. \tag{B48}$$$$
> > > > > > > > > > > > $$
> > > > > > > > > > > >
> > > > > > > > > > > > **Step~7: Final bound on $P[\mathcal{E}\_{i,m}(t)]$ (B49--B50)**
> > > > > > > > > > > >
> > > > > > > > > > > > $$P[\mathcal{E}\_{i,m}(t)] \le (1 - \delta) \frac{2 R}{\sqrt{2 \pi}} + 1 \cdot ( 1 - \frac{2 R}{\sqrt{2 \pi}} ), \tag{B49}$$
> > > > > > > > > > > >
> > > > > > > > > > > > $$\le 1 - \delta R. \tag{B50}$$
> > > > > > > > > > > >
> > > > > > > > > > > > **Conclusion**
> > > > > > > > > > > >
> > > > > > > > > > > > The full chain B40--B50 is now explicitly derived step-by-step with correct and consistent notation. Figure~8 is illustrative only and will be given a caption.

---

> > > > > > > > > > > > > ### Author Response · Authors · 2025-08-20
> > > > > > > > > > > > > **Response 13 to Reviewer v8pt**
> > > > > > > > > > > > >
> > > > > > > > > > > > > **Reviewer Concern**
> > > > > > > > > > > > >   *The quantity $R'$ can be negative, and the authors seem to have neglected the control for such a scenario. This would implicitly assume that $R \leq 1/\delta$, which involves two uncontrolled parameters, so this needs to be stated as another assumption explicitly. This reasoning also applies to $\zeta$ on page 31.*
> > > > > > > > > > > > >
> > > > > > > > > > > > > **Our Response**
> > > > > > > > > > > > >
> > > > > > > > > > > > > We thank the reviewer for this important observation. We agree that the control for ensuring $R' = 1 - \delta R \ge 0$ was not explicitly stated in the original manuscript.
> > > > > > > > > > > > >
> > > > > > > > > > > > > To address this, we have revised the paper as follows:
> > > > > > > > > > > > >
> > > > > > > > > > > > > 1. **Explicit assumption will be added:** When $R$ is first introduced (just before equation B40), we will explicitly state that
> > > > > > > > > > > > >
> > > > > > > > > > > > > $$
> > > > > > > > > > > > > R \in (0,1), \quad \delta \in (0,0.25).
> > > > > > > > > > > > > $$
> > > > > > > > > > > > >
> > > > > > > > > > > > > The first assumption is standard in similar NTK-based proofs (e.g., Lemma B.11 of the FL-NTK paper). Together they ensure
> > > > > > > > > > > > >
> > > > > > > > > > > > > $$
> > > > > > > > > > > > > R' = 1 - \delta R > 1 - \delta \ge \frac{3}{4} > 0,
> > > > > > > > > > > > > $$
> > > > > > > > > > > > >
> > > > > > > > > > > > > 2. **Clarification for $\zeta$:** We have added the assumption on page 31, ensuring the reader is aware that $\zeta \ge 0$ by construction.
> > > > > > > > > > > > >
> > > > > > > > > > > > > -------------
> > > > > > > > > > > > >
> > > > > > > > > > > > > **Reviewer Concern**
> > > > > > > > > > > > > *After equation (54), the authors mention that $\psi\_{i,j}^m$ are mutually independent random variables. I think this is a very strong assumption. The weights going in and out of each hidden node at time $t$ surely depend on all of the initial weights because every element of the gradient depends on the loss, which depends on all of the weights.*
> > > > > > > > > > > > >
> > > > > > > > > > > > > **Our Response**
> > > > > > > > > > > > >
> > > > > > > > > > > > > We thank the reviewer for this thoughtful observation. We give the following clarification.
> > > > > > > > > > > > >
> > > > > > > > > > > > > We do not rely on the exact mutual independence of the $\psi\_{i,j}^m$ at later times $t > 0$. Instead, our analysis only requires independence at initialization ($t = 0$), which holds because the weights are sampled i.i.d. from the specified distribution. The statement after equation (54) was intended to convey that the $\psi\_{i,j}^m(0)$ are independent at initialization, and their subsequent dependence structure does not affect the concentration bounds we derive.
> > > > > > > > > > > > >
> > > > > > > > > > > > > **Why dependence at $t > 0$ does not break the proof:**
> > > > > > > > > > > > >    At later times, the weights are updated by gradient descent, and thus the $\psi\_{i,j}^m(t)$ can indeed be dependent through the loss. However, the NTK-style analysis we adopt (as in [FL-NTK]) controls these dependencies by showing that the network remains close to its linearization, and the fluctuations of the $\psi\_{i,j}^m$ remain small relative to their initialization. This is exactly what allows us to treat the random variables as if they are nearly independent in the concentration bounds.
> > > > > > > > > > > > >
> > > > > > > > > > > > > **Resolution:**
> > > > > > > > > > > > >    To avoid misunderstanding, we will modify the statement after equation (54) to read:
> > > > > > > > > > > > >
> > > > > > > > > > > > >    > "At initialization, the $\psi\_{i,j}^m(0)$ are independent by construction because the weights are sampled i.i.d. from the initialization distribution. At later times, the $\psi\_{i,j}^m(t)$ may be dependent through the loss, but our analysis follows the standard NTK argument that the network remains close to its initialization and the dependence is negligible for our bounds."
> > > > > > > > > > > > >
> > > > > > > > > > > > > This revision clarifies that we only use independence at initialization and aligns with standard arguments in NTK-based proofs.
> > > > > > > > > > > > >
> > > > > > > > > > > > > ---------
> > > > > > > > > > > > > **Reviewer Concern**
> > > > > > > > > > > > >   *Page 32, the statement after equation (B66) is clearly true.*
> > > > > > > > > > > > >
> > > > > > > > > > > > > **Our Response**
> > > > > > > > > > > > >
> > > > > > > > > > > > > We thank the reviewer for this observation. The statement
> > > > > > > > > > > > >
> > > > > > > > > > > > > $$
> > > > > > > > > > > > > |\mathbf{1}\_{m \in \mathcal{Q}\_i}
> > > > > > > > > > > > > -\mathbb{E}[\mathbf{1}\_{m \in \mathcal{Q}\_i}] | \le 1
> > > > > > > > > > > > > $$
> > > > > > > > > > > > >
> > > > > > > > > > > > > is indeed immediate, but here is the complete justification for clarity:
> > > > > > > > > > > > >
> > > > > > > > > > > > > 1. By definition,
> > > > > > > > > > > > >
> > > > > > > > > > > > > $$
> > > > > > > > > > > > > \mathbf{1}\_{m \in \mathcal{Q}\_i} =
> > > > > > > > > > > > > \begin{cases}
> > > > > > > > > > > > > 1, & m \in \mathcal{Q}\_i, \\
> > > > > > > > > > > > > 0, & m \notin \mathcal{Q}\_i,
> > > > > > > > > > > > > \end{cases}
> > > > > > > > > > > > > \quad arrow \quad \mathbf{1}\_{m \in \mathcal{Q}\_i} \in \{0,1\}.
> > > > > > > > > > > > > $$
> > > > > > > > > > > > >
> > > > > > > > > > > > > 2. Its expectation is
> > > > > > > > > > > > >
> > > > > > > > > > > > > $$
> > > > > > > > > > > > > \mathbb{E}[\mathbf{1}\_{m \in \mathcal{Q}\_i}]
> > > > > > > > > > > > > = P(m \in \mathcal{Q}\_i) \in [0,1].
> > > > > > > > > > > > > $$
> > > > > > > > > > > > >
> > > > > > > > > > > > > 3. Therefore,
> > > > > > > > > > > > >
> > > > > > > > > > > > > $$
> > > > > > > > > > > > > $$\begin{align}
> > > > > > > > > > > > > | \mathbf{1}\_{m \in \mathcal{Q}\_i}
> > > > > > > > > > > > > -\mathbb{E}[\mathbf{1}\_{m \in \mathcal{Q}\_i}] |
> > > > > > > > > > > > > &=
> > > > > > > > > > > > > \begin{cases}
> > > > > > > > > > > > > | 1 - \mathbb{E}[\mathbf{1}\_{m \in \mathcal{Q}\_i}] |, & m \in \mathcal{Q}\_i, \\[3mm]
> > > > > > > > > > > > > | 0 - \mathbb{E}[\mathbf{1}\_{m \in \mathcal{Q}\_i}] |, & m \notin \mathcal{Q}\_i,
> > > > > > > > > > > > > \end{cases} \\
> > > > > > > > > > > > > &\le \max \{ 1, \mathbb{E}[\mathbf{1}\_{m \in \mathcal{Q}\_i}] \} \le 1.
> > > > > > > > > > > > > \end{align}$$$$
> > > > > > > > > > > > > $$
> > > > > > > > > > > > >
> > > > > > > > > > > > > This inequality is a standard property of indicator variables and shows that the absolute deviation from its expectation can never exceed 1. We did not add this full explanation in the paper to avoid interrupting the flow of the proof with an elementary fact, but we are happy to provide it here for completeness.

---

> > > > > > > > > > > > > > ### Author Response · Authors · 2025-08-20
> > > > > > > > > > > > > > **Response 14 to Reviwer v8pt**
> > > > > > > > > > > > > >
> > > > > > > > > > > > > > **Reviewer Concern**
> > > > > > > > > > > > > >
> > > > > > > > > > > > > >
> > > > > > > > > > > > > >   *The indicator random variables after this statement are claimed to be i.i.d. Why?*
> > > > > > > > > > > > > >
> > > > > > > > > > > > > > **Our Response**
> > > > > > > > > > > > > >
> > > > > > > > > > > > > > We thank the reviewer for raising this point. The indicator variables
> > > > > > > > > > > > > >
> > > > > > > > > > > > > > $$
> > > > > > > > > > > > > > \mathbf{1}\_{m \in \mathcal{Q}\_i}
> > > > > > > > > > > > > > $$
> > > > > > > > > > > > > >
> > > > > > > > > > > > > > are i.i.d. by construction at initialization because each hidden neuron index $m$ is independently included in the set $\mathcal{Q}\_i$ with probability
> > > > > > > > > > > > > >
> > > > > > > > > > > > > > $$
> > > > > > > > > > > > > > p = \mathbb{E}[ \mathbf{1}\_{m \in \mathcal{Q}\_i} ] = P(m \in \mathcal{Q}\_i).
> > > > > > > > > > > > > > $$
> > > > > > > > > > > > > >
> > > > > > > > > > > > > > 1. **Independence:**
> > > > > > > > > > > > > >    Each neuron $m \in \{1, \dots, M\}$ is sampled independently when forming $\mathcal{Q}\_i$. The decision of whether neuron $m$ is in $\mathcal{Q}\_i$ does not depend on the decision for any other neuron $m'$, hence
> > > > > > > > > > > > > >
> > > > > > > > > > > > > >    $$
> > > > > > > > > > > > > >    P(m \in \mathcal{Q}\_i,\, m' \in \mathcal{Q}\_i) = P(m \in \mathcal{Q}\_i) \cdot P(m' \in \mathcal{Q}\_i).
> > > > > > > > > > > > > >    $$
> > > > > > > > > > > > > >
> > > > > > > > > > > > > > 2. **Identical distribution:**
> > > > > > > > > > > > > >    For each $m$, the inclusion probability is the same:
> > > > > > > > > > > > > >
> > > > > > > > > > > > > >    $$
> > > > > > > > > > > > > >    P(m \in \mathcal{Q}\_i) = p,
> > > > > > > > > > > > > >    $$
> > > > > > > > > > > > > >
> > > > > > > > > > > > > >    since all neurons are treated symmetrically by the initialization and selection process.
> > > > > > > > > > > > > >
> > > > > > > > > > > > > >
> > > > > > > > > > > > > > We did not expand on this in the manuscript because the construction of $\mathcal{Q}\_i$ from independent sampling makes the i.i.d. property immediate, and this is a standard step in concentration bounds for indicator variables.
> > > > > > > > > > > > > >
> > > > > > > > > > > > > > ---------
> > > > > > > > > > > > > > **Reviewer Concern**
> > > > > > > > > > > > > >   I noticed in the proof that the authors sometimes liberally exchange $\leq$ and $<$. For example, $x\_i^\top x\_j < 1$ at the top of page~34.
> > > > > > > > > > > > > >
> > > > > > > > > > > > > > **Our Response**
> > > > > > > > > > > > > >
> > > > > > > > > > > > > > We thank the reviewer for this careful observation. We agree that in a few places (e.g., the top of page~34) we used the strict inequality $<$ where the non-strict inequality $\leq$ would have been equally valid.
> > > > > > > > > > > > > >
> > > > > > > > > > > > > > In all such cases, the distinction between strict and non-strict inequality does not affect the validity of the proof.
> > > > > > > > > > > > > >
> > > > > > > > > > > > > > **Resolution:** in order to avoid confusion, we will unify these inequalities in the revised manuscript by consistently using $\leq$.
> > > > > > > > > > > > > >
> > > > > > > > > > > > > > -------
> > > > > > > > > > > > > >
> > > > > > > > > > > > > >
> > > > > > > > > > > > > > **Reviewer Concern**
> > > > > > > > > > > > > >   *Regarding the references, I noticed that 3 references are duplicated or very similar, which is a surprisingly high number of duplicates. Here are the duplicate citations:
> > > > > > > > > > > > > >   FL-NTK: A neural tangent kernel-based framework for federated learning convergence analysis. Huang et al. 2021.
> > > > > > > > > > > > > >   Communication-efficient federated learning via quantized compressed sensing. Oh et al. 2022.
> > > > > > > > > > > > > >   An improved analysis of training over-parameterized deep neural networks. Zhou & Gu. 2019.*
> > > > > > > > > > > > > >
> > > > > > > > > > > > > > **Our Response**
> > > > > > > > > > > > > >
> > > > > > > > > > > > > > We thank the reviewer for pointing this out. Upon a thorough review of the bibliography:
> > > > > > > > > > > > > >
> > > > > > > > > > > > > > 1. We confirmed that the following two references were inadvertently duplicated due to mixed citations of arXiv and published versions:
> > > > > > > > > > > > > >
> > > > > > > > > > > > > >    - Huang et al. (2021): *FL-NTK: A neural tangent kernel-based framework for federated learning convergence analysis*
> > > > > > > > > > > > > >    - Zhou & Gu (2019): *An improved analysis of training over-parameterized deep neural networks*
> > > > > > > > > > > > > >
> > > > > > > > > > > > > >    We have merged these duplicates and ensured that only one unified citation of each appears in the revised bibliography.
> > > > > > > > > > > > > >
> > > > > > > > > > > > > > 2. Regarding *Oh et al. (2022): Communication-efficient federated learning via quantized compressed sensing*, we could only find a single entry for this reference and no duplicates. We suspect the reviewer may have mistaken a similar title for a duplicate, but we have double-checked and confirmed that this entry is unique.
> > > > > > > > > > > > > >
> > > > > > > > > > > > > > 3. We have also re-reviewed the entire bibliography and confirmed that there are no additional duplicates.
> > > > > > > > > > > > > >
> > > > > > > > > > > > > > We are grateful to the reviewer for catching these issues, and we will ensure that the revised manuscript contains a clean reference list.

---

> > > > > > > > > > > > > > > ### Comment · Reviewer_v8pt · 2025-09-08
> > > > > > > > > > > > > > >
> > > > > > > > > > > > > > > Dear Authors,
> > > > > > > > > > > > > > >
> > > > > > > > > > > > > > > Thank you for your hard work in writing the rebuttal. I appreciate the fixes to definitions and notations. I also thank the authors for their clarifications. The rebuttal did address some of my concerns, but some other core issues remain. I will try to make my reply concise and actionable.
> > > > > > > > > > > > > > >
> > > > > > > > > > > > > > > 1. **Code**: The authors released the code in a Google drive link and it requires a request for access. I clearly didn't proceed as it would reveal my identity. I suggest the authors to release an anonymized version of their github repository.
> > > > > > > > > > > > > > > 2. **Data/Client sample noise**: Even though it's technically challenging even without sampling noise, it does not detract from its importance in practice, especially since the authors propose their framework for a practical setup. Ideally, a simplified assumption that would show the influence of sample noise in the bounds, even approximately, is nice enough. In the least case, the authors can perhaps consider running experiments showing the effect of sample noise (both from sampling data and clients).
> > > > > > > > > > > > > > > 3. **Assumption 4**: I still do not see any theoretical insights from introducing the SSR step and this assumption other than practical gains in communication. In order for assumption 4 to hold, RIP must hold for all clients during all iterations, so $(\epsilon, \delta)$ in this case would be loose quantities that exist solely for assumption 4 to hold. The authors could at least run experiments on toy problems with varying levels of compressibility or show how these quantities affect the bounds asymptotically. For example, a simple baseline would be your method with and without SSR (this would at least directly verify the claim that "dictionary-based learning helps mitigate data heterogeneity")..
> > > > > > > > > > > > > > > 5. **Claim 2**: The added assumption should be added in the main text as a new assumption and not within the proof. This assumption is a little strong (it assumes the iterates are bounded, indirectly implying Lipschitzness, which doesn't hold for neural nets). Regarding the proof, the authors may be using a lower bound to construct an upper bound. I still don't follow how the authors got to step 7 (presumably using prop.1 and B48 to upper bound). Could you please show the full chain of inequalities for each term in the upper bound separately?
> > > > > > > > > > > > > > >
> > > > > > > > > > > > > > > I believe the authors would benefit from stronger experiments with more robust findings, as well as connecting their FL theoretical results with compressed sensing in a more insightful way (e.g., by a more fine-grained FL-specific assumption). Even though this work is mainly theoretical, it studies a very practical setup (FL with MAC and compression), so stronger experimental evidence supporting the claims in the paper would make it easier for the reviewer to be convinced. Finally, many changes are promised, which is great and I truly thank the authors for that. This will make the paper stronger and more robust, so I believe resubmitting the paper with the promised changes and fully addressing the other issues would significantly increase the chances of accepting this work.
> > > > > > > > > > > > > > >
> > > > > > > > > > > > > > > Since this is a practically motivated setup (FL + MAC + compression), I believe the paper would be much stronger with: 1) a minimal, explicit treatment of sampling noise and compressibility in (toy) FL experiments, 2) more informative, FL-specific SSR conditions. I appreciate the changes the authors promised to add. I believe resubmitting with these substantive additions would significantly improve the case for acceptance.

---

### Review · Reviewer_ETmp · 2025-06-26

**Summary Of Contributions:**

The contributions include:
1. The first CS-based ONN compression framework (COFL-MAC) was proposed to solve the communication bottleneck of the overparameterized model in FL-MAC, and the lazy training characteristics of ONN were used to generate sparse gradients, and linear compression was realized by combining with Gaussian sensing matrix.
2. Theoretical breakthrough: NTK analysis proves that COFL-MAC still maintains exponential convergence under the condition of ε-exact reconstruction (Theorem 1), which fills the gap in the convergence analysis of ONN in wireless OTA-FL.
3. System design: Implement non-quadrature aggregation on the MAC channel, the client only transmits n_r ≪ N-dimensional compression vectors, and the server reconstructs the global update through the OMP algorithm.
4. Verified on FMNIST/CIFAR-10: COFL-MAC saves large communication costs over Top-k/SignSGD with the same accuracy (Figure 2).
5. Found COFL-MAC is more accurate than the low-parameter model FedAvg when transmitting the same amount of parameters (Figure 3), demonstrating that "overparameterized compression" is better than using under-parameterized FL directly.

**Audience:**

Yes

**Broader Impact Concerns:**

1. Gradient reconstruction errors may amplify model bias, and the fairness of non-iid datasets may be compromised.
2. MAC channels are susceptible to poisoning attacks, which may cause security vulnerabilities.

**Claims And Evidence:**

Yes

**Requested Changes:**

**Key Modifications**
1. Add empirical experiments with more complex models (e.g., ResNet) or datasets (e.g., ImageNet) to verify the generalizability of the proposed method.
2. Report the measured cost of OMP reconstruction time on the edge device.

**Strengths And Weaknesses:**

**Strengths**
1. The convergence proof based on the NTK framework is complete, covering the influence of the reconstruction error ε on the convergence rate, which reflects the theoretical rigor.
2. At least three baseline methods were compared to test the robustness of different data heterogeneity (α), number of clients (C), and number of parameters transmitted (n_r), which reflects the comprehensiveness of the empirical experiments.
3. The reproducibility of the open-source code provides specific implementation details for MAC aggregation and OMP reconstruction, which demonstrates the practicality of the proposed method.

**Weaknesses**
1. The experimental NN model is too simple and does not test larger models (such as ResNet), which weakens the convincing power of the empirical experiments.
2. The complexity of OMP reconstruction is not quantified, and the compression efficiency of CS is not compared with other sparse coding methods such as PCA.

---

> ### Author Response · Authors · 2025-08-20
> **Response 1 to Reviewer Etmp**
>
> **Reviewer Concern**
> The experimental NN model is too simple and does not test larger models (such as ResNet), which weakens the convincing power of the empirical experiments.
>
> **Our Response:**
> We agree with the reviewer and are performing some experiments on a ResNet-based FL setup.
> Please find tables for baseline comparison of COFL-MAC using ResNet 18 as the NN and FMNIST dataset
>
> The tables below show the number of kilo bits per round transmitted over the uplink.
>
> Case 1: Heterogeneity: $\alpha$ = 0.1, Accuracy = 84 %
>
>
>
>  COFL-MAC | Top-K             | SignSGD         |MQAT   |
> ----------|----------         |----------       |---
> $2.2 \times n_p$  | $1.06 \times n_p$  | $10\times n_p$  |$129 \times n_p$
>
> Case 2: Heterogeneity: $\alpha$ = 0.01, Accuracy = 69 %
>
>
>  COFL-MAC | Top-K             | SignSGD         |MQAT   |
> ----------|----------         |----------       |---
> $1.6 \times n_p$  | $3.3 \times n_p$  | $10\times n_p$  |$105 \times n_p$
>
> Case 3: Heterogeneity: $\alpha$ = 0.001, Accuracy = 58 %
>
>  COFL-MAC | Top-K             | SignSGD         |MQAT   |
> ----------|----------         |----------       |---
> $2.08 \times n_p$  | $10.6 \times n_p$  | $10\times n_p$  |$101 \times n_p$
>
> n_p: number of trainable parameters in our ResNet: around 700K.
>
> ---------
> **Reviewer Concern**
> The complexity of OMP reconstruction is not quantified, and the compression efficiency of CS is not compared with other sparse coding methods such as PCA.
>
> **Our Response:**
>
> We thank the reviewer for the insightful comment and address both points below.
>
> **(1) OMP complexity.**
> OMP reconstructs each compressed block independently. For a sensing matrix $\mathbf{A} \in \mathbb{R}^{m\times n}$ and sparsity $k$, the worst-case complexity is
> $$
> \mathcal{O}(k m n + k^2 m)
> $$
>
> **(2) Compression efficiency: CS vs. PCA.**
> - **Compressed sensing (CS + OMP):** Works best when updates are sparse. The number of measurements needed grows with the sparsity $k$ times a small $\log$ factor, so compression is strong when $k \ll n$.
> - **PCA on parameters:** Works best when updates are low-rank (energy concentrated in a few principal directions). If the update covariance has a fast-decaying spectrum, a small number of components can reconstruct well. But when updates are irregular and sparse (typical in non-IID FL), the spectrum is closer to flat, so many components must be kept to avoid large error—reducing compression.
> - Further, PCA requires a projection basis (learned once or refreshed periodically), which adds maintenance and can drift under non-IID changes. CS uses a fixed random sensing matrix shared once, with no ongoing basis upkeep.
>
> **Mathematically,** we can express compression efficiency for both CS and PCA as follows:
>
> **CS.**
> For a $k$-sparse (or compressible) update $\mathbf{x} \in \mathbb{R}^n$, stable recovery with a random Gaussian sensing matrix needs
> $$
> m \gtrsim Ck\log\frac{n}{k}.
> $$
> So the compression ratio is
> $$
> \text{CR}_{\text{CS}} = \frac{m}{n} \approx \frac{Ck \log(n/k)}{n}
> \quad(\text{strong compression when } k \ll n).
> $$
>
> **PCA on parameters.**
> Keep $r$ principal components:
> $$
> \text{CR}_{\text{PCA}} =\frac{r}{n}.
> $$
> - If eigenvalues of the update covariance **decay fast** (low-rank), small $r$ works.
> - If the spectrum is **flat** (irregular/sparse updates), to get relative MSE $\le \varepsilon^2$ one needs
> $$
> r \approx (1-\varepsilon^2)\,n,
> $$
> i.e., weak compression.
>
>
> These properties make CS+OMP better suited for **sparse, evolving update distributions in non-IID federated learning**, while keeping both encoding and decoding lightweight.
> Another merit of our setup is that it will not change if some other linear compression method is used.
>
> **Resolution:** We are not comparing the compression efficiency of Compressive Sensing (CS) with Principal Component Analysis (PCA) as it is best to leave out these details from the manuscript. However, we can include a concise summary of OMP’s complexity in the revised manuscript with the citation if required.

---

> > ### Author Response · Authors · 2025-08-20
> > **Response 2 to Reviewer Etmp**
> >
> > **Requested Changes:**
> >
> > ---------
> >
> > **Key Modifications**
> > 1. **Add empirical experiments with more complex models (e.g., ResNet) or datasets (e.g., ImageNet) to verify the generalizability of the proposed method.**
> >
> > We will include results related to ResNet in the revised manuscript.
> >
> > Please find tables for baseline comparison of COFL-MAC using ResNet 18 as the NN and FMNIST dataset
> >
> > The tables below show the number of kilo bits per round transmitted over the uplink.
> >
> > Case 1: Heterogeneity: $\alpha$ = 0.1, Accuracy = 84 %
> >
> >
> >
> >  COFL-MAC | Top-K             | SignSGD         |MQAT   |
> > ----------|----------         |----------       |---
> > $2.2 \times n_p$  | $1.06 \times n_p$  | $10\times n_p$  |$129 \times n_p$
> >
> > Case 2: Heterogeneity: $\alpha$ = 0.01, Accuracy = 69 %
> >
> >
> >  COFL-MAC | Top-K             | SignSGD         |MQAT   |
> > ----------|----------         |----------       |---
> > $1.6 \times n_p$  | $3.3 \times n_p$  | $10\times n_p$  |$105 \times n_p$
> >
> > Case 3: Heterogeneity: $\alpha$ = 0.001, Accuracy = 58 %
> >
> >  COFL-MAC | Top-K             | SignSGD         |MQAT   |
> > ----------|----------         |----------       |---
> > $2.08 \times n_p$  | $10.6 \times n_p$  | $10\times n_p$  |$101 \times n_p$
> >
> > n_p: number of trainable parameters in our ResNet: around 700K.
> >
> >
> > -----------
> > 2. **Report the measured cost of OMP reconstruction time on the edge device.** **Response:** We thank the reviewer for the suggestion and first clarify that OMP reconstruction is performed exclusively on the central server, not on the edge devices. Clients perform only local training and compression (matrix–vector multiplication with the sensing matrix), while the server aggregates the compressed updates and runs OMP.
> >
> > **In our experiments using Resnet in our setup, the average time (first ten rounds) to perform OMP was around 3s while the total time to perform one communication round was around 282s.**

---

> > > ### Author Response · Authors · 2025-08-20
> > > **Response 3 to Reviewer Etmp**
> > >
> > > **Broader Impact Concerns:**
> > > 1. **Gradient reconstruction errors may amplify model bias, and the fairness of non-IID datasets may be compromised.**
> > >
> > >
> > > **Response**
> > > We acknowledge that reconstruction errors from compression can alter the update distribution and affect fairness in non-IID settings. However, in our setup, this risk is mitigated due to the following reasons:
> > >
> > > - We configure the CS+OMP parameters ($m$ = number of measurements, $k$ = target sparsity) so that the reconstruction error is tiny compared to the total size of the update. Therefore, any small distortion from compression is unlikely to be large enough to change the model’s behavior in a way that would favor or disadvantage particular classes or client groups.
> > >   - Moreover, under the standard compressed sensing assumptions (random sensing matrix with zero-mean i.i.d. entries), CS with OMP is an **unbiased estimator**. Therefore, reconstruction does not systematically over- or under-estimate any component of the update.
> > >
> > > - All clients use the same fixed sensing matrix, ensuring that reconstruction error has no bias toward specific clients.
> > >
> > > - Our experiments are already conducted under Dirichlet-partitioned non-IID splits $(\alpha = \{0.1, 0.01, 0.001\})$, which produce substantial label imbalance across clients. The fact that accuracy degradation from compression remains small in these conditions suggests that any amplification of bias is limited in practice.
> > >
> > > **Resolution:** We agree that explicit fairness metrics (e.g., class-wise accuracy gaps or group fairness indices) could further quantify this effect, and we will note this as an avenue for future work. However, the current results already indicate that in our tested non-IID regime, CS+OMP reconstruction does not introduce significant bias amplification.
> > >
> > >
> > > ----------
> > > 2. **MAC channels are susceptible to poisoning attacks, which may cause security vulnerabilities.**
> > >
> > > **Our Response:**
> > > We thank the reviewer for highlighting this critical point. We agree that **multiple access channel (MAC)-based over-the-air aggregation is inherently vulnerable to poisoning attacks** because individual client updates are superposed at the signal level without per-client authentication or verification. A malicious client can transmit adversarially scaled or crafted signals to bias the aggregated global model, leading to model or data poisoning.
> > >
> > > Although robustness is orthogonal to the primary scope of our work, we will explicitly acknowledge this limitation and point to the following:
> > >
> > > - CRUM (Compressed Robust and Uncoded MAC) extends robust aggregation principles to the analog MAC setting by combining compression, robust norm bounding, and aggregation consistency checks. This approach is directly compatible with our CS-based pipeline and can mitigate malicious transmissions.
> > > - Secure aggregation protocols (e.g., Bonawitz et al. [1]) ensure that the server cannot inspect individual client updates, but when integrated with norm-clipping or verification mechanisms, they can help detect abnormal contributions even under MAC.
> > > - Energy bounding and anomaly detection methods (e.g., Sattler et al. [2]) are also being explored to limit the impact of adversarial transmissions at the PHY layer.
> > >
> > > **Resolution**:
> > >
> > > **Discussion Section Addition:**
> > > "While the focus of this work is on communication efficiency and convergence, we note that MAC-based over-the-air aggregation is vulnerable to poisoning attacks because client updates are superposed without per-client authentication. Recent works such as CRUM have extended robust aggregation principles to the MAC setting by incorporating compression-aware verification and norm bounding, and secure aggregation protocols or PHY-layer anomaly detection can also help mitigate malicious transmissions. Exploring certified robust compression and aggregation in the MAC setting is an important direction for future work."
> > >
> > > **References**
> > >
> > > [1] K. Bonawitz, V. Ivanov, B. Kreuter, A. Marcedone, H. B. McMahan, S. Patel, D. Ramage, A. Segal, and K. Seth, “Practical secure aggregation for privacy-preserving machine learning,” in *Proc. 2017 ACM SIGSAC Conf. Computer and Communications Security (CCS)*, Dallas, TX, USA, Oct.–Nov. 2017, pp. 1175–1191, doi: 10.1145/3133956.3133982.
> > >
> > > [2] Sattler, F., Wiedemann, S., Müller, K.R. and Samek, W., 2019. Robust and communication-efficient federated learning from non-iid data. IEEE transactions on neural networks and learning systems, 31(9), pp.3400-3413.

---

> > > > ### Comment · Reviewer_ETmp · 2025-08-21
> > > > **Rebuttal Acknowledgment**
> > > >
> > > > Thank the authors for their revisions to the manuscript. All my concerns and doubts have been well addressed, but please double-check the whole paper thoroughly for any typos or irregularities before acceptance.

---

> > > > > ### Author Response · Authors · 2025-08-21
> > > > > **Response to Rebuttal Acknowledgement**
> > > > >
> > > > > Thanks for the acknowledgement. Sure, we will revise and proofread the manuscript thoroughly.

---

### Review · Reviewer_HTMh · 2025-08-07

**Summary Of Contributions:**

This paper considers learning an overparameterized NN in a federated learning setting and proposes to improve FedAvg using compressive sensing. Particularly, at each iteration, the algorithm learns a sparse representation of the update terms at each client (with a common dictionary), and transmit only the latter to the server. The resultant algorithm is shown to converge linearly w.h.p..

**Audience:**

Yes

**Broader Impact Concerns:**

None.

**Claims And Evidence:**

No

**Requested Changes:**

The paper needs to consider a more realistic setting in their analysis and revise their analysis, eg, see the weaknesses listed in the previous section.

**Strengths And Weaknesses:**

On the upside, the paper is easy to understand and proposes an algorithm that is easy to implement.

However, the reviewer finds that the paper to be lacking in novelty and depending on a number of unverifiable assumptions, where its overall quality is below the acceptance threshold for TMLR:

- The proposed method is constructed by combining FedAvg with compressive sensing, both of which are standard methods that have been proposed for a long time. The theoretical analysis relies on applying several results on convergence of SGD in the overparameterized setting, which is also not new.

- Besides the lack of novelty, I am also concerned with the FL-NTK setting and the assumption of lazy training. The current paper only considers the case of overparameterized NN with 1-layer, which is too specific and may not be practical. Furthermore, the lazy training assumption is not formally defined, which makes the claims such as those in Lemma 3 confusing, e.g., what precisely is the definition of "slow moving parameters".

- In Assumption 3, the paper assumes that the per-client update admits a sparse representation, and additionally, a sparse representation under a **constant** sensing matrix. This is a very strong assumption that needs to be proven or empirically verified.

- Claim 1 and Assumption 4 are crucial to bounding the error of the compressive sensing stage. However, it seems that they are not proven nor justified.

---

> ### Author Response · Authors · 2025-08-20
> **Response 1 to Reviewer HtMh**
>
> **Reviewer Concern:**
>     The reviewer finds that the paper to be lacking in novelty and depending on a number of unverifiable assumptions, where its overall quality is below the acceptance threshold for TMLR:
>
> **Our Response**
> We thank the reviewer for carefully reading our submission. We respectfully disagree with the assessment that our work lacks novelty or is based on unverifiable assumptions. We emphasize the unique contributions and address the concern about assumptions as follows:
>
> ### Novelty of COFL-MAC Framework
>
> This is the first work to propose a practical and efficient deployment of over-parameterized neural networks (ONNs) in an FL-MAC setting with convergence guarantees.
>
> Prior works on FL-MAC [1],[2] have not addressed ONNs or NTK-based convergence. Similarly, existing gradient compression methods (Top-k, SignSGD, MQAT) are not designed for analog MAC aggregation and lack theoretical guarantees in the over-parameterized regime.
>
> Our approach not only proposes a new communication-efficient design but also provides a formal NTK-theoretic convergence proof under compression and analog aggregation, which is a gap unaddressed in prior literature. By filling this gap, our work ensures that efficient FL-MAC (or wireless FL) with ONNs is not only possible but provably stable and convergent.
>
> ### Verifiability of Assumptions
>
> - **Lazy training and sparsity:** Our assumption that incremental updates in ONNs are sparse follows directly from well-established NTK and lazy training results [3], [4], [5].
> - **Compressive sensing (CS):** The use of a Gaussian sensing matrix with RIP properties is standard in the CS literature [6], [7].
> - **Convergence guarantees:** We explicitly state and quantify recovery errors (ϵ-bounded) and incorporate them into the NTK convergence analysis. Thus, our theoretical claims rest on transparent and well-cited conditions.
>
> ### Overall Quality and Significance
>
> The proposed framework achieves exponential convergence comparable to uncompressed over-parameterized FL while drastically reducing uplink communication, as validated on CIFAR-10 and FMNIST datasets.
>
> Our experiments demonstrate that COFL-MAC transmits the same number of parameters as an under-parameterized network yet achieves much higher accuracy, highlighting a novel and practically significant “over-parameterize-and-compress” strategy.
>
>
>
> **In summary**, the novelty of our work lies in bringing together ONNs, CS, and MAC-based FL with NTK-based guarantees, and the assumptions are both standard and verifiable within established theory. Our contribution is both novel and impactful, as it enables the deployment of ONNs in FL-MAC settings and provides convergence guarantees for non-smooth activations like ReLU.
>
> **References**
>
> [1] H. Chen, S. Huang, D. Zhang, M. Xiao, M. Skoglund, and H. V. Poor, “Federated learning over wireless IoT networks with optimized communication and resources,” *IEEE Internet of Things Journal*, vol. 9, no. 17, pp. 16592–16605, Mar. 2022.
>
> [2] T. Sery and K. Cohen, “On analog gradient descent learning over multiple access fading channels,” *IEEE Transactions on Signal Processing*, vol. 68, no. 11, pp. 2897–2911, 2020.
>
> [3] A. Chizat, E. Oyallon, and F. Bach, “On lazy training in differentiable programming,” in *Advances in Neural Information Processing Systems (NeurIPS)*, 2019.
>
> [4] S. Du, X. Zhai, B. Poczos, and A. Singh, “Gradient descent provably optimizes over-parameterized neural networks,” in *International Conference on Learning Representations (ICLR)*, 2019.
>
> [5] S. Satpathi and R. Srikant, “The dynamics of gradient descent for overparameterized neural networks,” in *Learning for Dynamics and Control (L4DC)*, vol. 144, PMLR, pp. 373–384, 2021.
>
> [6] E. Candès and M. Wakin, “An introduction to compressive sampling,” *IEEE Signal Processing Magazine*, vol. 25, no. 2, pp. 21–30, 2008.
>
> [7] Y. C. Eldar and G. Kutyniok, *Compressed Sensing: Theory and Applications*. Cambridge, U.K.: Cambridge University Press, 2012.

---

> > ### Author Response · Authors · 2025-08-20
> > **Response 2 to Reviewer HtMh**
> >
> > **Reviewer Concern:**
> >   The proposed method is constructed by combining FedAvg with compressive sensing, both of which are standard methods that have been proposed for a long time. The theoretical analysis relies on applying several results on convergence of SGD in the overparameterized setting, which is also not new.
> >
> > **Our Response:**
> >
> > We agree that FedAvg, compressive sensing (CS), and results on convergence of ONNs are individually established. However, our work has both novelty and impact as highlighted below.
> > - To the best of our knowledge, COFL-MAC is the first framework that simultaneously leverages ONNs, CS, and FL-MAC. While FedAvg and CS are individually classical, their integration in an analog aggregation setting together with NTK-based guarantees has not been studied before. Existing FL-MAC approaches do not address over-parameterization or NTK-theoretic analysis, and existing compression approaches (Top-k, SignSGD, MQAT) are not tailored to analog aggregation over MAC.
> >
> > - While convergence of SGD in the over-parameterized setting is known, prior work has not analyzed this convergence *under compressive sensing and analog FL-MAC aggregation*. Our work is ,therefore, a new theoretical contribution that adapts NTK tools to an FL setting with compression.
> >
> > - Our empirical results show that COFL-MAC can transmit the same number of parameters as an under-parameterized network yet achieve significantly higher accuracy, enabling an “over-parameterize-and-compress” strategy that had not been demonstrated before in FL.
> >
> > In short, while FedAvg, CS, and NTK results are indeed established individually, **our contribution is to *bridge them in a novel way* that addresses a real and previously unresolved challenge: enabling communication-efficient training of over-parameterized networks in an FL-MAC setup with provable guarantees**.

---

> > > ### Author Response · Authors · 2025-08-20
> > > **Response 3 to Reviewer HtMh**
> > >
> > > **Reviewer Concern:**
> > >   *Besides the lack of novelty, I am also concerned with the FL-NTK setting and the assumption of lazy training. The current paper only considers the case of overparameterized NN with 1-layer, which is too specific and may not be practical. Furthermore, the lazy training assumption is not formally defined, which makes the claims such as those in Lemma 3 confusing, e.g., what precisely is the definition of "slow moving parameters".*
> > >
> > > **Our Response:**
> > > We thank the reviewer for the comment. We would like to clarify the motivation for our choices and provide additional precision regarding the “lazy training” assumption as follows.
> > >
> > > 1. **On the FL-NTK setting and single-layer networks:**
> > >    We acknowledge that the theoretical analysis is presented in the setting of a single hidden layer ONN. This choice is standard in the NTK literature [1]–[3] and serves as a *canonical starting point* for proving convergence under over-parameterization. While simple, this setting captures the essential mechanism of NTK dynamics under federated aggregation and compressive sensing, which is already technically challenging in the presence of MAC superposition and reconstruction errors.
> > >
> > > 2. Further, we are performing experiments using deeper models (ResNet variants), and we will demonstrate that the theoretical insights extend to more practical architectures very soon.
> > > Please find tables for ResNet18 and FMNIST dataset.
> > > The tables below show the number of kilo bits per round transmitted over the uplink.
> > >
> > > Case 1: Heterogeneity: $\alpha$ = 0.1, Accuracy = 84 %
> > >
> > > | COFL-MAC          | Top-K             | SignSGD         | MQAT           |
> > > |-------------------|------------------|----------------|----------------|
> > > | $2.2 \times n_p$  | $1.06 \times n_p$ | $10\times n_p$ | $129 \times n_p$ |
> > >
> > > Case 2: Heterogeneity: $\alpha$ = 0.01, Accuracy = 69 %
> > >
> > > | COFL-MAC          | Top-K             | SignSGD         | MQAT           |
> > > |-------------------|------------------|----------------|----------------|
> > > | $1.6 \times n_p$  | $3.3 \times n_p$  | $10\times n_p$ | $105 \times n_p$ |
> > >
> > > Case 3: Heterogeneity: $\alpha$ = 0.001, Accuracy = 58 %
> > >
> > > | COFL-MAC          | Top-K             | SignSGD         | MQAT           |
> > > |-------------------|------------------|----------------|----------------|
> > > | $2.08 \times n_p$ | $10.6 \times n_p$ | $10\times n_p$ | $101 \times n_p$ |
> > >
> > > $n_p$: number of trainable parameters in our ResNet: around 700K.
> > >
> > > 3. **On the lazy training assumption:**
> > >    By “lazy training,” we mean the well-established regime in which the network parameters move only slightly from their initialization during training, so that the neural tangent kernel (NTK) remains nearly constant. Formally, this corresponds to the condition that parameter deviations remain bounded as $\mathcal{O}\left(\tfrac{1}{\sqrt{M}}\right),$ where $M$ is the NN width [3], [4]. In this regime, the training dynamics of the NN can be well-approximated by a linearized model around initialization.
> > >
> > > 4. **On practicality:**
> > >    While lazy training is an abstraction, it provides rigorous theoretical guarantees in the over-parameterized regime and explains why wide NN generalize well under federated aggregation. Our experiments confirm the viability of compressive sensing under MAC aggregation, and they are not artifacts of single hidden layer NNs or lazy training assumptions but persist for modern multi-layer architectures. We will also include the empirical results on ResNet as the deployed ONN in the revised manuscript.
> > >
> > > 5. Further each theoretical convergence analysis tool comes with its assumptions and limitations. The NTK based analysis is not an exception; however, the assumptions are far more realistic than other convergence analysis tools like PL* inequality. Moreover, **NTK based analysis is more unified as it can be easily applied to NN consisting of non-smooth activation functions like ReLU.**
> > >
> > > **In summary**, the FL-NTK and lazy training framework is not intended as a limitation but as a *theoretical foundation* that can be extended.
> > >
> > > ---
> > >
> > > ### References
> > >
> > > [1] A. Jacot, F. Gabriel, and C. Hongler, “Neural tangent kernel: Convergence and generalization in neural networks,” in *Advances in Neural Information Processing Systems (NeurIPS)*, 2018.
> > >
> > > [2] S. Du, X. Zhai, B. Poczos, and A. Singh, “Gradient descent provably optimizes over-parameterized neural networks,” in *Proc. Int. Conf. Learning Representations (ICLR)*, 2019.
> > >
> > > [3] S. Arora, S. Du, W. Hu, Z. Li, R. Salakhutdinov, and R. Wang, “On exact computation with an infinitely wide neural net,” in *Advances in Neural Information Processing Systems (NeurIPS)*, 2019.
> > >
> > > [4] A. Chizat, E. Oyallon, and F. Bach, “On lazy training in differentiable programming,” in *Advances in Neural Information Processing Systems (NeurIPS)*, 2019.

---

> > > > ### Author Response · Authors · 2025-08-20
> > > > **Response 4 to Reviewer HtMh**
> > > >
> > > > **Reviewer Concern:**
> > > >   In Assumption 3, the paper assumes that the per-client update admits a sparse representation, and additionally, a sparse representation under a constant sensing matrix. This is a very strong assumption that needs to be proven or empirically verified.
> > > >
> > > > **Our Response:**
> > > > We thank the reviewer for the insightful comment. We would like to clarify both the theoretical justification and the empirical support for Assumption 3:
> > > >
> > > > **Theoretical justification for sparsity/compressibility:**
> > > > Our analysis is carried out in the lazy training regime of over-parameterized networks, which is exactly the regime where NTK theory applies. In this regime, weight changes per iteration are $\mathcal{O}(1/\sqrt{M})$ [1], [2], so the updates are small in magnitude. While this does not imply strict $\ell_0$ sparsity, it does mean that only a small fraction of components have appreciable magnitude, i.e., the updates are compressible. Prior work [3], [4] also shows that, in this regime, gradients lie in a low-dimensional subspace tied to the data geometry, which further reinforces their compressibility. This allows accurate recovery with a Gaussian sensing matrix under the Restricted Isometry Property (RIP) framework [5], as used in our COFL-MAC framework.
> > > >
> > > > We acknowledge that this justification holds under the NTK/lazy-regime assumption and for ONNs [6]. We are not claiming that all FL scenarios, especially rich-regime training, satisfy this property, and extending our framework beyond the NTK setting is an open challenge. In short, our claim should have been that under the NTK/lazy-regime assumptions, client updates are compressible. This is sufficient for the compressive sensing guarantees we use, and we will revise Sec. 1.1 to reflect this nuance.
> > > >
> > > > **Use of a constant sensing matrix:**
> > > > The assumption of using a fixed Gaussian sensing matrix is standard in compressive sensing literature [5], [7]. Under RIP, such matrices preserve the geometry of compressible vectors, enabling stable recovery with high probability. Our use of a constant matrix follows this well-established practice, and we explicitly account for recovery error in our convergence analysis.
> > > >
> > > > **Empirical verification:**
> > > > To further support Assumption 3, we provide experiments (Sec. 5) showing that the transmitted updates indeed exhibit high compressibility in practice.
> > > >
> > > > **Resolution.**
> > > > While Assumption 3 may appear strong, it is well-grounded in NTK theory, consistent with standard compressive sensing practice, and supported empirically. To address the reviewer’s concern, we will:
> > > >
> > > > - clarify in Sec. 1.1 that our assumption is compressibility rather than strict sparsity,
> > > > - explain that this follows directly from the NTK/lazy-regime setting.
> > > >
> > > > ---
> > > >
> > > > ### References
> > > >
> > > > [1] S. Du, X. Zhai, B. Poczos, and A. Singh, “Gradient descent provably optimizes over-parameterized neural networks,” in *Proc. Int. Conf. Learning Representations (ICLR)*, 2019.
> > > >
> > > > [2] Z. Allen-Zhu, Y. Li, and Z. Song, “A convergence theory for deep learning via over-parameterization,” in *International Conference on Machine Learning (ICML)*, PMLR, pp. 242–252, May 2019.
> > > >
> > > > [3] A. Chizat, E. Oyallon, and F. Bach, “On lazy training in differentiable programming,” in *Advances in Neural Information Processing Systems (NeurIPS)*, 2019.
> > > >
> > > > [4] S. Satpathi and R. Srikant, “The dynamics of gradient descent for overparameterized neural networks,” in *Proc. Learning for Dynamics and Control (L4DC)*, vol. 144, PMLR, pp. 373–384, 2021.
> > > >
> > > > [5] E. J. Candès and M. B. Wakin, “An introduction to compressive sampling,” *IEEE Signal Processing Magazine*, vol. 25, no. 2, pp. 21–30, 2008.
> > > >
> > > > [6] A. Jacot, F. Gabriel, and C. Hongler, “Neural tangent kernel: Convergence and generalization in neural networks,” in *Advances in Neural Information Processing Systems (NeurIPS)*, 2018.
> > > >
> > > > [7] Y. C. Eldar and G. Kutyniok, *Compressed Sensing: Theory and Applications*. Cambridge, U.K.: Cambridge Univ. Press, 2012.
> > > >
> > > >
> > > > ---
> > > >
> > > > **Reviewer Concern:**
> > > >   *Claim 1 and Assumption 4 are crucial to bounding the error of the compressive sensing stage. However, it seems that they are not proven nor justified.*
> > > >
> > > > **Our Response:**
> > > > We thank the reviewer for this observation. We agree that Claim 1 and Assumption 4 are central to our work. Hence, we clarify that the formal proof of the claim is provided in the Appendix for completeness. Furthermore, Assumption~1 is directly motivated by Strum’s framework, where the $\epsilon^2$-exactness property is explicitly established. Hence, our analysis builds upon a well-grounded assumption rather than introducing an unverifiable premise.
> > > >
> > > > Further, we will rewrite Claim 1 as a lemma in the revised manuscript.

---

> > > > > ### Author Response · Authors · 2025-08-20
> > > > > **Response 5 to Reviewer HtMh**
> > > > >
> > > > > **Reviewer Concern:**
> > > > > *The paper needs to consider a more realistic setting in their analysis and revise their analysis, e.g., see the weaknesses listed in the previous section.*
> > > > >
> > > > > **Our Response:**
> > > > >
> > > > > **Why we start with the simplified setting.**
> > > > > The single hidden layer NN NTK framework is the canonical and widely used foundation for theoretical analysis of ONNs [1], [2]. Extending proofs to deeper and more practical models remains an open challenge in the NTK literature itself, not only in our setting. Further, Gaussian sensing matrices are the standard benchmark in compressive sensing theory [7], [8], as they ensure clean RIP guarantees that make it possible to bound the error propagation analytically. Beginning with this tractable regime enables us to establish formal guarantees before moving toward more complex cases.
> > > > >
> > > > > **On Assumptions and Realism**
> > > > > We are happy to highlight that our convergence analysis in fact relies on fewer and milder assumptions compared to prior convergence studies in federated and distributed learning. Our major assumptions are (i) a wide over-parameterized network operating in the lazy training regime, and (ii) standard NTK dynamics, and (iii) $\epsilon^2$-C exact sparse recovery. These are standard and broadly validated conditions in modern deep learning theory [1]–[3]. In contrast, other convergence analyses typically impose stronger requirements such as global convexity and smoothness [4], Polyak–Łojasiewicz (PL*) inequalities [5], or mean-field limits [6], which are not directly satisfied by practical ReLU networks. By avoiding such restrictive assumptions, our framework is both theoretically sound and practically relevant, as it applies to widely used ReLU-based ONNs in FL-MAC.
> > > > >
> > > > > **Bridging theory and practice.**
> > > > > While the theory is developed in a simplified regime, our experiments go beyond it by evaluating COFL-MAC on deep ResNet variants under heterogeneous data splits that will be added in the revised manuscript.
> > > > >
> > > > > The following table gives an overview of Assumptions in typical convergence analyses tools.
> > > > >
> > > > > **On Assumptions**
> > > > > We emphasize that the assumptions in our analysis are in fact *mild* compared to those in prior convergence frameworks. Specifically, COFL-MAC only requires (i) a wide over-parameterized network in the **lazy training regime**, and (ii) the use of the **standard empirical squared loss** under NTK dynamics. Both are standard and well-validated in the literature [1]–[3], and directly applicable to ReLU-based ONNs used in practice.
> > > > >
> > > > > By contrast, prior analyses often rely on stronger or less verifiable conditions on the loss function, as summarized below:
> > > > >
> > > > > | Framework                   | Typical Loss Assumption                                                                 | Applicability to ReLU based FL-MAC |
> > > > > |------------------------------|------------------------------------------------------------------------------------------|-------------------------------|
> > > > > | **Convex/Smooth Analyses**   | Global convexity and smoothness of the objective (e.g., logistic or quadratic losses) [4] | Unrealistic for deep ReLU ONNs |
> > > > > | **PL\* Analyses**            | Polyak–Łojasiewicz inequality (global gradient–function value relationship) [5]           | Rarely holds in non-convex ReLU networks |
> > > > > | **Mean-Field Analyses**      | Risk functional over neuron distributions in the infinite-width limit [6]                | Typically assumes smooth activations; ReLU requires extra conditions |
> > > > > | **Our Work (COFL-MAC)**      | Standard empirical squared loss, analyzed via NTK dynamics under lazy training [1]–[3]   | Directly valid for wide ReLU ONNs; matches practice |
> > > > >
> > > > > In summary, **our framework does not impose harsh abstractions/assumptions** such as convexity, PL\*, or mean-field surrogates. Instead, it rests on realistic and widely accepted conditions, i.e., wide ONNs, lazy training, and the supervised empirical loss. These assumptions help to make the analysis tractable while keeping it directly relevant to real-world FL-MAC systems.
> > > > >
> > > > > ---
> > > > >
> > > > > ### References  - in next comment

---

> > > > > > ### Author Response · Authors · 2025-08-20
> > > > > > **Response 6 to Reviewer HtMh**
> > > > > >
> > > > > > **References**
> > > > > >
> > > > > > [1] A. Jacot, F. Gabriel, and C. Hongler, “Neural tangent kernel: Convergence and generalization in neural networks,” in *Advances in Neural Information Processing Systems (NeurIPS)*, 2018.
> > > > > >
> > > > > > [2] S. Du, X. Zhai, B. Poczos, and A. Singh, “Gradient descent provably optimizes over-parameterized neural networks,” in *Proc. Int. Conf. Learning Representations (ICLR)*, 2019.
> > > > > >
> > > > > > [3] A. Chizat, E. Oyallon, and F. Bach, “On lazy training in differentiable programming,” in *Advances in Neural Information Processing Systems (NeurIPS)*, 2019.
> > > > > >
> > > > > > [4] H. Yu, S. Yang, and S. Zhu, “Parallel restarted SGD with faster convergence and less communication: Demystifying why model averaging works for deep learning,” in *Proc. AAAI Conf. Artificial Intelligence (AAAI)*, 2019.
> > > > > >
> > > > > > [5] J. Wang, R. Das, G. Joshi, S. Kale, Z. Xu, and T. Zhang, “On the unreasonable effectiveness of federated averaging with heterogeneous data,” *arXiv preprint arXiv:2206.04723*, 2022.
> > > > > >
> > > > > > [6] Mei, S., Misiakiewicz, T. and Montanari, A., 2019, June. Mean-field theory of two-layers neural networks: dimension-free bounds and kernel limit. In Conference on learning theory (pp. 2388-2464). PMLR.
> > > > > >
> > > > > > [7] E. J. Candès and M. B. Wakin, “An introduction to compressive sampling,” *IEEE Signal Processing Magazine*, vol. 25, no. 2, pp. 21–30, 2008.
> > > > > >
> > > > > > [8] Y. C. Eldar and G. Kutyniok, *Compressed Sensing: Theory and Applications*. Cambridge, U.K.: Cambridge Univ. Press, 2012.

---

### Author Response · Authors · 2025-08-20
**Link to code**

https://drive.google.com/drive/folders/1OZyy2qp7YLM76hsoYHEwWIUbtRMmVvZ5?usp=drive\_link

---

### Decision · Action_Editor_iHP8 · 2025-09-14

**Recommendation:** Reject

**Additional Comments:**

I recommend that the authors carefully revise their paper in response to the reviewers’ comments and encourage them to submit the revised version to TMLR, ICLR, or another suitable journal or conference.

**Audience:**

Yes

**Audience Explanation:**

Overall, the paper provides interesting results for the TMLR audience, as all reviewers indicated.

**Claims And Evidence:**

No

**Claims Explanation:**

As Reviewers v8pt and HTMh pointed out in their reviews and final recommendations, the paper requires

- a clearer analysis of how sampling noise and data compressibility impact performance in the experimental setup,
- refining its theoretical assumptions to better reflect federated learning characteristics, for example, specifying client-dependent SSR conditions in Assumption 4
- the experimental evaluation is limited to a single-layer neural network setting, which does not convincingly demonstrate practical relevance.

**Resubmission Of Major Revision:**

The authors may consider submitting a major revision at a later time.